# Therapeutic Potential of Latin American Medicinal Plants in Oral Diseases: From Dental Pain to Periodontal Inflammation—A Systematic Review

**DOI:** 10.3390/ijms262311502

**Published:** 2025-11-27

**Authors:** Valentina Ramírez-Torres, Cristian Torres-León, Liliana Londoño-Hernandez, Ricardo Gómez-García, Nathiely Ramírez-Guzmán

**Affiliations:** 1Centro de Estudios e Investigaciones Interdisciplinarios, Universidad Autónoma de Coahuila (UAdeC), Arteaga 25280, Coahuila, Mexico; valentinaramirez@uadec.edu.mx; 2Centro de Investigación y Jardín Etnobiológico, Universidad Autónoma de Coahuila (UAdeC), Viesca 27480, Coahuila, Mexico; 3Facultad de Ciencias Básicas, Tecnología e Ingeniería, Universidad Nacional Abierta y a Distancia (UNAD), Palmira 763531, Colombia; liliana.londono@unad.edu.co; 4Laboratório Associado, Escola Superior de Biotecnologia, CBQF—Centro de Biotecnologia e Química Fina, Universidade Católica Portuguesa, Rua Diogo Botelho 1327, 4169-005 Porto, Portugal; rgarcia@ucp.pt; 5Facultad de Ciencias Biológicas, Universidad Autónoma de Coahuila (UAdeC), Torreon 27276, Coahuila, Mexico

**Keywords:** dental pain, gingivitis, periodontitis, periodontal inflammation, wound healing, Latin American medicinal plants

## Abstract

Oral diseases pose a major public health challenge, especially in low-income countries where dental care is limited due to high costs. In this context, phytotherapy has gained attention as a complementary approach due to its bacteriostatic, anti-inflammatory, healing, and analgesic properties. These therapeutic effects are mainly attributed to plant-derived bioactive metabolites, which interact with cellular structures, especially the plasma membrane, to modulate inflammation, stimulate tissue regeneration, and support antimicrobial defense. This review systematically examined the scientific literature to identify Latin American medicinal plants with therapeutic potential in dentistry. Based on their clinical and ethnobotanical applications, the analysis focused on species with anti-inflammatory, healing, analgesic, and relaxing effects, particularly in conditions such as dental pain, gingivitis, and periodontitis. Given the close relationship between pain, inflammation, and periodontal disease, these conditions cannot be studied in isolation. Gingivitis and periodontitis often present with painful symptoms and inflammatory responses that overlap with mechanisms of tissue damage and repair. Therefore, broadening the scope of this review allows for a more comprehensive understanding of how Latin American medicinal plants can contribute not only to pain relief but also to periodontal health, inflammation control, and wound healing. Fifty plant species were identified. Among these, 35 exhibited anti-inflammatory activity, 28 had healing properties, 20 showed analgesic effects, and 12 were associated with relaxing properties. Mexico accounted for the highest proportion of species (60%), followed by Colombia and Peru (54%) and then Brazil (32%). These percentages represent the proportion of plant species reported in studies originating from each country, relative to the total number of species identified in the review. The most studied species were *Salvia rosmarinus* Spenn. (Lamiaceae), *Moringa oleifera* Lam. (Moringaceae), *Aloe vera* (L.) Burm.f. (Asphodelaceae), and *Ocimum basilicum* L. (Lamiaceae). Latin American medicinal plants demonstrate strong potential not only in dental therapy but also in the management of periodontal inflammation and oral diseases. However, further research and clinical validation are needed to ensure their safe integration into conventional treatments.

## 1. Introduction

Oral diseases constitute a major global public health concern due to their widespread prevalence and the substantial costs associated with their treatment. These conditions include dental caries, periodontal disease, congenital anomalies such as cleft lip and palate, dental trauma, and hereditary disorders, many of which are strongly influenced by socioeconomic determinants like poverty, limited access to healthcare services, school dropout, and harmful habits such as tobacco use and inadequate nutrition [1]. According to the World Health Organization (WHO), these diseases, along with oral and pharyngeal cancer, remain highly prevalent in industrialized countries and disproportionately affect low-income populations with restricted access to dental care [1]. The economic burden is considerable, as expenditures on oral healthcare account for approximately 5% to 10% of total health costs in various countries [2].

In response to these challenges, interest has grown in complementary and alternative strategies for oral healthcare. Indigenous communities have traditionally used medicinal plants to treat various ailments, including oral conditions [3]. Their bioactive compounds interact with cellular structures, particularly the plasma membrane, helping to regulate inflammation, stimulate tissue regeneration, and enhance antimicrobial defenses. Phytotherapy has demonstrated bacteriostatic, anti-inflammatory, wound-healing, and analgesic effects, suggesting its potential as an adjunct in dental treatment protocols [4].

Despite these promising properties, most research has focused on plant species used in Europe and Asia [5]. Although rich in biodiversity and cultural knowledge of plant-based remedies for oral health, Latin America remains underrepresented in the scientific literature [6]. A systematic review of Latin American species is needed to fill this gap and provide a structured overview of their therapeutic potential in dentistry. Therefore, this review aims to identify, analyze, and highlight the relevance of medicinal plants native to Latin America in the prevention and management of oral diseases.

## 2. Materials and Methods

This review followed the PRISMA-ScR (Preferred Reporting Items for Systematic Reviews and Meta-Analyses Extension for Scoping Reviews) guidelines to ensure methodological transparency and reproducibility [7]. Although no formal protocol was registered in platforms such as OSF or INPLASY, the review adhered to the core principles of systematic literature mapping.

A systematic literature search was conducted to identify and analyze Latin American medicinal plants with therapeutic potential in dental health. The search was performed in specialized scientific databases to ensure access to peer-reviewed and reliable sources. The consulted databases included Scielo, PubMed, Revista de Salud Pública del Paraguay, RCOE, Revista Facultad de Odontología Universidad de Antioquia, REDEL, Revista Granmense de Desarrollo Local, Revista Odontológica Mexicana, Farmacia Profesional, Farmacia Clínica, and Avances en Odontoestomatología. Additionally, Google Scholar and ResearchGate were used to broaden the scope of the literature search.

The complete search strategies, including search strings, date ranges, language limits, and the number of records retrieved from each database, are provided in Appendix A for transparency and reproducibility.

In the case of Google Scholar, the search yielded a considerably high number of records (over 6000). However, this result reflects the broad and unspecific nature of the platform, which includes duplicated literature, non-relevant documents, and sources of variable quality. For this reason, not all retrieved records were reviewed; instead, the predefined inclusion and exclusion criteria were rigorously applied. Through this process, only articles pertinent to the review’s objective were selected, ensuring a systematic and transparent screening while acknowledging the inherent limitations of Google Scholar as a search engine.

The literature search and screening process are summarized in Figure 1, following the PRISMA-ScR guidelines for scoping reviews. The search strategy included the following keywords: “Dental pain,” “Gingivitis,” “Periodontitis,” “Periodontal inflammation,” “Wound healing,” and “Latin American medicinal plants.”

Original research articles, literature reviews, and academic theses published between 2000 and 2025 in Spanish, English, or Portuguese were included. Eligible documents had to address the use of medicinal plants, phytotherapy, or herbal medicine in the dental field, covering aspects such as oral health, dental pain management, gingivitis, periodontitis, inflammatory processes, and wound healing of oral tissues. Only studies conducted in Latin American countries or including relevant applications in this region were considered. Conversely, reports unrelated to dentistry, even if they mentioned medicinal plants, were excluded, as well as grey literature without full-text access or peer review, duplicate articles identified through a reference manager and manual verification, and documents that did not provide data related to the therapeutic effects of interest (anti-inflammatory, antimicrobial, or healing) in oral health, as well as studies conducted outside the geographical scope of interest (Latin America).

All retrieved references from the different databases were exported in RIS format and imported into the reference manager Mendeley. The software’s duplicate detection tool was applied, followed by manual verification, to ensure the removal of duplicate records. This deduplication process reduced the total number of references before the screening stage.

A total of 6985 records were identified across all databases and journals. After removing 500 duplicates using Mendeley and manual verification, 6485 unique records remained. 6200 were excluded during the title and abstract screening stage because they did not meet the eligibility criteria. A total of 285 articles were assessed in full text, of which 222 were excluded due to reasons such as lack of relevance to oral health, incomplete data, absence of therapeutic outcomes of interest, or failure to correspond to the geographical scope of Latin America. Ultimately, 63 studies met all the inclusion criteria and were included in the review. The selection process is illustrated in the PRISMA-ScR flow diagram (Figure 1).

Records were identified through databases and journals (n = 6985). After removing duplicates (n = 500), 6485 records were screened, and 6200 were excluded. A total of 285 full-text articles were assessed for eligibility; 222 were excluded (not relevant to oral health, incomplete data, no therapeutic outcomes, or outside Latin America). Sixty-three studies were included in the final review.

In addition, a brief appraisal of evidence quality and study type was conducted to classify all included sources. In line with the reviewer’s recommendation, peer-reviewed clinical trials, in vivo/in vitro studies, and systematic reviews were prioritized, while reports and institutional documents were categorized as grey literature. Although a total of 63 studies met the inclusion criteria and were included in the review, a focused subset of 36 studies, selected based on methodological quality, relevance to dental applications, and completeness of outcome data, was appraised in greater detail.

This approach allowed us to emphasize higher-quality evidence without diminishing the relevance of the remaining studies. The detailed classification of this subset is provided in Appendix A, which summarizes the evidence level and main characteristics of the 36 references.

After applying these filters, 63 relevant articles were selected, documenting 50 plant species native to Latin America with potential applications in dentistry. The analysis revealed that Mexico, Colombia, and Peru exhibited the greatest diversity of reported species, with propolis, rosemary, moringa, aloe vera, and basil being the most frequently studied for their therapeutic benefits. A detailed list of all 50 plant species and their reported uses is provided in Table 5. In addition, a small number of species not originally native to Latin America (e.g., *Zingiber officinale*, *Ocimum sanctum*) were retained and explicitly labeled as external comparators. These plants were included in the synthesis for contextual purposes; however, it is essential to note that they are also cultivated and widely used in Latin America, as documented in the literature, which justifies their inclusion in the review.

Each plant demonstrated specific effects on oral health:*Propolis* (resinous product of *Apis mellifera* L., Apidae): Antimicrobial and anti-inflammatory activity, effective against gingivitis and periodontitis.*Salvia rosmarinus* Spenn. (Lamiaceae): Used for pain relief and wound healing in the oral cavity.*Moringa* (*Moringa oleifera*): Anti-inflammatory effects beneficial to oral tissues.*Aloe vera* (L.) Burm.f. (Asphodelaceae) and *Ocimum basilicum* L. (Lamiaceae): Known for healing and antibacterial properties relevant to disease prevention and oral care.

This review was based entirely on previously published literature and did not involve any new experiments or interventions involving human or animal subjects.

## 3. Pain and Its Relationship with Dentistry

The International Association for the Study of Pain (IASP) defines pain as a distressing sensory and emotional state resulting from actual or perceived tissue damage. Beyond its physiological component, pain impacts multiple aspects of daily life, including work, family, and social functioning, ultimately reducing overall well-being. Since pain is inherently subjective, it must be evaluated through self-reported measures, as no absolute objective indicators exist for its confirmation [8].

Pain perception varies significantly according to its etiology, intensity, and each individual’s pain threshold. Although it cannot be measured precisely, standardized assessment tools such as the Visual Analog Scale (VAS) allow for approximate evaluation and improved clinical management. Only Colombia and Australia have formally included the VAS in their national clinical protocols [8].

Oral pain is one of the most frequent discomforts in the facial and cervical regions, commonly associated with pulpal pain due to the dental pulp’s sensitivity to harmful or infectious stimuli [9]. This form of pain is classified as deep somatic, typically characterized by a diffuse, dull, and throbbing quality. Given the extensive innervation of oral structures, particularly by the trigeminal nerve (cranial nerve V), oral pain is often intense and may radiate toward the ears, eyes, or adjacent areas. This complexity underscores the need for pain management strategies that extend beyond conventional pharmacological methods.

Medicinal plants have emerged as promising alternatives for managing dental pain in this context. Various bioactive compounds found in *Salvia rosmarinus* Spenn. (Lamiaceae), *Moringa oleifera* Lam. (Moringaceae), and *Aloe vera* (L.) Burm.f. (Asphodelaceae) have shown analgesic effects by modulating inflammation and regulating nociceptive pathways. These phytotherapeutic agents interact with pain mediators such as prostaglandins, potentially reducing the intensity of orofacial pain [10].

The IASP also emphasizes that oral pain is a sensory and emotional reaction to real or perceived harm affecting the orofacial region [8]. As scientific evidence expands, incorporating natural compounds into dental pain protocols may offer effective alternatives that reduce dependence on synthetic analgesics and help minimize their adverse effects.

## 4. Causes of Dental Pain

Dental pain can originate from various pathological conditions affecting the oral cavity, including infections, trauma, odontalgia, somatic pain involving the mucosa and periodontium, nutritional deficiencies (such as vitamin shortages), and autoimmune diseases [11]. These conditions compromise the structural integrity of dental tissues and support systems, resulting in different types of pain depending on the location and pathophysiology of the lesion.

Physiologically, dental pain is classified as either somatic or neuropathic. Somatic pain is the most common, affecting the gingiva, periodontium, blood vessels, and maxillary bones. It is usually throbbing and localized [12]. Neuropathic pain, by contrast, arises from direct injury to nerve fibers and is often described as diffuse and radiating. A typical example is pulpal pain when the lesion reaches the neurovascular bundle within the dental pulp [13].

An inflammatory response is triggered when oral tissue damage occurs, activating nociceptive pathways. The inflammatory response begins with the release of mediators such as prostaglandin E2 (PGE2) and bradykinin, which sensitize nerve fibers and facilitate pain transmission [12]. Neuropeptides such as substance P and CGRP further promote the release of histamine and cytokines, intensifying the pain experience [12]. The regulation of these inflammatory mediators is a key target of emerging therapeutic strategies, including the use of natural compounds with analgesic potential; however, further research is needed to validate their clinical effectiveness.

Pain is considered chronic when it is persistent and acute when it is sudden and short-term. Despite being subjective, pain intensity can be estimated using tools such as the Visual Analog Scale (VAS), which rates perception on a scale of 1 to 10 [12].

Beyond biological triggers, psychological factors such as stress, anxiety, and past pain experiences can significantly influence pain perception and treatment outcomes [14]. Furthermore, dental pain can undermine the integrity of enamel and dentin, increasing the risk of pulpitis, a severe inflammation caused by trauma, fracture, or infection.

To further strengthen this clinical framework, it is essential to note that several studies in periodontology and implantology have already employed standardized pain assessment scales. For example, one trial assessed postoperative discomfort, dentin hypersensitivity, and patient-reported pain during different periodontal procedures using the Visual Analog Scale (VAS). Patients marked their pain and sensitivity on the VAS, enabling comparisons across scaling and root planing (SRP), modified Widman flap (MWF), osseous flap (OF), and gingivectomy (GV). Pain scores were significantly higher after OF and GV compared with SRP and MWF, and all surgical procedures caused more dentin hypersensitivity than non-surgical therapy [15].

Additionally, larger clinical investigations have incorporated the Numeric Rating Scale (NRS) to assess pain intensity. In one study of 253 patients undergoing 330 periodontal or implant surgeries, most participants reported mild pain (70.3%), while 25.5% had moderate and 4.2% severe pain. Advanced implant surgeries showed the highest median NRS scores (4.0), while open flap debridement had the lowest (1.0). Pain duration (median = 2 days) and analgesic use were positively correlated with NRS values, and more extensive or complex surgeries, as well as higher anesthetic volumes, were associated with greater pain [16].

Table 1 outlines the multifactorial nature of dental pain, which can arise from pulpal lesions and surrounding anatomical structures. For instance, pericoronitis, which often affects erupting third molars, can lead to localized inflammation, pain, and difficulty chewing if left untreated. Similarly, infectious alveolitis resulting from the dislodgement of a blood clot post-extraction exposes the alveolar bone and causes intense, persistent pain that requires clinical management.

Pain may also stem from sinusitis, particularly when the oral and sinus cavities are connected, allowing oral bacteria to infect the sinus mucosa.

In addition, temporomandibular joint (TMJ) disorders are frequent sources of orofacial pain. Displacement of the articular disk affects mandibular movement, producing symptoms such as jaw pressure, clicking sounds, and difficulty chewing [19].

## 5. Conventional Pharmacology for Pain Management

The first step involves a thorough clinical evaluation when a patient presents to the dental office with pain or discomfort. This process includes obtaining a comprehensive medical history, identifying potential etiological factors, and assessing contraindications for pharmacological interventions.

In conventional dentistry, pain management typically involves commonly prescribed medications such as acetaminophen (paracetamol), nonsteroidal anti-inflammatory drugs (NSAIDs) like ibuprofen and naproxen, and, in more severe cases, opioids or their derivatives. Drug selection is guided by the severity of pain and the patient’s overall medical condition. While effective, these drugs are often associated with adverse effects: NSAIDs may cause gastrointestinal irritation, opioids can lead to sedation and dependency, and excessive intake of paracetamol carries the risk of hepatotoxicity.

Pharmacological strategies in dental pain management rely heavily on cyclooxygenase (COX) inhibitors, particularly NSAIDs, because they block prostaglandin synthesis and reduce inflammation and pain [22]. However, since prostaglandins also protect the gastric mucosa and regulate renal function, prolonged use of NSAIDs can lead to ulcers, kidney dysfunction, or gastrointestinal bleeding, especially in vulnerable patients.

Although paracetamol is not classified as an NSAID, it inhibits prostaglandin synthesis centrally, offering analgesic and antipyretic effects [22]. While generally safe, overdose may result in liver toxicity, requiring careful dosing. Metamizole is another analgesic option, though its use is restricted due to potential risks such as allergic reactions and hematologic complications like agranulocytosis [22].

In cases where other medications are insufficient, opioids may be prescribed. These drugs bind to specific receptors in the central and peripheral nervous systems, providing potent analgesia [22]. However, their use in dentistry remains limited due to the risk of adverse effects, including respiratory depression, sedation, dependence, nausea, and constipation, which require close monitoring.

Additionally, combination analgesics, often including substances like caffeine or sedatives, have been developed to enhance efficacy. Although they can improve pain relief in certain clinical scenarios, their use remains controversial due to concerns about drug interactions and potential side effects [22].

Given these limitations, current research has shifted toward identifying complementary therapies that can enhance pain control while minimizing adverse outcomes. In this context, natural bioactive compounds are gaining relevance, and several studies are evaluating their potential role in modulating dental pain.

## 6. Use of Medicinal Plants for Reducing Oral Pain

The search for therapeutic alternatives with fewer adverse effects than conventional drugs has fueled a growing interest in medicinal plants. These contain bioactive compounds that promote healing by stimulating the production of collagen and proteoglycans, both of which are essential for tissue regeneration. According to the World Health Organization (WHO), more than 201 countries regulate and use medicinal plants for their therapeutic and nutritional benefits [1].

At the cellular level, the plasma membrane plays a critical role in mediating interactions between plant-derived compounds and human cells. This dynamic structure regulates the passage of molecules between the intracellular and extracellular environments. Surrounding each cell, the extracellular matrix (ECM) supports cell communication and molecular exchange. Structurally, the ECM and the basement membrane (BM) share components such as type IV collagen and polysaccharide elements, which are also found in plant cell walls. This resemblance may enhance the bioavailability of plant-based compounds and improve their therapeutic effects, especially in the oral cavity, where the mucosal barrier facilitates rapid absorption.

Among the most studied plant-derived compounds are flavonoids, which are well-known for their anti-inflammatory activity. These molecules modulate the production of inflammatory cytokines, such as interleukins IL-1 and IL-6, and tumor necrosis factor-alpha (TNF-α), thus reducing inflammation and protecting oral tissues. Recent studies have suggested that flavonoids and other secondary metabolites may modulate inflammatory signaling pathways associated with orofacial pain, which supports the findings of this review.

Despite structural differences, plant and human cells share key biological mechanisms in their membranes that allow for the interaction and transport of proteins, enzymes, and other molecules through specific transport systems. Mechanisms such as exocytosis and endocytosis enable the absorption of plant compounds into oral tissues, allowing their integration into metabolic pathways [23]. This cellular interaction helps explain the reported effects of medicinal plants on pain relief and tissue repair in the oral cavity.

Furthermore, the plant cell wall and the human ECM are similar in regulating molecular exchange and cellular signaling. These functional parallels are particularly relevant to natural medicine, providing insight into how specific plant compounds can interact with oral mucosal cells to exert therapeutic effects. For example, flavonoids and alkaloids may cross these cellular barriers via facilitated diffusion or active transport mechanisms that could explain their growing interest in oral healthcare [23]. Overall, the structural and functional interaction between plant and human cells supports the therapeutic potential of natural compounds in oral health. Nonetheless, further clinical studies are needed to determine the full extent of their effects on pain modulation and tissue regeneration.

## 7. Dental Products

Dental research has increasingly explored the use of natural products as alternatives to conventional treatments, aiming to mitigate the adverse effects of antibiotics and chemical antiseptics. According to the World Health Assembly, traditional medicine plays a fundamental role in managing oral diseases in many developing countries [1]. In this context, several studies have evaluated the impact of medicinal plants on oral health, highlighting their antibacterial, anti-inflammatory, and wound-healing properties.

One key study demonstrated that mouth rinses containing 5% *Echinacea purpurea* (L.) Moench (Asteraceae), *Commiphora myrrha* (Nees) Engl. (Burseraceae), and *Matricaria chamomilla* L. (Asteraceae) significantly reduced bacterial plaque and gingival bleeding. Similarly, *Aloe vera* (L.) Burm.f. (Asphodelaceae), and grapefruit seed extract rinses showed superior efficacy in controlling gingivitis compared to distilled water. Additionally, 1% Sage extract, recognized for its anti-plaque properties, has been successfully used as a rinse to enhance oral hygiene. In endodontic treatments, 3.3% *Carica papaya* oil has been shown to enhance dental permeability and exert notable antiseptic effects. Likewise, *Matricaria chamomilla* L. (Asteraceae) (syn. *Matricaria recutita* L.)

Extracts have demonstrated outcomes comparable to those of chlorhexidine, effectively reducing inflammation and controlling pathogenic bacteria. In animal models, *Propolis*—a resinous product of *Apis mellifera* L. (Apidae)—has also exhibited wound-healing activity without adverse effects [22].

In line with these findings, additional randomized clinical trials provide further support for the efficacy of herbal formulations. For instance, in a randomized clinical trial involving 70 hospitalized patients (mean age, 44.9 years; 67.1% male), *Echinacea purpurea* (L.) was compared with chlorhexidine and a control group for 4 days. Both echinacea and chlorhexidine significantly reduced oral microbial flora compared with the control, supporting echinacea as a viable alternative [24]. Similarly, a randomized, double blind, placebo-controlled pilot trial including 30 orthodontic patients (10–40 years old) demonstrated that a 1% *Matricaria chamomilla* L. (Asteraceae) mouthwash reduced plaque (−25.6%) and gingival bleeding (−29.9%) comparably to 0.12% chlorhexidine, while the placebo showed increased indices. *Matricaria chamomilla* L. (Asteraceae) was well tolerated without adverse effects, reinforcing its role as a safe herbal alternative in the management of gingivitis [25].

From an ethnobotanical perspective, indigenous communities in Colombia’s Atrato Medio Antioqueño region have reported using species such as *Piper*, *Manekia*, and *Schradera* to strengthen teeth, prevent caries, and control periodontal pockets. Chewing these plants releases bioactive metabolites with antimicrobial potential [23]. The incorporation of these traditional practices reinforces the potential therapeutic role of plant-based compounds in modulating oral health, aligning with the findings presented in this review.

Other investigations have evaluated the use of phytotherapeutic compounds in treating gingivitis and periodontitis. For instance, *Solanum lycopersicum* L. (Solanaceae) (tomato plant) in gel form has shown anti-hemorrhagic and anti-inflammatory effects, while *Eucalyptus*, incorporated into chewing gum, has reduced plaque accumulation and gingival inflammation [26]. These findings highlight the increasing interest in developing natural formulations as adjunctive therapies to conventional dental protocols.

Moreover, mouthwashes containing various plant extracts have shown promising results. *Solanum lycopersicum* (tomato plant) in gel form has shown anti-hemorrhagic and anti-inflammatory effects, while *Eucalyptus globulus* Labill. (Myrtaceae), incorporated into chewing gum, it has been shown to reduce plaque accumulation and gingival inflammation. Likewise, *Mangifera indica* L. (Anacardiaceae).

Gel has anti-inflammatory and wound-healing properties, offering therapeutic benefits in the management of chronic periodontitis [26].

Finally, a clinical trial conducted on high school students using a *Zingiber officinale* Roscoe (Zingiberaceae) based mouth rinse showed a significant reduction in plaque and gingival bleeding compared to the control group, in which inflammation levels remained unchanged [27]. While these findings suggest notable therapeutic potential, further research is necessary to standardize formulations, optimize dosage, and assess long-term clinical effectiveness.

## 8. Mechanism of Action of Plants in Inflammation

Conventional drugs have shown effectiveness in treating inflammatory diseases; however, prolonged use often results in adverse effects such as fluid retention and hypertension. This has increased interest in medicinal plants, whose bioactive compounds provide anti-inflammatory alternatives with a lower risk of side effects. Chronic inflammation, one of the major contributors to global mortality, is commonly managed using NSAIDs and corticosteroids, yet their long-term impact has often been underestimated [27]. The discomfort caused by symptoms such as edema, redness, and functional impairment has driven patients to seek more accessible natural therapies. As a result, interest has grown in studying plant-based bioactive molecules that can modulate inflammatory responses, paving the way for new complementary approaches in pain management [27].

At the biological level, inflammation is an immune response triggered by chemical, biological, or physical stimuli that affect vascularized connective tissues. This process involves the release of pro-inflammatory cytokines and plays a central role in maintaining oral health, particularly in chronic conditions such as periodontal disease [27]. Identifying natural compounds that interact with these inflammatory mediators is gaining increasing attention, especially for their potential in pain relief and tissue regeneration [28].

As illustrated in Figure 2, inflammation begins with the release of mediators by mast cells, which promote vascular changes and the recruitment of immune cells to the affected site. These processes initiate an inflammatory cascade that is later regulated by feedback mechanisms, ultimately facilitating wound healing and restoring tissue integrity.

Non-steroidal anti-inflammatory drugs (NSAIDs) inhibit cyclooxygenase 1 and 2 (COX-1 and COX-2). While COX-2 is directly linked to inflammation at the injury site and its inhibition is therapeutically beneficial, COX-1 plays essential physiological roles, such as protecting the gastric mucosa. Therefore, its inhibition may cause adverse effects. These limitations have spurred the search for alternative treatments with fewer risks, encouraging research into plant-based anti-inflammatory compounds [28].

To provide a mechanistic integration of plant bioactives, Figure 3 presents a flowchart mapping plant metabolites to molecular targets and oral outcomes. This schematic illustrates how compounds such as phenols, flavonoids, saponins, and vitamins modulate pathways (e.g., COX/LOX inhibition, NF-κB downregulation, TRP modulation), ultimately leading to improved oral outcomes, including reduced plaque, gingivitis, pain, and enhanced healing.

Although inflammation is necessary for tissue repair, its dysregulation can extend tissue damage. The natural resolution phase involves the production of prostaglandins and leukotrienes, which reduce neutrophil numbers and enhance monocyte activity, favoring tissue regeneration. This understanding has led to an interest in therapies that modulate rather than completely inhibit inflammation, contrasting with the mechanism of NSAIDs [9].

Additionally, acupuncture has been investigated as a complementary method for managing dental pain. This technique stimulates the release of endogenous opioid peptides, such as β-endorphins, enkephalins, neoendorphins, and dynorphins, through the nervous system, resulting in analgesic effects. These peptides can also be released in the digestive tract and adrenal medulla, broadening their therapeutic potential [29].

Beyond these general mechanisms, recent evidence has identified specific plant-derived metabolites that target well-characterized molecular receptors in oral tissues. These examples offer mechanistic insight and underscore the therapeutic relevance of natural compounds in orofacial inflammation.

### Mechanisms of Action of Selected Plant Metabolites in Oral Inflammation

*Salvia rosmarinus* Spenn. (Lamiaceae). Rosmarinic acid, the predominant bioactive compound in Rosmarinus officinalis, has been linked to anti-inflammatory activity in oral contexts. Studies on human gingival fibroblasts challenged with lipopolysaccharides revealed that it reduces oxidative stress, preserves intracellular glutathione, and suppresses the expression of pro-inflammatory mediators such as IL-1β, IL-6, TNF-α, and iNOS, an effect linked to the inhibition of NF-κB signaling during periodontal inflammation [29]. Evidence in vascular smooth muscle cells also indicates that rosmarinic acid downregulates MAPK (ERK, JNK, p38) and NF-κB pathways, leading to a lower release of inflammatory mediators, including iNOS, TNF-α, and IL-8 under LPS stimulation. Although these results were obtained in non-oral systems, they support the molecular plausibility of this compound in modulating signaling cascades relevant to periodontal disease [30].

*Moringa oleifera* Lam. (Moringaceae) is a rich source of phytochemicals such as quercetin and isothiocyanates, which act on inflammatory pathways important in oral health. In a rat model of periodontal inflammation, ethanolic leaf extracts enriched with quercetin attenuated tissue damage by suppressing NF-κB activation, thereby reducing pro-inflammatory cytokines and supporting periodontal regeneration [31]. Moringa isothiocyanate-1 (MIC-1) has also been shown to influence Nrf2 and NF-κB signaling in LPS-induced inflammation, thereby lowering the expression of IL-6, IL-1β, and TNF-α, while alleviating oxidative stress. Beyond preclinical findings, a randomized controlled trial involving 36 patients with gingivitis demonstrated that a Moringa oleifera mouthwash significantly decreased plaque and gingival indices after 14 days of use, showing efficacy comparable to chlorhexidine and superior to saline [32]. Altogether, this evidence provides a strong mechanistic and clinical rationale for Moringa oleifera as a natural adjunct in periodontal therapy.

*Aloe vera* (L.) Burm.f. (Asphodelaceae). Acemannan, the main polysaccharide of Aloe vera, exhibits immunomodulatory properties within oral environments. In human gingival fibroblasts, acemannan binds to Toll-like receptor 5 (TLR5), activating NF-κB signaling and promoting the release of IL-6 and IL-8. This process not only enhances tissue repair but also strengthens host immune defenses in periodontal tissues, offering dual benefits in regeneration and antimicrobial protection [33]. More broadly, Aloe vera and its major active metabolites, including acemannan and aloin, are recognized for their anti-inflammatory, antioxidant, and wound-healing properties, reinforcing their potential applications in oral medicine [34].

*Ocimum basilicum* L. Linalool, a monoterpene present in *Ocimum basilicum*, has been shown to exert immunomodulatory effects relevant to periodontal inflammation. In vitro studies demonstrated that linalool reduces the overproduction of pro-inflammatory cytokines (IL-6, IL-10, IL-17, IFN-γ) in human immune cells exposed to Porphyromonas gingivalis, a key pathogen in periodontitis. These outcomes are linked to the downregulation of the NF-κB and MAPK signaling pathways, which control the transcription of inflammatory genes. By moderating excessive immune responses without impairing cell viability, linalool may protect periodontal tissues from chronic inflammatory damage [35].

*Propolis* is a resinous product of *Apis mellifera* L. (Apidae). Current research highlights the role of caffeic acid phenethyl ester (CAPE), the principal phenolic constituent of propolis, in regulating inflammatory activity in periodontal cells. In human gingival fibroblasts stimulated with LPS and IFN-α, CAPE markedly suppressed TNF-α, IL-6, and IL-8 production, with no cytotoxicity observed. In contrast, ethanolic propolis extract (EEP) produced weaker and less consistent effects on cytokine release. These findings suggest that CAPE exerts a more selective and potent anti-inflammatory effect, supporting its potential application in controlling periodontal inflammation [30].

## 9. Internal Inflammatory Process

Inflammation within the oral cavity is frequently associated with periodontal pathogens, including *Porphyromonas gingivalis*, *Prevotella intermedia*, and *Actinobacillus actinomycetemcomitans*. These microorganisms stimulate the production of various inflammatory mediators, including cytokines, leukotrienes, prostaglandins, and thromboxanes, which are secreted by immune cells, epithelial cells, and fibroblasts. These mediators initiate the periodontal inflammatory cascade. Under normal physiological conditions, this response facilitates the elimination of harmful stimuli and promotes tissue healing. However, when inflammation becomes dysregulated or chronic, it can destroy periodontal tissues and lead to progressive bone resorption [9].

Among the inflammatory mediators, cytokines play a central role in modulating both inflammation and healing by balancing pro-inflammatory and anti-inflammatory signaling. Chemokines are also crucial, as they attract immune cells to the site of infection and contribute to bone remodeling through the activation of osteoclasts. Although essential in orchestrating the inflammatory response, prostaglandins may exacerbate tissue damage when overproduced, particularly by enhancing bone resorption. Additionally, extracellular matrix metalloproteases involved in tissue repair can lead to structural degradation and tissue weakening if produced in excess [9].

## 10. Properties of Medicinal Plants

Interest in medicinal plants has increased substantially in recent years, driven by their accessibility and the growing awareness of the adverse effects of conventional pharmaceuticals. It is estimated that there are approximately 260,000 plant species worldwide, of which around 10% possess documented medicinal properties [36]. Their integration into healthcare, including dental applications, has contributed to the development of alternative treatments with reduced side effect profiles.

However, using these plants requires accurate botanical knowledge regarding their identification, preparation, and administration. Although they are of natural origin, excessive or improper use can result in toxicity. It is essential to accurately identify the plant species, understand its origin, and determine which part of the plant—root, stem, leaves, or flowers—should be used, as each contains distinct bioactive compounds with varying therapeutic effects. This ancestral knowledge, transmitted through oral tradition, literature, and scientific research, has been crucial in expanding the scope of natural medicine. In many developing countries, these practices remain particularly relevant as accessible alternatives for treating various health conditions [36].

## 11. Biodiversity of Medicinal Plants in Latin America

Latin America is one of the most biodiverse regions in the world, offering a vast supply of raw materials for the pharmaceutical industry. However, despite this potential, the lack of regulation and international recognition has limited the sustainable development of these medicinal resources. Commercializing natural compound-based drugs often undervalues local raw materials, fostering an extractivist model that threatens biodiversity and compromises crop quality.

In response, various initiatives have emerged to regulate and promote the sustainable production of these plants. Within this framework, the Ibero-American Program of Science and Technology for Development has supported agroecological strategies to ensure sustainability in medicinal plant cultivation. Nonetheless, the primary challenge remains the revaluation of these crops, ensuring fair pricing that fosters responsible practices and contributes to regional biodiversity conservation [37].

## 12. Regulation of Medicinal Plants in Colombia

Natural and herbal medicine is recognized by the World Health Organization (WHO) as a safe and effective alternative for treating various conditions. However, its integration into national healthcare systems remains challenging due to economic and structural barriers. The long-term objective is to make this medicine accessible for managing common ailments, such as stomach pain and influenza [38]. Although traditionally empirical, natural medicine requires scientific validation to prevent adverse effects and interactions with conventional drugs and food. Demonstrating its efficacy is key to enabling its proper integration into the healthcare system [38].

In Colombia, Law 1164 of 2007 regulates alternative medicine by establishing a specialized committee and permitting its practice exclusively by certified healthcare professionals. Practices such as herbology, acupuncture, and moxibustion are categorized as complementary therapies, although they are not currently included in the main regulatory committee [39]. The Ministry of Health and Social Protection has promoted the Technical Guidelines for integrating Alternative and Complementary Medicines and Therapies (MTAC), supporting their inclusion in primary healthcare and alignment with the General System of Social Security in Health (SGSSS). Laws 1438 of 2011 and 1752 of 2015 further support this framework in accordance with WHO recommendations.

## 13. Medicinal Plants for Inflammation and Gingival Problems

Medicinal plants have shown promising effects in managing inflammation and gingival conditions due to their antimicrobial, anti-inflammatory, and healing properties. These plant species are utilized in various formulations, including infusions, mouthwashes, and topical applications, to alleviate oral symptoms and stimulate tissue regeneration. As summarized in Table 2, several plants are commonly used for gingival disorders across Latin America and other regions, highlighting their pharmacological diversity.

All dosages included in this table are supported by clinical studies, and whenever possible, standardized regimens are indicated. Additionally, a concise safety summary is provided for each plant, highlighting contraindications, potential drug interactions (e.g., anticoagulants), pregnancy-related considerations, hypersensitivity, and potential mucosal irritation. This approach ensures that the therapeutic use of these medicinal plants is presented with both efficacy and safety considerations, thereby enhancing the clinical relevance of the information.

## 14. Medicinal Plants for Periodontal Problems

Various medicinal plants have been effectively used to treat periodontal conditions due to their anti-inflammatory, antimicrobial, and healing properties. These plants are used in various presentations, including gels, rinses, infusions, and topical creams. As detailed in Table 3, several species contribute to managing biofilm accumulation, gingival inflammation, and tissue regeneration through their bioactive compounds.

The data presented in this table consolidate available clinical evidence on preparations and doses of medicinal plants for periodontal problems. Additionally, a brief overview of safety aspects is provided, highlighting relevant contraindications, potential drug interactions, and pregnancy considerations. By combining therapeutic potential with safety information, the table aims to offer a balanced reference that can guide clinical decision-making.

## 15. Results and Discussion

The classification of all cited references by source type is presented in Appendix A. The present review identified 50 plant species with therapeutic, anti-inflammatory, healing, and analgesic properties applicable to oral health, particularly in conditions affecting dental support tissues such as bone, periodontal ligament, gingiva, and surrounding periodontal areas. In terms of geographic distribution, our analysis revealed that Mexico had the highest biodiversity (60%), followed by Colombia and Peru (54%) and Brazil (32%). These percentages represent the proportion of plant species reported in studies from each country relative to the total number of species identified in this review.

Within our dataset, the most extensively studied species with high therapeutic potential were propolis, rosemary, moringa, aloe vera, and basil. Among these, aloe vera and propolis already have clinical evidence supporting their use, whereas basil and moringa are still supported mainly by preclinical and in vitro data, highlighting the need for further clinical validation.

Oral pain is one of the most frequent and intense clinical manifestations due to the dense innervation of the region, particularly by the trigeminal nerve (cranial nerve V). Inflammation exacerbates this condition through the release of chemical mediators, such as prostaglandins [28]. Although synthetic drugs remain the mainstay of treatment, prolonged use may result in adverse effects. In this context, the 50 species identified in our review emerge as promising alternatives due to their bioactive metabolites, which exhibit antimicrobial, anti-inflammatory, and analgesic activity, thereby supporting inflammation modulation and tissue regeneration [22].

It is essential to note that the following examples are external to the 50 species identified in this review. For instance, Waizel et al. [46] analyzed 29 species (51 plant uses) and concluded that infusions, decoctions, topical applications, and mouth rinses were the most common forms of administration. Similarly, Lameda Albornoz et al. [23] reported 30 species with potential applications in periodontal treatment, noting that pomegranate was less effective in controlling subgingival plaque and gingivitis compared to other plants. Likewise, Moya Jiménez [49] investigated medicinal plants in Ecuadorian communities and found that chamomile was widely used as an analgesic and anti-inflammatory agent, either alone (56%) or in combination with other remedies (44%).

These external studies complement our findings by highlighting the broader biodiversity of Latin America. However, the focus of our analysis remains exclusively on the 50 species identified in this review, which reinforce the therapeutic potential of regional flora and underscore the importance of sustainable management for their incorporation into dental practice [39].

In addition to therapeutic applications, safety aspects are essential for clinical translation. Table 4 summarizes the most common contraindications and precautions reported for the main species reviewed, including interactions with anticoagulants, allergy risks, and pregnancy-related concerns. Finally, to consolidate the findings of this review, Table 5 presents the 50 Latin American medicinal plants identified with dental applications, detailing their scientific name, family, plant part used, preparation type, indication, evidence type, and corresponding references. This comprehensive synthesis strengthens the relevance of phytotherapeutic diversity in Latin America and highlights its potential contribution to future dental research and practice.

## 16. Conclusions and Future Perspectives

The findings of this review suggest that medicinal plants offer a promising source of bioactive compounds with potential applications in dentistry, particularly due to their antimicrobial, anti-inflammatory, and analgesic properties. However, most of the current evidence derives from in vitro and preclinical studies, while only a few species (e.g., *Aloe vera*, *Propolis*, *Calendula officinalis*) have supportive clinical trials. Thus, clinical translation remains preliminary and requires rigorous validation through well-designed randomized controlled trials.

The development of optimized phytotherapeutic formulations (e.g., gels, mouthwashes, biofilms) could contribute to the management of periodontal and mucosal diseases, provided that issues of dosage, stability, and bioavailability are standardized. Comparative studies against conventional treatments (ibuprofen, paracetamol, and chlorhexidine) and long-term safety evaluations are essential before recommending routine dental use.

To facilitate clarity for readers and avoid overstating clinical potential, we have also included a summary table (Table 6) that highlights representative species, their study types, experimental models, and main findings. This addition allows for a transparent appraisal of the heterogeneity of evidence and helps contextualize the therapeutic promise of these plants within the current limitations of the field.

Finally, attention to regulatory frameworks and sustainable sourcing will be critical to ensure that future phytotherapeutic approaches remain safe, effective, and accessible.

## Figures and Tables

**Figure 1 ijms-26-11502-f001:**
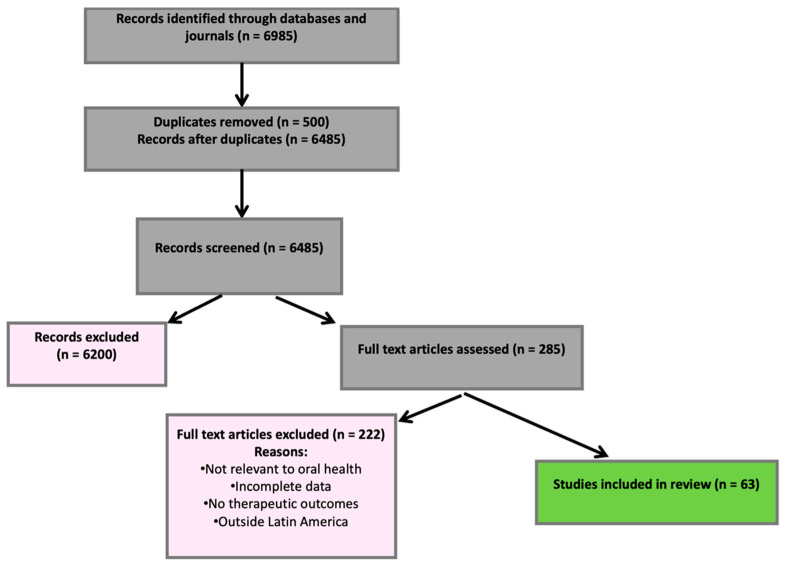
PRISMA flow diagram showing the identification and selection of studies.

**Figure 2 ijms-26-11502-f002:**
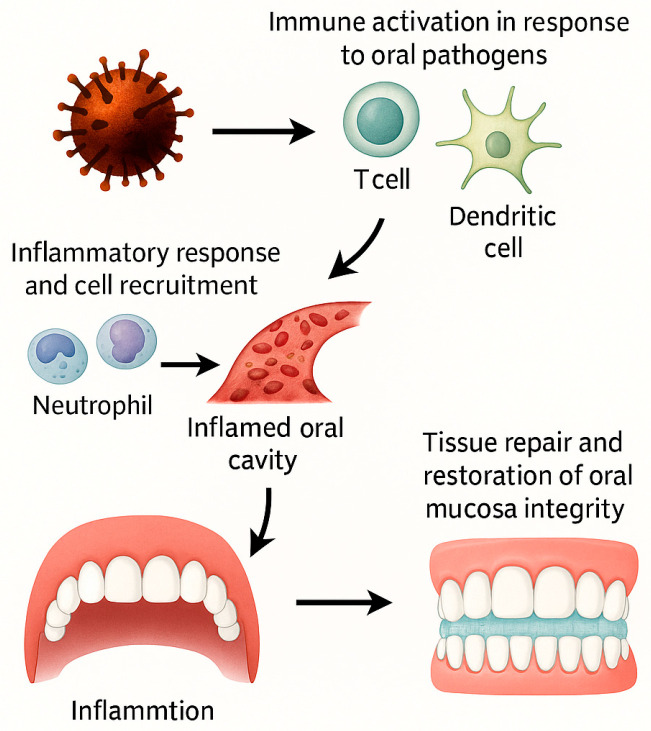
Inflammatory cascade in oral tissues triggered by natural compounds. Oral pathogens activate immune cells (T cells, dendritic cells), leading to the release of inflammatory mediators that recruit neutrophils and other immune cells to the site of injury. This cascade promotes vascular changes and tissue infiltration, resulting in an inflamed oral cavity. Through feedback mechanisms, the inflammatory process transitions into tissue repair and restoration of oral mucosa integrity. This figure illustrates the dual role of inflammation in both oral pathology and healing, highlighting potential targets for plant-derived bioactive compounds.

**Figure 3 ijms-26-11502-f003:**
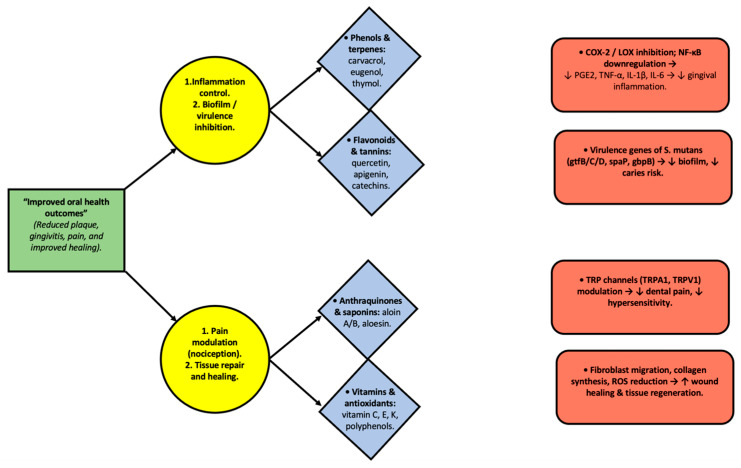
Integrative diagram illustrating the relationship between plant-derived metabolites, their molecular targets, and oral health outcomes. Phenols, flavonoids, anthraquinones, saponins, and vitamins act through specific pathways, including COX-2/LOX inhibition, NF-κB downregulation, modulation of Streptococcus mutans virulence genes, TRP channel regulation, and stimulation of fibroblast activity. These interactions contribute to reducing inflammation, inhibiting biofilm formation, modulating nociception, and enhancing wound healing, ultimately leading to improved oral health outcomes. Arrows indicate the direction of interaction and functional flow between bioactive compounds, molecular mechanisms, and oral effects.

**Table 1 ijms-26-11502-t001:** Primary Oral Diseases and Their Clinical Approach.

	Etiology	Symptoms	Risk Factors	Clinical Management	References
Caries	Bacterial acids cause demineralization of both enamel and dentin.	Thermal sensitivity, progressive pain, pulpitis in advanced cases.	Sugar consumption, poor hygiene, genetic predisposition.	Remineralization, restorations, and root canal treatment in severe cases.	[17,18]
Enamel Fracture	Trauma, bruxism, extensive restorations.	Pain while chewing, thermal sensitivity, and visible fracture in some cases.	Bruxism, repeated trauma, poorly fitted restorations.	Resin or crowns are used for moderate fractures, and root canal treatment is performed if the pulp is affected.	[17]
Dental Crack	Microfractures allow bacterial infiltration.	Discomfort while chewing, progressive sensitivity, and recurrent cavities.	Enamel wear, trauma, acidic diet.	Fissure sealants, preventive restorations.	[19]
Irritation	Acidic or irritating dental materials.	Burning sensation, temporary discomfort in soft tissues.	Improper use of dental materials, pre-existing hypersensitivity.	Use desensitizing agents to adjust adhesive materials.	[19]
Root Exposure	Gingival recession due to periodontitis or aggressive brushing. Bacterial biofilm causes inflammation and destruction of periodontal tissue.	Pain with thermal stimuli, persistent hypersensitivity.	Aggressive brushing, periodontal disease, aging.	Desensitizing toothpaste, changes in brushing technique.	[20]
Periodontal Disease	Bacterial biofilm causes inflammation and destruction of periodontal tissue.	Gingival bleeding, inflammation, tooth mobility, abscesses.	Poor hygiene, smoking, systemic diseases (diabetes).	Biofilm control, scaling, and root planing, periodontal surgery if necessary.	[17,21]

**Table 2 ijms-26-11502-t002:** Medicinal plants: uses, indications, dosage, and bioactive compounds across different regions.

Criteria/Plant	*Zingiber officinale* Roscoe (Zingiberaceae)	*Eucalyptus globulus* Labill. (Myrtaceae)	*Calendula officinalis* L. (Asteraceae)	*Psidium guajava* L. (Myrtaceae)
Uses	Expectorant, antitussive, anti-inflammatory, antioxidant	Antiseptic, anti-inflammatory, antibacterial, analgesic, disinfectant	Anti-inflammatory, healing	Astringent, antimicrobial, immunological
Indications	Stomach pain, nausea, cold, arthritis, xerostomia, gingivitis	Gingivitis, bronchitis, asthma, pharyngitis, diabetes, cystitis, tonsillitis	Gingivitis, stomatitis, ulcers	Gingivitis, ulcers, diarrhea, gastritis, cavities
Dosage	Powder, soups, purees; 250–1000 mg/day (~400 mg/day)	10 mL mouthwash, twice daily for 5 min, for 14 days	Mouth rinse prepared by diluting 2 mL mother tincture in 6 mL water (1:3), twice daily for 6 months	Guava leaf extract (0.15%) used as mouth rinse: 10 mL diluted 1:1 with water, twice daily for 30 days
Compounds	Terpenes derivatives, gingerol, shogaol	Essential oil, tannins, flavonoids, glycosides	Essential oils, salicylic acid, flavonoids	Tannins, flavonoids, vitamin C
Countries	Australia, India, Jamaica, China, Peru, Brazil	Colombia, Venezuela, Argentina, Brazil	Colombia, Mexico, Brazil, Peru	Bolivia, Colombia, Mexico, Brazil, Peru
Safety/ Contraindications	Avoid in patients taking anticoagulants (warfarin, aspirin) → bleeding risk; high doses cause gastric irritation; not recommended in pregnancy (large amounts)	Avoid ingestion of concentrated essential oil (toxic); contraindicated in pregnancy and children <6 years; may cause mucosal irritation	Avoid in patients allergic to Asteraceae family; not recommended in pregnancy/lactation without supervision	Safe in moderate use; avoid in hypersensitivity; high doses may cause constipation or mucosal irritation
Evidence type	Pilot randomized cross-over, single-blind clinical trial.	Randomized controlled clinical trial, in vivo, with 74 human participants (no caries or periodontal disease)	In vivo, randomized controlled clinical trial in humans (n = 240, gingivitis patients).	In vivo, randomized, double-blind, controlled clinical trial.
Primary targets/pathways	Postoperative pain control (Visual Analogue Scale—VAS). Gingival inflammation reduction (Modified Gingival Index—MGI).	Inhibition of bacterial biofilm formation (measured via absorbance and crystal violet staining)	Dental plaque formation, gingival inflammation, and sulcular bleeding (clinical pathways assessed via PI, GI, SBI, OHI-S).	Plaque Index (PI) Gingival Index (GI) Microbial counts (CFU) in plate samples. Salivary antioxidant levels.
Oral results/indications	Ginger powder may be a safe alternative to ibuprofen for managing postoperative pain and gingival inflammation in periodontal surgery	Eucalyptus oil can be used as an effective alternative to chlorhexidine for oral hygiene maintenance, without adverse effects.	The Calendula mouthwash is effective in reducing dental plaque and gingivitis, serving as a useful adjunct to scaling.	Comparable to chlorhexidine and superior to placebo in reducing PI, GI, and microbial counts. Improved salivary antioxidant levels (not statistically significant). Useful adjunct to professional prophylaxis.
References	[40,41,42]	[43,44,45]	[46,47]	[46,47,48]

**Table 3 ijms-26-11502-t003:** Therapeutic properties and applications of selected medicinal plants in oral health.

Criteria/Plant	*Origanum vulgare* L. (Lamiaceae)	*Aloe vera* (L.) Burm.f (Asphodelaceae)	*Ocimum sanctum* L. (Lamiaceae)	*Moringa oleifera* Lam. (Moringaceae)
Uses	Antioxidant, antimicrobial, antifungal	Antioxidant, antimicrobial, anti-inflammatory, astringent, analgesic, healing	Antimicrobial and antiparasitic activity, carminative, antispasmodic, sedative, insecticide	Digestive, anti-inflammatory, antimicrobial, antiparasitic
Indications	Indicated for combating bacterial strains	Gingivitis, periodontitis, gut flora, burns, ulcers, stomatitis	Biofilm control, digestive discomfort, dyspepsia, bloating	Periodontitis, nervous disorders, circulatory system disorders
Dosage	Oil	10 mL, twice daily; 2% gel, three times daily for ≥10 days	Mouthwash with aqueous holy basil extract (3.5 g%), used twice daily for 4 days	5 mL, twice daily for 28 days in young adults with gingivitis
Compounds	Carvacrol, thymol, apigenin, luteolin, aglycones, alcohols	Aloin A and B, aloesin A, B, and C, glucomannans, polysaccharides, tannins, saponins	Eugenol, linalool, estragole, carotenoids, calcium, phosphorus	Minerals: calcium, iron, magnesium, zinc. Vitamins: B1, B2, B3, C, E, K. Includes antioxidants
Countries	Chile, Bolivia, Peru	Mexico, Dominican Republic, Venezuela, China, Brazil, Colombia	Mexico, Colombia, Venezuela, Bolivia, Iran, India, Pakistan	Cuba, Guatemala, Mexico, Colombia, Venezuela, Spain
Safety/ Contraindications	Generally well tolerated; high doses may cause abdominal discomfort, nausea, constipation/diarrhea, dizziness, headache. Rare hypersensitivity reactions. Contraindicated in pregnancy (abortifacient risk).	Avoid in pregnancy (risk of uterine contractions/abortifacient). Possible GI discomfort (diarrhea, constipation, nausea), headache, dizziness, rare hypersensitivity.	May significantly prolong prothrombin time; caution in patients on anticoagulant therapy.	A clinical case of pulmonary embolism reported after prolonged use of Moringa leaf extract; caution in patients on anticoagulants/antiplatelets or with bleeding disorders.
Evidence type	In vitro study (disk diffusion, MIC, MBC, biofilm assays, RT-qPCR, molecular docking, cytotoxicity test on HaCaT cells).	In vivo, randomized controlled clinical trial (30 orthodontic patients, Aloe vera vs. 0.2% chlorhexidine, 35 days).	In vivo, triple-blind randomized controlled clinical trial	In vivo, randomized clinical crossover study (20 subjects with mild to moderate gingivitis).
Primary targets/ pathways	Virulence genes of *S. mutans* (gtfB/C/D, spaP, gbpB, vicR, relA, brpA); inhibition of acid production, hydrophobicity, and biofilm formation. Key compounds: carvacrol, γ-terpinene, p-cymene.	↓ IL-1β, IL-17 (GCF), ↑ fibroblast migration, ↓ *P. gingivalis* biofilm	COX-2 and LOX enzyme inhibition (anti-inflammatory); NF-κB pathway and cytokine (TNF-α, IL-6) suppression, esp. via eugenol (in THP-1 cells)	Inhibition of *Streptococcus mutans* growth and cariogenic biofilm; phenolic compounds in ethanol extract likely interfere with bacterial adhesion and acid production.
Oral results/indications	potential anti-caries agent due to anti-biofilm and anti-virulence activity.	Significant reduction in PI, GI, and BOP from baseline to day 35 in both groups. Aloe vera mouthwash showed effects comparable to chlorhexidine, with no adverse effects reported.	Comparable to chlorhexidine in reducing PI, GI, and gingival bleeding; significantly better than placebo; no adverse effects reported.	Significant reduction in PI and GI compared to baseline and miswak dentifrice. Demonstrated efficacy in reducing plaque accumulation and gingival inflammation. Safe and effective as an adjunctive oral hygiene aid.
References	[49,50,51,52]	[49,53,54,55,56]	[49,57,58,59,60]	[15,49,61,62,63]

**Table 4 ijms-26-11502-t004:** Safety considerations for top medicinal plants in dentistry.

Plant (Family)	Interactions (e.g., Anticoagulants)	Allergy Risks	Pregnancy/Lactation	Mucosal Irritation	Dose/Form/Duration	Evidence Type	References
*Zingiber officinale* Roscoe (Zingiberaceae)	Avoid with anticoagulants (warfarin, aspirin); bleeding risk at high doses	Rare hypersensitivity	Not recommended in pregnancy (large amounts)	May cause gastric irritation	Up to 1000 mg/day (powdered form)	Clinical and experimental	[40,41,42]
*Eucalyptus globulus* Labill. (Myrtaceae)	None reported; avoid ingestion of essential oil (toxic)	Possible allergic reaction in sensitive individuals	Contraindicated in pregnancy and children <6 years	May cause mucosal irritation	Used as infusion or diluted essential oil	Clinical reports	[43,44,45]
*Calendula officinalis* L. (Asteraceae)	None reported	Contraindicated in Asteraceae allergy	Not recommended during pregnancy or lactation without supervision	Mild irritation in sensitive individuals	Topical or oral rinse	Clinical and ethnobotanical	[46,47]
*Psidium guajava* L. (Myrtaceae)	None reported	Avoid in hypersensitivity	Generally safe in moderate use	Possible mild irritation at high doses	Oral gel, infusion, topical	Clinical and ethnobotanical	[48,49]
*Origanum vulgare* L. (Lamiaceae)	Possible interaction with anticoagulants at high doses	Rare hypersensitivity	Contraindicated in pregnancy (abortifacient risk)	GI discomfort (nausea, diarrhea, dizziness)	Oil or fluid extract	Clinical and ethnobotanical	[49,50,51,52]
*Aloe vera* (L.) Burm.f. (Asphodelaceae)	None reported	Rare hypersensitivity	Avoid in pregnancy (risk of uterine contractions)	Possible GI discomfort (diarrhea, nausea, dizziness)	Topical gel or mouth rinse	Clinical trials	[49,53,54,55,56]
*Ocimum sanctum* L. (Lamiaceae) CE	May prolong prothrombin time; caution with anticoagulant therapy	None reported	Not reported	None reported	Mouthwash, twice daily for 4 days	Clinical	[49,57,58,59,60]
*Moringa oleifera* Lam. (Moringaceae)	Caution in patients on anticoagulants/antiplatelets	None reported	Use with caution in pregnancy	None reported	5 mL leaf extract twice daily for 28 days	Clinical and ethnobotanical	[15,49,61,62,63]

**Table 5 ijms-26-11502-t005:** List of 50 Latin American medicinal plants with dental applications, including scientific name, family, plant part, preparation, indication, evidence type, and references.

#	Common Name	Scientific Name	Traditional/Dental Uses	Relevant Pharmacological Activity	Precautions/Contraindications	Administration/Form	Countries Reported in Latin America	Main References
1	Anamú	*Petiveria alliacea* L. (Petiveriaceae)	Dental pain, oral inflammation.	Anti-inflammatory, analgesic, antimicrobial	Pregnancy, lactation, prolonged use	Infusion, poultice	Colombia, Venezuela, Brazil	[64]
2	Cadillo	*Bidens pilosa* L. (Asteraceae)	Mouth ulcers, gastric ulcers.	Anti-inflammatory, healing	Pregnancy, lactation, kidney failure	Infusion, syrup, capsules	Peru, Colombia, Central America	[65]
3	Mountain cinnamon	*Drimys winteri* J.R.Forst. & G.Forst. (Winteraceae)	Allergic processes, stomach pain; reduces bacterial plaque index.	Analgesic, anti-inflammatory	Not reported	Infusion (oral)	Chile, Argentina	[66]
4	Chaparro	*Curatella americana* L. (Dilleniaceae)	Effective against *Candida albicans*, *C. tropicalis*, and *C. parapsilosis*, all highly relevant in oral candidiasis and denture stomatitis.	Anti-inflammatory, astringent	Pregnancy/reproductive effects: may alter estrous cycle and embryonic development; caution in pregnancy	Infusion (oral)	Brazil (Cerrado), Venezuela, Colombia	[67]
5	Chilca	*Baccharis latifolia* (Ruiz & Pav.) Pers. (Asteraceae)	Stomach pain, oral inflammations.	Anti-inflammatory, antioxidant	Safety not established; lacks pharmacological/toxicological validation	Infusion (oral, topical)	Peru, Bolivia, Colombia	[68]
6	Cilantro	*Coriandrum sativum*	Efficacy against *Candida* spp. isolated from the oral cavity of patients with periodontitis, including *Candida albicans*.	Anti-inflammatory, antimicrobial	Pregnancy, lactation, children <3 years	Infusion (oral)	Brazil, Colombia, Central America	[69]
7	Guava	*Psidium guajava* L. (Myrtaceae)	Comparable to chlorhexidine and superior to placebo in reducing PI, GI, and microbial counts. Improved salivary antioxidant levels (not statistically significant). Useful adjunct to professional prophylaxis.	Local anti-inflammatory	Hypersensitivity to the plant	Topical gel	Colombia (endemic)	[70]
8	Gualanday	*Jacaranda caucana* Pittier (Bignoniaceae)	Leishmaniasis, boils, skin infections, oral inflammation.	Antiseptic, healing	External use only	Topical gel	Colombia (endemic)	[71]
9	Plantain	*Plantago major* L. (Plantaginaceae)	Aqueous mouthwash trial in a patient reduced gingival inflammation and dental biofilm without side effects.	Healing, anti-inflammatory	External use only	Topical (poultice, ointment)	Colombia, Ecuador, Bolivia	[71]
10	Chamomile (EC)	*Matricaria chamomilla* L. (Asteraceae) (syn. *Matricaria recutita* L.).	Gingivitis, dental pain, oral inflammation, reduced plaque.	Anti-inflammatory, antispasmodic	Hypersensitivity	Infusion, oral extract	Colombia, Bolivia, Peru	[72]
11	Cashew	*Minthostachys mollis* (Benth.) Griseb. (Lamiaceae)	Essential oil effectively inhibited *Candida albicans* in a dental-relevant model.	Anti-inflammatory, hypoglycemic	Contact dermatitis, CNS toxicity in excess	Bark decoction (oral)	Colombia, Brazil, Venezuela	[71,73]
12	Muña	*Minthostachys mollis*	Essential oil active against *E. faecalis*, *P. gingivalis*, and *C. albicans* (most sensitive); key compounds: menthone and eucalyptol.	Antimicrobial against *Streptococcus mutans*	Prolonged or high-dose use linked to liver and lung toxicity in animal studies; caution advised due to possible hepatotoxic effects	Infusion, tincture	Ecuador, Peru, Bolivia	[71,74]
13	Oregano (EC)	*Origanum vulgare* L. (Lamiaceae)	Potential anti-caries agent due to anti-biofilm and anti-virulence activity.	Antioxidant, antimicrobial, antifungal	Generally well tolerated; high doses may cause abdominal discomfort, nausea, constipation/diarrhea, dizziness, headache Rare hypersensitivity reactions Contraindicated in pregnancy (abortifacient risk)	Oil, fluid extract	Chile, Bolivia, Peru, Colombia	[75]
14	Arnica (EC)	*Arnica montana* L. (Asteraceae)	Arnica reduced post-extraction pain almost as effectively as ibuprofen, with good tolerance and no adverse effects.	Topical anti-inflammatory, analgesic	Risk of gastrointestinal irritation and for safety reasons typical of homeopathic preparations	Gel, cream, topical tincture	Bolivia, Mexico, Colombia	[71]
15	Calendula (EC)	*Calendula officinalis* L. (Asteraceae)	The Calendula mouthwash is effective in reducing dental plaque and gingivitis, serving as a useful adjunct to scaling.	Anti-inflammatory, healing, antimicrobial	Hypersensitivity	Infusion, mouth rinse, cream	Bolivia, Colombia, Chile, Peru	[47,76]
16	Aloe	*Aloe vera* (L.) Burm.f. (Asphodelaceae)	Significant reduction in PI, GI, and BOP from baseline to day 35 in both groups. Aloe vera mouthwash showed effects comparable to chlorhexidine, with no adverse effects reported.	Anti-inflammatory, healing, antimicrobial	Avoid in pregnancy (risk of uterine contractions/abortifacient). Possible GI discomfort (diarrhea, constipation, nausea), headache, dizziness, rare hypersensitivity	Topical gel, mouth rinse	Colombia, Bolivia, Mexico, Venezuela	[49,53,54,55,56]
17	Onion (EC)	*Allium cepa* L. (Amaryllidaceae)	Bactericidal activity even in resting cells and remained stable after 48 h; promising for preventing dental caries and periodontitis.	Antimicrobial, healing, anti-inflammatory	Gastric irritation if in excess	Extract, poultice, infusion	Colombia, Bolivia, Mexico, Peru	[71,77]
18	Basil (EC)	*Ocimum sanctum* L. (Lamiaceae))	Comparable to chlorhexidine in reducing PI, GI, and gingival bleeding; significantly better than placebo; no adverse effects reported.	Antimicrobial and antiparasitic activity, carminative, antispasmodic, sedative, insecticide	May significantly prolong prothrombin time; caution in patients on anticoagulant therapy	Mouthwash with aqueous holy basil extract (3.5 g%), used twice daily for 4 days	Mexico, Colombia, Venezuela, Bolivia, Iran, India, Pakistan	[78]
19	Tree tomato	*Solanum lycopersicum* L. (Solanaceae).	Activity by inhibiting *Streptococcus mutans* (large inhibition zone) and *Porphyromonas gingivalis* (moderate effect).	Antioxidant, antihypertensive	Not reported	Infusion, juice	Colombia, Ecuador, Bolivia	[71,79]
20	Lemon balm (EC)	*Melissa officinalis* L. (Lamiaceae)	Lemon balm oil hydrogel reduced *Candida albicans* adhesion on denture surfaces, showing antifungal potential.	Sedative, oral antimicrobial, anti-inflammatory	Not reported	Infusion, extract, essence	Colombia, Ecuador, Bolivia	[71,80]
21	Lavender (EC)	*Lavandula angustifolia* Mill. (Lamiaceae)	Lavender oil reduced sulfur compounds from *F. nucleatum*, showing anti-halitosis potential comparable to chlorhexidine.	Antiseptic, anti-inflammatory, anxiolytic	Animal studies found no acute toxicity or irritation at high doses; human long-term safety still unclear	Infusion, tincture, oil	Chile, Colombia, Bolivia	[71]
22	Papaya	*Carica papaya* L.	Stomatitis, post-extraction healing, CPLE toothpaste (± mouthwash) effectively reduced interdental bleeding compared to control.	Anti-inflammatory, healing, proteolytic	Latex may irritate	Juice, latex, seeds	Colombia, Bolivia, Mexico, Peru	[71,79]
23	*Moringa oleifera* (EC)	*Moringa oleifera* Lam. (Moringaceae)	Periodontitis, nervous disorders, circulatory system disorders.	Digestive, anti-inflammatory, antimicrobial, antiparasitic	A clinical case of pulmonary embolism reported after prolonged use of Moringa leaf extract; caution in patients on anticoagulants/antiplatelets or with bleeding disorders	5 mL, twice daily for 28 days in young adults with gingivitis	Cuba, Guatemala, Mexico, Colombia, Venezuela, Spain	[81]
24	Chulena	*Calceolaria thyrsiflora* Graham	Gum inflammation, tongue ulcers.	Oral anti-inflammatory, vulnerary	Not reported	Infusion, rinse	Chile, possibly Bolivia, Colombia	[71]
25	Vervain	*Verbena litoralis* Kunth	GI problems, diarrhea, sedative use.	Anti-inflammatory, sedative	CNS depressants, not with alcohol	Extract, oral infusion	Bolivia, Peru, Colombia	[71]
26	Spearmint (EC)	*Mentha spicata* L.	Stomach pain, halitosis, oral hygiene.	Antiseptic, digestive, oral antimicrobial	Hypersensitivity, nausea if excessive	Infusion, mouth rinse	Colombia, Bolivia, Peru	[71]
27	*Zingiber officinale* Roscoe (Zingiberaceae) (EC)	*Zingiber officinale* Roscoe (Zingiberaceae)	Ginger powder may be a safe alternative to ibuprofen for managing postoperative pain and gingival inflammation in periodontal surgery.	Expectorant, antitussive, anti-inflammatory, antioxidant	Avoid in patients taking anticoagulants (warfarin, aspirin) → bleeding risk; high doses cause gastric irritation; not recommended in pregnancy (large amounts)	Powder, soups, purees; 250–1000 mg/day (~400 mg/day [46])	Australia, India, Jamaica, China, Peru, Brazil	[40,41,42]
28	Proteaceae	*Oreocallis grandiflora* (Lam.) R. Br.	Treatment of ulcers, mouth rinses for pain, cleansing.	Anti-inflammatory, antimicrobial, healing, antioxidant	toxicity, allergic reactions, limited use	Infusion, decoction, gel, rinse, topical form	Peru, Colombia, Mexico, Brazil	[71]
29	Amazonian clematis	*Clematis guadeloupae* Pers.	Preserve teeth, gum relief.	Presumed anti-inflammatory/astringent	Avoid on sensitive mucosa	Bark/stem infusion, poultice	Peru, Bolivia, Colombia, Ecuador	[71]
30	Lareta	*Azorella ruizii* (sin. *Laretia acaulis*)	Topical resin for toothache; decoction digestive.	Topical anti-inflammatory, mild analgesic	Not reported orally	Resin, decoction	Chile, Argentina, Bolivia, Peru	[71]
31	Mallow (EC)	*Malva sylvestris* L.	Ulcers, mucositis, dry mouth. In vitro anti-inflammatory co-culture model with periodontal pathogen (*Aggregatibacter actinomycetemcomitans*).	Oral anti-inflammatory, antimicrobial, anti-caries, healing	No enamel staining observed with hydroalcoholic extract application, suggesting it maintains aesthetic safety	Infusion, rinse, poultice	Latin America (Mexico to Venezuela)	[82]
32	Maqui	*Aristotelia chilensis* (Molina) Stuntz	Potential antifungal for oral candidiasis (reduces hyphae and boosts nystatin).	Anti-inflammatory, antimicrobial (anti-*Candida*), antioxidant	Caution on sensitive mucosa	Infusion, topical extracts	Chile, Argentina, Bolivia, Colombia, Peru	[71]
33	Walnut (EC)	*Juglans regia* L. (Juglandaceae)	Decoctions for canker sores, tonsillitis.	Antimicrobial, antifungal, anti-caries, whitening	Juglone may irritate/stain	Decoction rinse	Latin America	[71]
34	Thyme (EC)	*Thymus vulgaris* L. (Lamiaceae)	Potential adjunct irrigant in root canal therapy, helping lower NaOCl use and reduce cytotoxicity.	Antimicrobial (*S. mutans*), anti-inflammatory, antioxidant	May irritate mucosa; avoid in pregnancy/lactation	Infusion, tincture, essential oil, rinse	Chile, Bolivia, Colombia, Peru, Mexico	[71]
35	Rosemary (EC)	*Salvia rosmarinus* Spenn. (*Rosmarinus officinalis* L.)	Rosemary extract dentifrice reduced early Streptococcus mutans biofilm formation, suggesting cariostatic potential.	Antimicrobial (*S. mutans*, *P. gingivalis*), anti-inflammatory, antioxidant	Gastric irritation; contraindicated in epilepsy, pregnancy	Infusion, tincture, diluted essential oil	Chile, Bolivia, Colombia, Peru, Mexico, Argentina	[71,83]
36	Chilca (int.)	*Baccharis latifolia* (Ruiz & Pav.) Pers. (Asteraceae)	Oral inflammation, ulcers inhibition of *S. aureus* and *E. faecalis* (both relevant in endodontic and oral infections) suggests potential applicability in controlling oral Gram-positive pathogens.	Anti-inflammatory, healing, antioxidant, antimicrobial	Avoid in pregnancy/lactation; high doses irritate	Not reported	Bolivia, Colombia, Peru, Ecuador, Chile	[84]
37	Leliantu	*Geum chilense* Balb. ex Lindl. (Rosaceae)	Toothache, inflamed gums. The modulatory effect of the extracts on neutrophil function, attributed mainly to gemin A, supports the traditional use of this plant material for oral inflammation, including mucositis, gingivitis, and periodontitis.	Anti-inflammatory, antioxidant, astringent	May affect cyclosporine metabolism	Infusion, decoction, topical use	Chile, Bolivia, Argentina	[71]
38	Cranberry (EC)	*Vaccinium macrocarpon* Aiton (Ericaceae)	Antimicrobial and antioxidant. Cranberry mouthwash reduced *S. mutans* by 68%, showing efficacy comparable to chlorhexidine and supporting its use as an alternative oral rinse.	Anti-biofilm, immunomodulatory	Acidic pH may erode enamel; warfarin interaction	Juice, extract, rinse	Ecuador, Colombia, Bolivia, Peru	[85]
39	Clove (EC)	*Syzygium aromaticum* (L.) Merr. & L.M. Perry (Myrtaceae)	Analgesic, antiseptic, *Streptococcus mutans*; in vitro antimicrobial and antibiofilm assays.	Eugenol: anesthetic, anti-inflammatory, antibacterial, antifungal	Toxic in excess; not in children/pregnancy	Oil, tincture, rinse, paste	Ecuador, Colombia, Peru, Bolivia, Central America	[86,87]
40	Echinacea (EC)	*Echinacea purpurea* (L.) Moench (Asteraceae)	Oral rinses for gingivitis, canker sores. Inhibited Streptococcus mutans growth and reduced biofilm formation. Showed antibacterial and antifungal activity against cariogenic bacteria and *Candida albicans*. Demonstrated low cytotoxicity, suggesting potential as a safe herbal mouthwash.	Anti-inflammatory, immunomodulatory	Hepatotoxic if >8 weeks	Infusion, tincture, capsules, rinse	Ecuador, Colombia, Bolivia, Peru, Central America	[88,89]
41	Eucalyptus (EC)	*Eucalyptus globulus* Labill. (Myrtaceae)	Eucalyptus oil can be used as an effective alternative to chlorhexidine for oral hygiene maintenance, without adverse effects.	Anti-inflammatory, antioxidant, antimicrobial	Not for children; mucosa irritation	Infusion, essential oil, rinse	Andes (Ecuador–Chile), Colombia, Peru, Mexico, Brazil	[43,44,45]
42	Devil’s claw (EC)	*Harpagophytum procumbens* (Burch.) DC. ex Meisn. (Pedaliaceae)	Periodontitis, oral inflammation. Its strong anti-inflammatory and analgesic properties—effective in arthritis and neuropathic pain models—suggest potential usefulness in managing oral inflammatory conditions such as mucositis or post-procedural pain.	Anti-inflammatory, antioxidant	Contraindicated in pregnancy/lactation	Infusion, capsules, gels	Ecuador, Bolivia, Colombia, Peru, Brazil, Argentina	[71]
43	Apiaceae (Hydrocotyle)	*Hydrocotyle bonariensis* Comm. ex Lam. (Araliaceae)	Shows anti-inflammatory and analgesic effects; may help manage oral inflammation, mucositis, or post-procedural pain.	Anti-inflammatory, healing, antioxidant	Not reported	Infusion, decoction, extracts	Peru, Bolivia, Colombia, Ecuador, Brazil, Argentina	[90]
44	Acmella	*Spilanthes acmella* (*S. acmella*),	Toothache, caries, anesthetic.	Analgesic, antimicrobial, healing	Not reported	Infusion, fresh extract	Peru, Bolivia, Colombia, Ecuador, Brazil, Venezuela	[71,91]
45	Ullucu	*Ullucus tuberosus* Caldas (Basellaceae)	Toothache, inflammation.	Anti-inflammatory, antimicrobial	No major toxicity; caution with oxalates	Cataplasms, decoctions	Andes (Peru, Bolivia, Colombia, Ecuador, Chile, Argentina)	[71]
46	Coca	*Erythroxylum coca* Lam. (Erythroxylaceae)	Toothache, gingival pain. Traditionally used as a natural anesthetic for toothache and oral pain relief—via chewing or poultice application.	Topical anesthetic, analgesic, antimicrobial	Chewing coca leaves may cause oral epithelial changes and is linked to risk of oral squamous cell carcinoma (OSCC), even without classic risk factors	Leaf chewing, infusions, rinses	Andes (Peru, Bolivia, Colombia, Ecuador, Argentina)	[92]
47	Lima bean	*Phaseolus lunatus* L. (Fabaceae)	Toothache.	Anti-inflammatory, analgesic, antimicrobial	Unprocessed seeds have cyanogenic compounds and lectins that can release toxic hydrogen cyanide, making proper preparation essential for safe use	Cooking, poultices	Peru, Bolivia, Colombia, Ecuador, Chile, Mexico	[93]
48	Andean gentian	*Gentianella rima* (G.Don) Fabris (Gentianaceae)	Toothache, gum inflammation.	Bitter tonic, anti-inflammatory	Not in pregnancy/gastritis	Infusion, decoction	Andes (Peru, Bolivia, Colombia, Ecuador)	[71]
49	Black mulberry	*Morus nigra* L. (Moraceae)	Gingivitis, stomatitis, sore throat. Black mulberry juice mouthwash reduced plaque and gingival inflammation, showing comparable effectiveness to chlorhexidine in managing gingivitis.	Anti-inflammatory, antioxidant	Hypoglycemia risk	Infusion, syrup, gargles	Peru, Bolivia, Colombia, Ecuador, Brazil, Mexico, Chile, Paraguay	[94,95]
50	White myrtle	*Luma chequen* (Molina) A.Gray (Myrtaceae)	Effective in reducing ulcer severity and improving oral health outcomes in RAS patients, with good safety profile.	Aromatic oils, antimicrobial	Not reported	Infusion, rinse, aromatherapy	Andes (Peru, Chile, Argentina, Bolivia, Ecuador)	[71]

Note: EC = External Comparator. The “EC” label is used exclusively to identify species whose botanical origin is not Latin American (e.g., European, Asian, North American, or African plants such as *Matricaria chamomilla*, *Origanum vulgare*, *Arnica montana*, *Calendula officinalis*, *Allium cepa*, *Ocimum sanctum*, *Melissa officinalis*, *Lavandula angustifolia*, *Moringa oleifera*, *Mentha spicata*, *Malva sylvestris*, *Juglans regia*, *Thymus vulgaris*, *Rosmarinus officinalis*, *Vaccinium macrocarpon*, *Syzygium aromaticum*, *Echinacea purpurea*, and *Harpagophytum procumbens*). However, it is essential to emphasize that the scientific literature cited for each of these species clearly documents their cultivation, availability, or therapeutic applications in Latin American countries—particularly in Colombia, Mexico, Peru, Bolivia, Chile, Ecuador, and Brazil. Therefore, even though some species are of exogenous origin, their presence, agricultural adaptation, and ethnopharmacological use in the region are well supported by current scientific evidence, making their inclusion in the table fully valid and contextually appropriate within the Latin American scope of this study.

**Table 6 ijms-26-11502-t006:** Safety considerations and contraindications of selected medicinal plants relevant to dentistry.

Plant (Family)	Interactions (e.g., Anticoagulants)	Allergy Risks	Pregnancy/ Lactation	Mucosal Irritation	Dose/Form/ Duration	Evidence Type/ Main Findings	References
*Propolis* (Apidae product)	Possible interaction with anticoagulants (high doses)	Rare allergic reactions in sensitive individuals (bee-related)	Use with caution during pregnancy and lactation	None reported	Topical gel and oral solution; combined with SRP in Wistar rats (in vivo model)	In vivo study: combined with SRP, significantly reduced IL-1β, TNF-α, and MDA; propolis alone showed no relevant effects.	[96]
*Aloe vera* (L.) Burm.f. (Asphodelaceae)	None reported	Rare hypersensitivity	Safe in animal models; avoid excessive oral use during pregnancy	None reported	Aloe vera gel + β-TCP for 8 weeks in dogs with furcation defects	In vivo study: enhanced periodontal regeneration (bone, PDL fibers, vascularization) with no adverse effects.	[49,53,54,55,56]
Basil (*Ocimum basilicum* L.) (Lamiaceae)	None reported	Possible mild allergy to Lamiaceae family	Not reported	None reported	Hydroalcoholic extract (1–10%) in agar diffusion assays (*S. mutans*, *E. faecalis*, *S. sanguinis*)	In vitro study: demonstrated antibacterial activity with both individual and synergistic effects, although less potent than chlorhexidine.	[57]

## Data Availability

No new data were created or analyzed in this study. Data sharing is not applicable to this article.

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
