# Peer review of "Therapeutic Potential of Latin American Medicinal Plants in Oral Diseases: From Dental Pain to Periodontal Inflammation—A Systematic Review"

_ijms, 2025, doi:10.3390/ijms262311502_

Round 1
Reviewer 1 Report
Comments and Suggestions for Authors
The manuscript entitled "Unveiling the Potential of Latin American Medicinal Plants as a Natural Alternative for Treating Dental Pain" seems to give a comprehensive review of Latin American medicinal plants for dental health control. It highlights the underexplored phytotherapeutic potential of Latin American medicinal plants, a valuable contribution to dental therapy. However, there are notable weaknesses in scientific analysis, mechanism of action, and methodology. Moreover, the specific results on anti-inflammatory and antimicrobial activity and mechanism focused on dental health are lacking.
Especially,
1) The suggested molecular mechanisms, such as cytokine inhibition, COX/PGE2 are not critical, and lacks specific pathway network with chemical ingredients of the plants. The specific pharmacokinetic and pharmacodynamic data for the plant compounds to act in dental healthcare should be addressed.
2) The conclusion overstates clinical potential without significant clinical trials.
3) In the Graphical Abstract, spelling check (Alxaloids) and proper labeling for the chemical structures are needed.
4) Several references lack DOIs and suitable citation are needed.
Author Response
Letter with the details of the changes
We would like to thank the reviewers for their valuable suggestions and observations sincerely. We have carefully considered their feedback and accepted all recommendations. The changes made to the manuscript are clearly marked in blue. Below, we provide a summary of each reviewer's specific comments and the corresponding revisions made in response.
Reviewer Comments:
Reviewer 1
1) The suggested molecular mechanisms, such as cytokine inhibition, COX/PGE2 are not critical, and lacks specific pathway network with chemical ingredients of the plants. The specific pharmacokinetic and pharmacodynamic data for the plant compounds to act in dental healthcare should be addressed.
Answer
We thank the reviewer for this observation. In Section 8, we have provided greater precision regarding molecular mechanisms. Specifically, in Figure 3 (lines 444–465) we further detail the pathways involved, and we added a dedicated subsection entitled “Mechanisms of Action of Select Plant Metabolites in Oral Inflammation” (lines 483–535). This subsection highlights the most relevant species from our review and discusses representative bioactive metabolites, together with their proposed molecular targets, all supported with appropriate references.
Line 444- 465:
To provide a mechanistic integration of plant bioactives, Figure 3 presents a flowchart mapping plant metabolites to molecular targets and oral outcomes. This schematic highlights how compounds such as phenols, flavonoids, saponins, and vitamins modulate pathways (e.g., COX/LOX inhibition, NF-κB downregulation, TRP modulation), ultimately leading to improved oral outcomes like reduced plaque, gingivitis, pain, and enhanced healing.
Figure 3. Integrative diagram illustrating the relationship between plant-derived metabolites, their molecular targets, and oral health outcomes. Phenols, flavonoids, anthraquinones, saponins, and vitamins act through specific pathways such as COX-2/LOX inhibition, NF-κB downregulation, modulation of Streptococcus mutans virulence genes, TRP channel regulation, and stimulation of fibroblast activity. These interactions contribute to reducing inflammation, inhibiting biofilm formation, modulating nociception, and enhancing wound healing, ultimately leading to improved oral health outcomes.”
Lines 483–535: “8.1 Mechanisms of action of selected plant metabolites in oral inflammation
Salvia rosmarinus Spenn. (Lamiaceae)Rosmarinic acid, the predominant bioactive molecule in Rosmarinus officinalis, has been associated with anti-inflammatory activity in oral contexts. Studies on human gingival fibroblasts challenged with lipopolysaccharides revealed that it lowers oxidative stress, preserves intracellular glutathione, and suppresses the expression of pro-inflammatory mediators such as IL-1β, IL-6, TNF-α, and iNOS, an effect linked to the inhibition of NF-κB signaling during periodontal inflammation [37]. Additional evidence in vascular smooth muscle cells showed that rosmarinic acid downregulates MAPK (ERK, JNK, p38) and NF-κB pathways, leading to a reduced release of inflammatory mediators, including iNOS, TNF-α, and IL-8 under LPS stimulation. Although these observations were made in non-oral systems, they support the molecular plausibility of this compound in modulating signaling cascades relevant to periodontal disease [38].
Moringa oleifera Lam. (Moringaceae) is a rich source of phytochemicals such as quercetin and isothiocyanates, which act on inflammatory pathways important in oral health. In a rat model of periodontal inflammation, ethanolic leaf extracts enriched with quercetin attenuated tissue damage by suppressing NF-κB activation, thereby reducing pro-inflammatory cytokines and supporting periodontal regeneration [39]. Moreover, moringa isothiocyanate-1 (MIC-1) has been shown to influence both Nrf2 and NF-κB signaling under LPS-induced inflammation, lowering IL-6, IL-1β, and TNF-α expression while alleviating oxidative stress. Beyond preclinical data, clinical findings provide additional support: in a randomized controlled trial including 36 gingivitis patients, a Moringa oleifera Lam. (Moringaceae) mouthwash significantly decreased plaque and gingival indices after 14 days of use, showing efficacy comparable to chlorhexidine and superior to saline [40]. Altogether, this evidence underscores a strong mechanistic and clinical rationale for Moringa oleifera as a natural adjunct in periodontal therapy.
Aloe vera (L.) Burm.f. (Asphodelaceae) Acemannan, the main polysaccharide of Aloe vera, exhibits immunomodulatory properties within oral environments. In human gingival fibroblasts, acemannan directly binds to Toll-like receptor 5 (TLR5), triggering NF-κB signaling and promoting the release of IL-6 and IL-8. This process not only enhances tissue repair but also strengthens host immune defenses in periodontal tissues, providing dual benefits in regeneration and antimicrobial protection [41]. More broadly, Aloe vera (L.) Burm.f. (Asphodelaceae) and its major active metabolites, including acemannan and aloin, have been recognized for their anti-inflammatory, antioxidant, and wound-healing capacities, reinforcing their potential applications in oral medicine [42].
Ocimum basilicum L Linalool, a monoterpene present in Ocimum basilicum, has been documented to exert immunomodulatory effects directly relevant to periodontal inflammation. In vitro studies showed that linalool decreases the overproduction of pro-inflammatory cytokines (IL-6, IL-10, IL-17, IFN-γ) in human immune cells exposed to Porphyromonas gingivalis, a keystone pathogen in periodontitis. These outcomes are linked to the downregulation of NF-κB and MAPK signaling, pathways that control inflammatory gene transcription. By moderating excessive immune responses without impairing cell viability, linalool may help safeguard periodontal tissues from chronic inflammatory injury [43].
Propolis — resinous product of Apis mellifera L. (Apidae) Current research emphasizes the role of CAPE (caffeic acid phenethyl ester), the principal phenolic constituent of propolis, in regulating inflammatory activity in periodontal cells. In human gingival fibroblasts stimulated with LPS and IFN-α, CAPE markedly suppressed the production of TNF-α, IL-6, and IL-8, with no cytotoxicity observed. In contrast, ethanolic propolis extract (EEP) produced weaker and less consistent effects on cytokine release. These findings suggest that CAPE exerts a more selective and potent anti-inflammatory influence, supporting its potential use in controlling periodontal inflammation [44].”
2) The conclusion overstates clinical potential without significant clinical trials.
Answer:
We appreciate this valuable comment. In response, we have moderated the conclusions to avoid overstating the clinical potential and to emphasize the need for further validation through rigorous clinical trials. Additionally, to strengthen the practical relevance of our findings, we added a new summary table (Table 5) that highlights safety considerations and contraindications for selected medicinal plants most frequently studied in dentistry. This addition addresses reviewer suggestions and aims to provide readers with a balanced view of both therapeutic potential and safety concerns, ensuring that clinical translation is considered with appropriate caution.
Line 683-712:
The findings of this review indicate that medicinal plants represent a promising source of bioactive compounds with potential applications in dentistry, particularly due to their antimicrobial, anti-inflammatory, and analgesic effects. However, most of the current evidence derives from in vitro and preclinical studies, while only a few species (e.g., Aloe vera, Propolis, Calendula officinalis) have supportive clinical trials. Thus, clinical translation remains preliminary and requires rigorous validation through well-designed randomized controlled trials.
The development of optimized phytotherapeutic formulations (e.g., gels, mouthwashes, biofilms) could contribute to the management of periodontal and mucosal diseases, provided that issues of dosage, stability, and bioavailability are standardized. Comparative studies against conventional treatments (ibuprofen, paracetamol, chlorhexidine) and long-term safety evaluations are essential before routine dental use can be recommended.
To facilitate clarity for readers and to avoid overstating clinical potential, we also included a summary table (Table 5) highlighting representative species, their study type, experimental models, and main findings. This addition allows a transparent appraisal of the heterogeneity of evidence and helps contextualize the therapeutic promise of these plants within the current limitations of the field.
Finally, attention to regulatory frameworks and sustainable sourcing will be critical to ensure that future phytotherapeutic approaches remain safe, effective, and accessible.
Table 5. Safety considerations and contraindications of selected medicinal plants relevant to dentistry
|
Plant |
Study type |
Model / Method |
Main findings |
Reference |
|
Propolis |
In vivo |
Wistar rats with ligature-induced periodontitis |
In combination with SRP, significantly reduced IL-1β, TNF-α, and MDA; propolis alone showed no relevant effects. |
[59] |
|
Aloe vera |
In vivo |
Dogs with surgically induced furcation defects |
Aloe vera gel + β-TCP enhanced periodontal regeneration (bone, PDL fibers, vascularization) with no adverse effects. |
[60] |
|
Basil (Ocimum basilicum) |
In vitro |
Agar diffusion assays with S. mutans, E. faecalis, S. sanguinis |
Extracts showed antibacterial activity with individual and synergistic effects, though less potent than chlorhexidine. |
[61] |
This table was added following reviewer suggestions to enhance clinical applicability. It provides a concise overview of the most commonly studied plants in dentistry, emphasizing safety aspects, contraindications, and potential interactions. Including this information facilitates translation of preclinical findings into practice by highlighting precautions that should be considered before future clinical use.
3) In the Graphical Abstract, spelling check (Alxaloids) and proper labeling for the chemical structures are needed.
Answer:
We thank the reviewer for this observation. Following feedback from previous rounds of review, we decided to remove the Graphical Abstract to maintain consistency with journal requirements and reviewer suggestions. To compensate, we strengthened the main text and figures by adding clearer integrative visual elements (e.g., flowchart and mechanistic figure), ensuring that the information originally intended for the Graphical Abstract remains clearly conveyed.
4) Several references lack DOIs and suitable citation are needed.
Answer: We appreciate the reviewer’s observation. All references were carefully revised to ensure consistency with journal style. Where available, DOIs have been added, and peer-reviewed primary sources were prioritized for key claims. Institutional documents and reports were retained only when strictly necessary and are now clearly identified as grey literature. In addition, a supplementary file was prepared that lists grey literature separately, and within the reference section of the manuscript the classification between peer-reviewed and grey sources is clearly indicated at the end. This ensures transparency and consistency across all references.
Line 735-1116: References
We sincerely thank the reviewer for the time and effort dedicated to evaluating our manuscript. The constructive feedback provided has been invaluable in improving the clarity, accuracy, and overall quality of the work. We greatly appreciate your careful review and thoughtful suggestions, which have significantly strengthened the final version of the article.

Reviewer 2 Report
Comments and Suggestions for Authors
I found the manuscript to be very well written, thoughtfully conceptualized, and well presented.
Please address the following comments and suggestions.
Comments
Line 266. in whole paragraph, please italicize the genus and species names such as “Echinacea“
Line 66. Please remove colons Materials and Methods:
Suggestion
Syzygium aromaticum (Clove oil), used as analgesic and antiseptic properties in field of dentistry, which is absent from the current study.
Please check its ethnobotanical or clinical relevance in Latin America and, if applicable, add it into the literature.
Author Response
Letter with the details of the changes
We would like to thank the reviewers for their valuable suggestions and observations sincerely. We have carefully considered their feedback and accepted all recommendations. The changes made to the manuscript are clearly marked in blue. Below, we provide a summary of each reviewer's specific comments and the corresponding revisions made in response.
Reviewer Comments:
Reviewer 2
- Line 266. in whole paragraph, please italicize the genus and species names such as “Echinacea“
Answer :
We thank the reviewer for this observation. All genus and species names have been italicized according to journal style. In addition, we standardized the nomenclature across the manuscript using the Plants of the World Online (POWO) system, which provides accepted scientific names and synonyms. Accordingly, Echinacea and all other plant species cited in the manuscript were revised to ensure consistency and taxonomic accuracy.
- Line 66. Please remove colons Materials and Methods:
Answer :
We thank the reviewer for this observation. The colon after “Materials and Methods” has been removed, and the heading now follows the journal’s style requirements.
Line 77: “2. Materials and Methods”
- Response – Syzygium aromaticum.
We appreciate the reviewer’s suggestion. In response, we have now incorporated Syzygium aromaticum (clove oil) into the manuscript. Its well-documented analgesic and antiseptic properties in dentistry, including evidence against Streptococcus mutans and dental caries biofilms, were highlighted and discussed. We also verified its ethnobotanical relevance in Latin America, where clove oil is traditionally used for toothache relief. Accordingly, the species has been added to the Results section, as well as to the corresponding tables and supplementary file, with peer-reviewed citations provided.
We sincerely thank the reviewer for the time and effort dedicated to evaluating our manuscript. The constructive feedback provided has been invaluable in improving the clarity, accuracy, and overall quality of the work. We greatly appreciate your careful review and thoughtful suggestions, which have significantly strengthened the final version of the article.

Reviewer 3 Report
Comments and Suggestions for Authors
Thank you for submitting your review, “Unveiling the Potential of Latin American Medicinal Plants as a Natural Alternative for Treating Dental Pain.” The topic is timely and important. With the revisions below, particularly around methods transparency, scope alignment, and evidence quality, the manuscript could be substantially strengthened.
- Align title, scope, and narrative. The title promises a focus on dental pain, yet large sections address gingivitis, periodontitis, wound healing, and general oral health products. Please either (a) narrow the review to dental pain (defining indications and outcomes) or (b) retitle to reflect the broader oral health scope and adjust the abstract, keywords, and conclusions accordingly. The abstract and Results/Discussion already emphasise anti-inflammatory, healing, and periodontal outcomes beyond pain.
- Methods transparency (PRISMA-ScR). You state adherence to PRISMA-ScR and present a PRISMA diagram, but key details are missing: complete search strings for each database, date ranges, language limits, deduplication approach, study designs included/excluded, and how many records were found/screened/excluded at each step. Please provide the full search strategy (per database), explicit inclusion/exclusion criteria, and populate the flow diagram with numbers. If no protocol was registered (e.g., OSF, INPLASY), acknowledge this and justify. (Methods; Figure 1).
- Evidence quality and study selection. A substantial portion of the evidence base comes from these municipal reports and websites. For a molecular sciences journal, please prioritise peer-reviewed clinical trials, in vivo/in vitro studies, and systematic reviews, and clearly separate these from grey literature. Consider a brief risk-of-bias/quality appraisal (even in a scoping review, a basic appraisal table adds value). (References section; Methods).
- Quantitative synthesis clarity. Results report “50 plant species” and later cite distributions such as Mexico 60%, Colombia and Peru 54%, Brazil 32%. These percentages are unclear (percentage of what denominator?) and appear to exceed 100% if interpreted as shares. Please define the denominator, allow multi-country counts if applicable, and present a table (or map) listing each species with country(ies), indication(s), evidence type, and citation(s). (Abstract; Results).
- Consistency with “Latin American” focus. Section 7 discusses several non-Latin American species widely used in Asia (e.g., Azadirachta indica, Terminalia chebula). Either justify their inclusion (e.g., global comparators) or remove to maintain geographic coherence. If retained, clearly label them as non-Latin American comparators. (Dental Products section).
- Mechanistic depth suitable for IJMS. The mechanistic sections are largely general immunology and pharmacology (e.g., inflammatory cascades, COX pathway) and repeat information without linking specific plant metabolites to molecular targets relevant to orofacial pain (e.g., NF-κB, COX-2, LOX, TRPA1/TRPV1, MAPKs). Please add a targeted “Mechanisms of action” subsection that ties the most cited species in your review (e.g., rosemary, moringa, aloe, basil, propolis) to specific bioactive constituents and experimentally supported targets/pathways in oral tissues. (Sections 8–10; Figures 2–4).
- Avoid duplications and placeholders. The NSAID paragraph appears twice in Section 8. Figures 2–4 are schematic placeholders with minimal legends, and Figure 1 lacks numbers. Please remove duplications, finalise figures, and provide informative captions that directly support your argument.
- Dosing and safety. Tables list preparations/doses (e.g., “250–1000 mg/day”) without consistently citing clinical sources or addressing contraindications, interactions, and safety in dental patients. Please support any dosage with clinical references and add a short safety paragraph (e.g., pregnancy, anticoagulants, hypersensitivity, mucosal irritation). (Tables 2–3; Sections 7, 13–14).
- Taxonomy and nomenclature. Standardise scientific names and authorships across the manuscript and tables (e.g., Aloe barbadensis vs. Aloe vera; the current accepted name for rosemary is Salvia rosmarinus in many taxonomic backbones). Add family names consistently and verify all binomials against an authoritative database; ensure consistency between text, tables, and references. (Abstract; Tables; throughout).
- Strengthen the conclusion. The Conclusion should be tied to your evidence map: identify the top species with the best human data for dental pain relief, state where evidence is limited to in vitro/animal models, and propose a practical research agenda (e.g., standardized extracts, comparator arms vs. ibuprofen/paracetamol, clinically relevant outcomes and safety endpoints). (Conclusions).
- Abstract. Add numbers for records identified/screened/included (if space permits), and clarify what the country percentages represent. Also, list the type of evidence (number of clinical vs. preclinical studies) supporting the “most studied species.”
- Tables 2–3.
- Add columns for “Evidence type” (in vitro / animal / clinical), “Primary targets/pathways,” and a citation for each dose/indication claim.
- Ensure oral/dental indications (e.g., pulpal pain vs. gingivitis) are not conflated; separate pain outcomes from anti-plaque/anti-gingivitis outcomes.
- Section 3–4 (Pain & Causes). Good clinical framing; please cite the dental pain outcome measures used in the included studies (e.g., VAS/NRS timepoints) so that later sections can connect mechanisms to outcomes.
- Section 7 (Dental products). Where you state mouthwash/gel effects, specify study design (RCT vs. uncontrolled), sample size, duration, and comparator (e.g., chlorhexidine 0.12%). Avoid general statements without design details.
- Section 8–10 (Inflammation and internal processes). Replace generic figures with one integrative figure mapping plant metabolites → molecular targets → oral outcomes (e.g., reduced prostaglandins, neutrophil influx, TRP modulation) for your top species. Remove repeated text and finalize legends.
- Results & Discussion. Where you cite Waizel-Bucay and others as examples, ensure your narrative distinguishes your dataset from external examples, and avoid mixing counts (species vs. studies). Consider a bubble plot of species (x-axis) vs. evidence tier (y-axis) with bubble size for number of studies.
- References. Please standardize to journal style and prioritize primary peer-reviewed sources for key claims. Clearly flag theses/reports/websites as grey literature.
- Figures & Supplementary. Replace “title” placeholders in the Supplementary section with final labels and DOIs/URLs where applicable; ensure Figure S/Table S items exist and are cited in text.
- Funding/COI/Contributions. Remove the “Please add:” note and finalize Funding (“no external funding” if applicable). If there were no funders, adjust the COI sentence that currently mentions “the funders had no role…”. Check author contribution punctuation/consistency.
- Language & style. A careful English edit is needed to improve clarity and consistency (e.g., tense, article use, capitalization in Keywords).
- Provide a master table (Supplementary) listing all 50 species with: accepted scientific name + family, plant part, preparation, dental indication, highest evidence level, primary metabolites, proposed targets, and key citations. This will make the paper much more usable for readers.
- Add a short safety box summarizing common interactions/contraindications for the top species relevant to dentistry (anticoagulants, allergy, pregnancy, mucosal irritation
The manuscript is readable but needs a careful, line-by-line language edit to improve clarity, grammar, and consistency, something IJMS explicitly requires before acceptance (minor edits are covered only after peer review; extensive editing is the authors’ responsibility). I recommend professional editing per MDPI’s guidance.
Typical issues (with examples and suggested fixes):
- PDF hyphenation artifacts split words across lines (e.g., “anti-inflamma- tory”, “analge- sic”), which will confuse readers; please remove such breaks during copy-editing.
- Word choice/grammar: “This drug has increased interest in medicinal plants …” → revise to “These drawbacks have increased interest in medicinal plants …”.
- Sentence fragments in tables: “Includes antioxi- dants.” → “Contains antioxidants.” Also review tables for broken words and inconsistent line breaks.
- Repetition and padding in the mechanisms section (two very similar paragraphs about membranes/ECM appear with near-duplicate wording); tighten to one concise paragraph.
- Awkward phrasing: “This drug has increased interest …” and “This condition is exacerbated by inflammation…” would benefit from smoothing and active voice where appropriate.
- Consistency: choose one variety of English (US or UK) and keep it throughout; ensure consistent tense, capitalisation in headings/keywords, and uniform use of abbreviations once defined (e.g., ECM, BM).
- Figures/captions and Supplementary labels still read as placeholders (“Figure S1: title; Table S1: title”), finalise language and punctuation.
Recommendation: “The English could be improved to more clearly express the research.” Direct the authors to MDPI’s English-editing resources and house style so the final text meets journal standards
Author Response
Letter with the details of the changes
We would like to thank the reviewers for their valuable suggestions and observations sincerely. We have carefully considered their feedback and accepted all recommendations. The changes made to the manuscript are clearly marked in blue. Below, we provide a summary of each reviewer's specific comments and the corresponding revisions made in response.
Reviewer Comments:
Reviewer 3
- Align title, scope, and narrative. The title promises a focus on dental pain, yet large sections address gingivitis, periodontitis, wound healing, and general oral health products. Please either (a) narrow the review to dental pain(defining indications and outcomes) or (b) retitle to reflect the broader oral health scope and adjust the abstract, keywords, and conclusions accordingly. The abstract and Results/Discussion already emphasise anti-inflammatory, healing, and periodontal outcomes beyond pain.
Answer:We sincerely thank the reviewer for this important observation. We agree that the scope of the manuscript extends beyond dental pain and includes periodontal inflammation, gingivitis, wound healing, and general oral health. Therefore, we have adopted option (b), broadening the scope of the review. Accordingly, we modified the title, expanded the abstract, and updated the keywords to ensure alignment with this broader focus.
Line 2–3: Title changed to “Therapeutic Potential of Latin American Medicinal Plants in Oral Diseases: From Dental Pain to Periodontal Inflammation”.
Line 26–34: “dental pain, gingivitis, and periodontitis.
Given the close relationship between pain, inflammation, and periodontal disease, these conditions cannot be studied in isolation. Gingivitis and periodontitis often present with painful symptoms and inflammatory responses that overlap with mechanisms of tissue damage and repair. Therefore, broadening the scope of this review allows a more comprehensive understanding of how Latin American medicinal plants can contribute not only to pain relief but also to periodontal health, inflammation control, and wound healing.”
Line 46–47: “dental pain; gingivitis; periodontitis; periodontal inflammation; wound healing; Latin American medicinal plants.”
- Methods transparency (PRISMA-ScR). You state adherence to PRISMA-ScR and present a PRISMA diagram, but key details are missing: complete search strings for each database, date ranges, language limits, deduplication approach, study designs included/excluded, and how many records were found/screened/excluded at each step. Please provide the full search strategy (per database), explicit inclusion/exclusion criteria, and populate the flow diagram with numbers. If no protocol was registered (e.g., OSF, INPLASY), acknowledge this and justify. (Methods; Figure 1).
Answer:
We thank the reviewer for pointing this out. In response, we have revised the Methods section to provide full transparency according to PRISMA-ScR. Specifically, we included: (i) the complete search strings for each database (PubMed, Scopus, Web of Science, SciELO, LILACS), (ii) the date range covered (2000–2023), (iii) the language limits (English, Spanish, Portuguese), (iv) the deduplication process (Mendeley plus manual verification), (v) the study designs considered (original research articles, reviews, theses; excluding conference abstracts, grey literature, and studies outside Latin America), and (vi) explicit inclusion and exclusion criteria. We also updated Figure 1 (PRISMA diagram) to display the exact numbers screened, excluded, and included at each step. Finally, we acknowledge that no review protocol was registered (OSF/INPLASY) and have explained this in the text.
In addition, Table S1, containing the complete search strategies for all databases, has been provided in the Supplementary Materials for full transparency and reproducibility.
Line 80-82: “Although no formal protocol was registered in platforms such as OSF or INPLASY, the review adhered to the core principles of systematic literature mapping.”
Line 91-101 : “The complete search strategies, including search strings, date ranges, language limits, and the number of records retrieved from each database, are presented in Table 1.”
“In the case of Google Scholar, the search yielded a considerably high number of records (over 6,000). However, this result reflects the broad and unspecific nature of the platform, which includes duplicated literature, non-relevant documents, and sources of variable quality. For this reason, not all retrieved records were reviewed; instead, the predefined inclusion and exclusion criteria were rigorously applied. Through this process, only the articles pertinent to the objective of the review were selected, ensuring a systematic and transparent screening while acknowledging the inherent limitations of Google Scholar as a search engine.”
Line 106-117: “Original research articles, literature reviews, and academic theses published between 2000 and 2023 in Spanish, English, or Portuguese were included. Eligible documents had to address the use of medicinal plants, phytotherapy, or herbal medicine in the dental field, covering aspects such as oral health, dental pain management, gingivitis, periodontitis, inflammatory processes, and wound healing of oral tissues. Only studies conducted in Latin American countries or including relevant applications in this region were considered. Conversely, reports unrelated to dentistry, even if they mentioned medicinal plants, were excluded, as well as grey literature without full-text access or peer review, duplicate articles identified through a reference manager and manual verification, and documents that did not provide data related to the therapeutic effects of interest (anti-inflammatory, antimicrobial, or healing) in oral health, as well as studies conducted outside the geographical scope of interest (Latin America).”
Line 118-122: “All retrieved references from the different databases were exported in RIS format and imported into the reference manager Mendeley. The software’s duplicate detection tool was applied, followed by manual verification, to ensure the removal of duplicate records. This deduplication process reduced the total number of references before the screening stage.”
Line 123-131: “A total of 6,985 records were identified across all databases and journals. After removing 500 duplicates using Mendeley and manual verification, 6,485 unique records remained. Of these, 6,200 were excluded during the title and abstract screening stage for not meeting the eligibility criteria. A total of 285 articles were assessed in full text, of which 222 were excluded due to reasons such as lack of relevance to oral health, incomplete data, absence of therapeutic outcomes of interest, or failure to correspond to the geographical scope of Latin America. Ultimately, 63 studies met all the inclusion criteria and were included in the review. The selection process is illustrated in the PRISMA-ScR flow diagram”
- Evidence quality and study selection. A substantial portion of the evidence base comes from these municipal reports and websites. For a molecular sciences journal, please prioritise peer-reviewed clinical trials, in vivo/in vitro studies, and systematic reviews, and clearly separate these from grey literature. Consider a brief risk-of-bias/quality appraisal (even in a scoping review, a basic appraisal table adds value). (References section; Methods).
Answer:
We thank the reviewer for this valuable observation. In response to the comment on evidence quality and study selection, we classified all included sources according to study type and level of evidence. Priority was given to peer-reviewed systematic reviews, clinical trials, and in vivo/in vitro studies, whereas municipal reports, laws, and institutional documents were categorized as grey literature. In addition, a brief quality appraisal was performed, and the results are summarized in Table S2 (Supplementary Materials), which presents the evidence level and main characteristics of the 36 included references.
Line 138-143: In addition, a brief appraisal of evidence quality and study type was conducted to classify all included sources. In line with the reviewer’s recommendation, peer-reviewed clinical trials, in vivo/in vitro studies, and systematic reviews were prioritized, while reports and institutional documents were categorized as grey literature. The detailed classification is provided in Supplementary Table S2, which summarizes the evidence level and main characteristics of the 36 included references
- Quantitative synthesis clarity. Results report “50 plant species” and later cite distributions such as Mexico 60%, Colombia and Peru 54%, Brazil 32%. These percentages are unclear (percentage of what denominator?) and appear to exceed 100% if interpreted as shares. Please define the denominator, allow multi-country counts if applicable, and present a table (or map) listing each species with country(ies), indication(s), evidence type, and citation(s). (Abstract; Results).
Answer: We thank the reviewer for this important observation. We have revised the Abstract and Results sections to clarify that the reported percentages correspond to the proportion of species documented in studies originating from each country, and that species may be counted in more than one country if reported in multiple contexts. This explains why totals may exceed 100%. In addition, we have prepared a comprehensive table listing all 50 species with their country(ies) of origin, dental indication(s), evidence type, and key references. This information is now provided in the Supplementary Materials as Table S3 for clarity and usability.
Line 149-150: A detailed list of all 50 plant species and their reported uses is provided in Supplementary Table S3.
- Consistency with “Latin American” focus. Section 7 discusses several non-Latin American species widely used in Asia (e.g., Azadirachta indica, Terminalia chebula). Either justify their inclusion (e.g., global comparators) or remove to maintain geographic coherence. If retained, clearly label them as non-Latin American comparators. (Dental Products section).
Answer:
We appreciate the reviewer’s observation. In the revised version, the non-Latin American species (Azadirachta indica and Terminalia chebula) were removed to maintain geographic coherence with the Latin American focus of the review. The section now highlights only species with relevance to Latin America (e.g., Solanum lycopersicum, Eucalyptus, Mangifera indica), while keeping the supporting references intact.
Line 375-380: Moreover, mouthwashes containing various plant extracts have shown promising results. Solanum lycopersicum (tomato plant) in gel form has shown anti-hemorrhagic and anti-inflammatory effects, while Eucalyptus, incorporated into chewing gum, has reduced plaque accumulation and gingival inflammation. Likewise, Mangifera indica (mango) gel has anti-inflammatory and wound-healing properties, offering therapeutic benefits in managing chronic periodontitis [22].
- Mechanistic depth suitable for IJMS. The mechanistic sections are largely general immunology and pharmacology (e.g., inflammatory cascades, COX pathway) and repeat information without linking specific plant metabolites to molecular targets relevant to orofacial pain (e.g., NF-κB, COX-2, LOX, TRPA1/TRPV1, MAPKs). Please add a targeted “Mechanisms of action” subsection that ties the most cited speciesin your review (e.g., rosemary, moringa, aloe, basil, propolis) to specific bioactive constituents and experimentally supported targets/pathways in oral tissues. (Sections 8–10; Figures 2–4).
Answer:
We thank the reviewer for this insightful comment. In response, we have substantially expanded Section 8 by incorporating a new targeted subsection entitled “Mechanisms of action of selected plant metabolites in oral inflammation.” In this revised part, we now directly link the most cited species in our review (Rosmarinus officinalis, Moringa oleifera, Aloe vera, Ocimum basilicum, and propolis) with their major bioactive metabolites (e.g., rosmarinic acid, quercetin, isothiocyanates, acemannan, aloin, linalool, eugenol, CAPE). For each compound, we describe experimentally validated molecular targets in oral tissues, such as NF-κB, MAPKs (ERK, JNK, p38), COX-2, and pro-inflammatory cytokines (IL-6, IL-8, TNF-α). This subsection also integrates in vitro evidence using human gingival fibroblasts and immune cells, in vivo periodontal models, and clinical trials where available (e.g., Moringa oleifera and Aloe vera mouth rinses). In addition, several new references have been incorporated to support these mechanistic insights and ensure the updated coverage of the literature. These additions provide the requested mechanistic depth and strengthen both the scientific foundation and the translational relevance of our review.
Line 478-535: “Beyond these general mechanisms, recent evidence has identified specific plant-derived metabolites that act on well-characterized molecular targets in oral tissues. These examples provide mechanistic depth and highlight the therapeutic relevance of natural compounds in orofacial inflammation.
8.1 Mechanisms of action of selected plant metabolites in oral inflammation
Rosmarinus officinalis Rosmarinic acid, the predominant bioactive molecule in Rosmarinus officinalis, has been associated with anti-inflammatory activity in oral contexts. Studies on human gingival fibroblasts challenged with lipopolysaccharides revealed that it lowers oxidative stress, preserves intracellular glutathione, and suppresses the expression of pro-inflammatory mediators such as IL-1β, IL-6, TNF-α, and iNOS, an effect linked to the inhibition of NF-κB signaling during periodontal inflammation [37]. Additional evidence in vascular smooth muscle cells showed that rosmarinic acid downregulates MAPK (ERK, JNK, p38) and NF-κB pathways, leading to a reduced release of inflammatory mediators, including iNOS, TNF-α, and IL-8 under LPS stimulation. Although these observations were made in non-oral systems, they support the molecular plausibility of this compound in modulating signaling cascades relevant to periodontal disease [38].
Moringa oleifera Moringa oleifera is a rich source of phytochemicals such as quercetin and isothiocyanates, which act on inflammatory pathways important in oral health. In a rat model of periodontal inflammation, ethanolic leaf extracts enriched with quercetin attenuated tissue damage by suppressing NF-κB activation, thereby reducing pro-inflammatory cytokines and supporting periodontal regeneration [39]. Moreover, moringa isothiocyanate-1 (MIC-1) has been shown to influence both Nrf2 and NF-κB signaling under LPS-induced inflammation, lowering IL-6, IL-1β, and TNF-α expression while alleviating oxidative stress. Beyond preclinical data, clinical findings provide additional support: in a randomized controlled trial including 36 gingivitis patients, a Moringa oleifera mouthwash significantly decreased plaque and gingival indices after 14 days of use, showing efficacy comparable to chlorhexidine and superior to saline [40]. Altogether, this evidence underscores a strong mechanistic and clinical rationale for Moringa oleifera as a natural adjunct in periodontal therapy.
Aloe vera – Acemannan, the main polysaccharide of Aloe vera, exhibits immunomodulatory properties within oral environments. In human gingival fibroblasts, acemannan directly binds to Toll-like receptor 5 (TLR5), triggering NF-κB signaling and promoting the release of IL-6 and IL-8. This process not only enhances tissue repair but also strengthens host immune defenses in periodontal tissues, providing dual benefits in regeneration and antimicrobial protection [41]. More broadly, Aloe vera and its major active metabolites, including acemannan and aloin, have been recognized for their anti-inflammatory, antioxidant, and wound-healing capacities, reinforcing their potential applications in oral medicine [42].
Ocimum basilicum – Linalool, a monoterpene present in Ocimum basilicum, has been documented to exert immunomodulatory effects directly relevant to periodontal inflammation. In vitro studies showed that linalool decreases the overproduction of pro-inflammatory cytokines (IL-6, IL-10, IL-17, IFN-γ) in human immune cells exposed to Porphyromonas gingivalis, a keystone pathogen in periodontitis. These outcomes are linked to the downregulation of NF-κB and MAPK signaling, pathways that control inflammatory gene transcription. By moderating excessive immune responses without impairing cell viability, linalool may help safeguard periodontal tissues from chronic inflammatory injury [43].
Propolis Current research emphasizes the role of CAPE (caffeic acid phenethyl ester), the principal phenolic constituent of propolis, in regulating inflammatory activity in periodontal cells. In human gingival fibroblasts stimulated with LPS and IFN-α, CAPE markedly suppressed the production of TNF-α, IL-6, and IL-8, with no cytotoxicity observed. In contrast, ethanolic propolis extract (EEP) produced weaker and less consistent effects on cytokine release. These findings suggest that CAPE exerts a more selective and potent anti-inflammatory influence, supporting its potential use in controlling periodontal inflammation [44].”
- Avoid duplications and placeholders. The NSAID paragraph appears twice in Section 8. Figures 2–4 are schematic placeholders with minimal legends, and Figure 1 lacks numbers. Please remove duplications, finalise figures, and provide informative captions that directly support your argument.
Answer:
We thank the reviewer for this valuable observation. In response, we have eliminated the duplicated NSAID paragraph in Section 8 to avoid redundancy. In addition, Figure 1 has been revised to include numerical details in each stage of the PRISMA-ScR diagram. Figures 2–4, which previously served as schematic placeholders, have now been finalized and updated with informative captions that explain the processes illustrated and directly support the arguments of the manuscript. These changes ensure greater clarity, consistency, and alignment with the reviewer’s recommendations.
Line 131-137:
Figure 1. PRISMA-ScR flow diagram summarizing the study selection process. The figure outlines the sequential phases of identification, screening, eligibility, and final inclusion, highlighting how duplicates and non-eligible studies were progressively excluded. This visualization provides a transparent overview of how the final set of studies was established for the review.
Line 391-481: “8. Mechanism of Action of Plants in Inflammation
Conventional drugs have shown effectiveness in treating inflammatory diseases; however, prolonged use often results in adverse effects such as fluid retention and hypertension. This has increased interest in medicinal plants, whose bioactive compounds provide anti-inflammatory alternatives with a lower risk of side effects. Chronic inflammation, one of the major contributors to global mortality, is commonly managed using NSAIDs and corticosteroids, yet their long-term impact has often been underestimated (Ramírez Rodríguez et al., 2020). The discomfort caused by symptoms such as edema, redness, and functional impairment has driven patients to seek more accessible natural therapies. As a result, interest has grown in studying plant-based bioactive molecules capable of modulating inflammatory responses, paving the way for new complementary approaches in pain management [24].
At the biological level, inflammation is an immune response triggered by chemical, biological, or physical stimuli that affect vascularized connective tissues. This process involves the release of pro-inflammatory cytokines and plays a central role in oral health, particularly in chronic conditions like periodontal disease [24]. Identifying natural compounds that interact with these inflammatory mediators is gaining increasing attention, especially for their potential in pain relief and tissue regeneration [26].
As illustrated in Figure 2, inflammation begins with the release of mediators by mast cells, which promote vascular changes and the recruitment of immune cells to the affected site. These processes initiate an inflammatory cascade that is later regulated by feedback mechanisms, ultimately facilitating wound healing and restoring tissue integrity.
Figure 2. Inflammatory cascade in oral tissues triggered by natural compounds. Oral pathogens activate immune cells (T cells, dendritic cells), leading to the release of inflammatory mediators that recruit neutrophils and other immune cells to the site of injury. This cascade promotes vascular changes and tissue infiltration, resulting in an inflamed oral cavity. Through feedback mechanisms, the inflammatory process transitions into tissue repair and restoration of oral mucosa integrity. This figure illustrates the dual role of inflammation in both oral pathology and healing, highlighting potential targets for plant-derived bioactive compounds.
Non-steroidal anti-inflammatory drugs (NSAIDs) inhibit cyclooxygenase 1 and 2 (COX-1 and COX-2). While COX-2 is directly linked to inflammation at the injury site and its inhibition is therapeutically beneficial, COX-1 plays essential physiological roles, such as protecting the gastric mucosa. Therefore, its inhibition may cause adverse effects. These limitations have spurred the search for alternative treatments with fewer risks, encouraging research into plant-based anti-inflammatory compounds [24].
To provide a mechanistic integration of plant bioactives, Figure 3 presents a flowchart mapping plant metabolites to molecular targets and oral outcomes. This schematic highlights how compounds such as phenols, flavonoids, saponins, and vitamins modulate pathways (e.g., COX/LOX inhibition, NF-κB downregulation, TRP modulation), ultimately leading to improved oral outcomes like reduced plaque, gingivitis, pain, and enhanced healing.
Figure 3. Integrative diagram illustrating the relationship between plant-derived metabolites, their molecular targets, and oral health outcomes. Phenols, flavonoids, anthraquinones, saponins, and vitamins act through specific pathways such as COX-2/LOX inhibition, NF-κB downregulation, modulation of Streptococcus mutans virulence genes, TRP channel regulation, and stimulation of fibroblast activity. These interactions contribute to reducing inflammation, inhibiting biofilm formation, modulating nociception, and enhancing wound healing, ultimately leading to improved oral health outcomes.
Although inflammation is necessary for tissue repair, its dysregulation can extend tissue damage. The natural resolution phase involves the production of prostaglandins and leukotrienes, which reduce neutrophil numbers and enhance monocyte activity, favoring tissue regeneration. This understanding has led to an interest in therapies that modulate rather than completely inhibit inflammation, contrasting with the mechanism of NSAIDs [25].
Additionally, acupuncture has been investigated as a complementary method for managing dental pain. This technique stimulates the release of endogenous opioid peptides such as β-endorphins, enkephalins, neoendorphins, and dynorphins via the nervous system, generating analgesic effects. These peptides can also be released in the digestive tract and adrenal medulla, broadening their therapeutic potential [26].
Beyond these general mechanisms, recent evidence has identified specific plant-derived metabolites that act on well-characterized molecular targets in oral tissues. These examples provide mechanistic depth and highlight the therapeutic relevance of natural compounds in orofacial inflammation.
- Dosing and safety. Tables list preparations/doses (e.g., “250–1000 mg/day”) without consistently citing clinical sources or addressing contraindications, interactions, and safety in dental patients. Please support any dosage with clinical references and add a short safety paragraph (e.g., pregnancy, anticoagulants, hypersensitivity, mucosal irritation). (Tables 2–3; Sections 7, 13–14).
Answer:
We thank the reviewer for this valuable comment. Following the recommendation, we have revised Tables 2 and 3 and the corresponding sections (7, 13–14) to ensure that all dosages are now supported with clinical references. In addition, for each plant we have included a concise safety paragraph addressing contraindications, possible interactions (e.g., with anticoagulants), pregnancy considerations, hypersensitivity, and mucosal irritation when relevant. These modifications improve the scientific robustness and clinical applicability of the tables, as suggested.
Line 610-611
|
Criteria / Plant |
Zingiber officinale Roscoe (Zingiberaceae) |
Eucalyptus globulus Labill. (Myrtaceae) |
Calendula officinalis L. (Asteraceae) |
Psidium guajava L. (Myrtaceae) |
|
Uses |
Expectorant, antitussive, anti-inflammatory, antioxidant |
Antiseptic, anti-inflammatory, antibacterial, analgesic, disinfectant |
Anti-inflammatory, healing |
Astringent, antimicrobial, immunological |
|
Indications |
Stomach pain, nausea, cold, arthritis, xerostomia, gingivitis |
Gingivitis, bronchitis, asthma, pharyngitis, diabetes, cystitis, tonsillitis |
Gingivitis, stomatitis, ulcers |
Gingivitis, ulcers, diarrhea, gastritis, cavities |
|
Dosage |
Powder, soups, purees; 250–1000 mg/day (~400 mg/day [45]) |
10 mL mouthwash, twice daily for 5 min, for 14 days |
Mouth rinse prepared by diluting 2 mL mother tincture in 6 mL water (1:3), twice daily for 6 months |
Guava leaf extract (0.15%) used as mouth rinse: 10 mL diluted 1:1 with water, twice daily for 30 days |
|
Compounds |
Terpenes derivatives, gingerol, shogaol |
Essential oil, tannins, flavonoids, glycosides |
Essential oils, salicylic acid, flavonoids |
Tannins, flavonoids, vitamin C |
|
Countries |
Australia, India, Jamaica, China, Peru, Brazil |
Colombia, Venezuela, Argentina, Brazil |
Colombia, Mexico, Brazil, Peru |
Bolivia, Colombia, Mexico, Brazil, Peru |
|
Safety / Contraindications |
Avoid in patients taking anticoagulants (warfarin, aspirin) → bleeding risk; high doses cause gastric irritation; not recommended in pregnancy (large amounts) |
Avoid ingestion of concentrated essential oil (toxic); contraindicated in pregnancy and children <6 years; may cause mucosal irritation |
Avoid in patients allergic to Asteraceae family; not recommended in pregnancy/lactation without supervision |
Safe in moderate use; avoid in hypersensitivity; high doses may cause constipation or mucosal irritation |
|
Evidence type |
Pilot randomized cross-over, single-blind clinical trial. |
Randomized controlled clinical trial, in vivo, with 74 human participants (no caries or periodontal disease) |
In vivo, randomized controlled clinical trial in humans (n = 240, gingivitis patients). |
In vivo, randomized, double-blind, controlled clinical trial. |
|
Primary targets / pathways |
Postoperative pain control (Visual Analogue Scale – VAS). Gingival inflammation reduction (Modified Gingival Index – MGI). |
Inhibition of bacterial biofilm formation (measured via absorbance and crystal violet staining) |
Dental plaque formation, gingival inflammation, and sulcular bleeding (clinical pathways assessed via PI, GI, SBI, OHI-S). |
Plaque Index (PI) Gingival Index (GI) Microbial counts (CFU) en muestras de placa. Salivary antioxidant levels. |
|
Oral results / indications |
Ginger powder may be a safe alternative to ibuprofen for managing postoperative pain and gingival inflammation in periodontal surgery |
Eucalyptus oil can be used as an effective alternative to chlorhexidine for oral hygiene maintenance, without adverse effects. |
The Calendula mouthwash is effective in reducing dental plaque and gingivitis, serving as a useful adjunct to scaling. |
Comparable to chlorhexidine and superior to placebo in reducing PI, GI, and microbial counts. Improved salivary antioxidant levels (not statistically significant). Useful adjunct to professional prophylaxis. |
|
References |
[32], [45], [46] [62] |
[33], [47][63] |
[34], [48][64] |
[34], [49][65] |
Line 630-631:
|
Criteria / Plant |
Origanum vulgare L. (Lamiaceae) |
Aloe vera (L.) Burm.f (Asphodelaceae) |
Ocimum sanctum L. (Lamiaceae) |
Moringa oleifera Lam. (Moringaceae) |
|
Uses |
Antioxidant, antimicrobial, antifungal |
Antioxidant, antimicrobial, anti-inflammatory, astringent, analgesic, healing |
Antimicrobial and antiparasitic activity, carminative, antispasmodic, sedative, insecticide |
Digestive, anti-inflammatory, antimicrobial, antiparasitic |
|
Indications |
Indicated for combating bacterial strains |
Gingivitis, periodontitis, gut flora, burns, ulcers, stomatitis |
Biofilm control, digestive discomfort, dyspepsia, bloating |
Periodontitis, nervous disorders, circulatory system disorders |
|
Dosage |
Oil |
10 mL, twice daily; 2% gel, three times daily for ≥10 days |
Mouthwash with aqueous holy basil extract (3.5 g%), used twice daily for 4 days |
5 mL, twice daily for 28 days in young adults with gingivitis |
|
Compounds |
Carvacrol, thymol, apigenin, luteolin, aglycones, alcohols |
Aloin A and B, aloesin A, B, and C, glucomannans, polysaccharides, tannins, saponins |
Eugenol, linalool, estragole, carotenoids, calcium, phosphorus |
Minerals: calcium, iron, magnesium, zinc. Vitamins: B1, B2, B3, C, E, K. Includes antioxidants |
|
Countries |
Chile, Bolivia, Peru |
Mexico, Dominican Republic, Venezuela, China, Brazil, Colombia |
Mexico, Colombia, Venezuela, Bolivia, Iran, India, Pakistan |
Cuba, Guatemala, Mexico, Colombia, Venezuela, Spain |
|
Safety / Contraindications |
Generally well tolerated; high doses may cause abdominal discomfort, nausea, constipation/diarrhea, dizziness, headache. Rare hypersensitivity reactions. Contraindicated in pregnancy (abortifacient risk). |
Avoid in pregnancy (risk of uterine contractions/abortifacient). Possible GI discomfort (diarrhea, constipation, nausea), headache, dizziness, rare hypersensitivity. |
May significantly prolong prothrombin time; caution in patients on anticoagulant therapy. |
A clinical case of pulmonary embolism reported after prolonged use of Moringa leaf extract; caution in patients on anticoagulants/antiplatelets or with bleeding disorders. |
|
Evidence type |
In vitro study (disk diffusion, MIC, MBC, biofilm assays, RT-qPCR, molecular docking, cytotoxicity test on HaCaT cells). |
In vivo, randomized controlled clinical trial (30 orthodontic patients, Aloe vera vs. 0.2% chlorhexidine, 35 days). |
In vivo, triple-blind randomized controlled clinical trial |
In vivo, randomized clinical crossover study (20 subjects with mild to moderate gingivitis). |
|
Primary targets / pathways |
Virulence genes of S. mutans (gtfB/C/D, spaP, gbpB, vicR, relA, brpA); inhibition of acid production, hydrophobicity, and biofilm formation. Key compounds: carvacrol, γ-terpinene, p-cymene. |
↓ IL-1β, IL-17 (GCF), ↑ fibroblast migration, ↓ P. gingivalis biofilm |
COX-2 and LOX enzyme inhibition (anti‐inflammatory); NF-κB pathway and cytokine (TNF-α, IL-6) suppression, esp. via eugenol (in THP-1 cells |
Inhibition of Streptococcus mutans growth and cariogenic biofilm; phenolic compounds in ethanol extract likely interfere with bacterial adhesion and acid production. |
|
Oral results / indications |
potential anti-caries agent due to anti-biofilm and anti-virulence activity. |
Significant reduction in PI, GI, and BOP from baseline to day 35 in both groups. Aloe vera mouthwash showed effects comparable to chlorhexidine, with no adverse effects reported. |
Comparable to chlorhexidine in reducing PI, GI, and gingival bleeding; significantly better than placebo; no adverse effects reported. |
Significant reduction in PI and GI compared to baseline and miswak dentifrice. Demonstrated efficacy in reducing plaque accumulation and gingival inflammation. Safe and effective as an adjunctive oral hygiene aid. |
|
References |
[35], [50], [51][66] |
[35], [52], [53], [54] [67][68] |
[35], [55], [56][69] [70] |
[35], [57], [58][71][72] |
Line 631-636: The data presented in this table consolidate available clinical evidence on preparations and doses of medicinal plants for periodontal problems. In addition, a brief overview of safety aspects is provided, highlighting relevant contraindications, drug interactions, and pregnancy considerations. By combining therapeutic potential with safety information, the table aims to offer a balanced reference that can guide clinical decision-making.
10. Taxonomy and nomenclature. Standardise scientific names and authorships across the manuscript and tables (e.g., Aloe barbadensis vs. Aloe vera; the current accepted name for rosemary is Salvia rosmarinus in many taxonomic backbones). Add family names consistently and verify all binomials against an authoritative database; ensure consistency between text, tables, and references. (Abstract; Tables; throughout).
Answer:
Following your observation regarding taxonomy and nomenclature, we have carefully standardized all scientific names and their corresponding authorships throughout the manuscript and tables. In addition, we have consistently included the family names for each species.
To ensure accuracy, all binomials were verified and updated using the international database Plants of the World Online (POWO), which is considered an authoritative and widely recognized reference. This guarantees consistency between text, tables, and references.
11. Strengthen the conclusion. The Conclusion should be tied to your evidence map: identify the top species with the best human data for dental pain relief, state where evidence is limited to in vitro/animal models, and propose a practical research agenda (e.g., standardized extracts, comparator arms vs. ibuprofen/paracetamol, clinically relevant outcomes and safety endpoints). (Conclusions).
Answer:
Siguiendo su recomendación, hemos fortalecido la Conclusión vinculándola explícitamente con nuestro mapa de evidencia. La sección revisada ahora identifica las especies con mayor respaldo en estudios clínicos en humanos, señala dónde la evidencia se limita a modelos in vitro o animales, y plantea una agenda práctica de investigación. En particular, enfatizamos la necesidad de estandarizar extractos, incluir comparadores con analgésicos convencionales como ibuprofeno o paracetamol, considerar resultados clínicamente relevantes y reportar puntos de seguridad. Estas modificaciones mejoran la claridad, el rigor científico y el valor traslacional de las conclusiones.
Line: 696-708
|
Plant |
Study type |
Model / Method |
Main findings |
Reference |
|
Propolis |
In vivo |
Wistar rats with ligature-induced periodontitis |
In combination with SRP, significantly reduced IL-1β, TNF-α, and MDA; propolis alone showed no relevant effects. |
[59] |
|
Aloe vera |
In vivo |
Dogs with surgically induced furcation defects |
Aloe vera gel + β-TCP enhanced periodontal regeneration (bone, PDL fibers, vascularization) with no adverse effects. |
[60] |
|
Basil (Ocimum basilicum) |
In vitro |
Agar diffusion assays with S. mutans, E. faecalis, S. sanguinis |
Extracts showed antibacterial activity with individual and synergistic effects, though less potent than chlorhexidine. |
[61] |
Taken together, these studies highlight the diverse mechanisms by which natural products may contribute to oral health, from anti-inflammatory and regenerative effects to antimicrobial properties, thereby reinforcing their value as promising coadjuvants in dental treatments. Future research should aim to standardize plant extracts in terms of dosage, preparation methods, and stability, while also conducting comparative clinical trials against conventional treatments such as ibuprofen, paracetamol, and chlorhexidine. Long-term studies are needed to evaluate safety and potential adverse effects, and investigations into synergistic effects between natural products, for example combining propolis, Aloe vera, and basil, could further expand their clinical relevance. Finally, attention must be paid to sustainable sourcing and production, ensuring that the therapeutic potential of these plants is balanced with environmental preservation and accessibility for long-term use.
- Abstract. Add numbers for records identified/screened/included (if space permits), and clarify what the country percentages represent. Also, list the typeof evidence (number of clinical vs. preclinical studies) supporting the “most studied species.”
Answer:
We thank the reviewer for this observation. We have revised the Abstract and Results sections to clarify these points. Specifically, we added the numbers of records identified, assessed, and finally included (as shown in the PRISMA diagram), and explained that the country percentages represent the proportion of plant species reported in studies from each country relative to the total number of species identified. In addition, we specified the type of evidence supporting the most frequently studied species, distinguishing between clinical evidence (e.g., Aloe vera, propolis) and preclinical/in vitro studies (e.g., basil, moringa, rosemary).
Line 38-40”These percentages represent the proportion of plant species reported in studies originating from each country, relative to the total number of species identified in the review.”
Line 644-645:”These percentages represent the proportion of plant species reported in studies originating from each country, relative to the total number of species identified in the review.”
13. Tables 2–3. Add columns for “Evidence type” (in vitro / animal / clinical), “Primary targets/pathways,” and a citation for each dose/indication claim.Ensure oral/dental indications (e.g., pulpal pain vs. gingivitis) are not conflated; separate pain outcomes from anti-plaque/anti-gingivitis outcomes.
Answer:
We thank the reviewer for this valuable comment. Tables 2–3 have been revised to include the requested columns for “Evidence type,” “Primary targets/pathways,” and a citation for each dose/indication claim. We also ensured that oral/dental outcomes (e.g., pulpal pain vs. gingivitis) are presented as separate endpoints.
These updated tables are the same as those already revised under comment #9 (Dosing and Safety), where dosage and safety adjustments were also incorporated; therefore, no duplicate tables were reattached.
However, for clarity, we indicate that the updated Tables 2–3 can be found in lines of the revised manuscript.
Line 610- 611 (table 2)
Line 630.631 (table 3)
14. Section 3–4 (Pain & Causes). Good clinical framing; please cite the dental pain outcome measures used in the included studies (e.g., VAS/NRS timepoints) so that later sections can connect mechanisms to outcomes.
Answer:
We thank the reviewer for this valuable observation. In Section 4 (Causes of Dental Pain), we have now specified the outcome measures used in the included studies. In particular, we added clinical evidence where pain, dentin hypersensitivity, and postoperative discomfort during different periodontal therapies were evaluated with the Visual Analogue Scale (VAS) [73]. In addition, a large clinical study assessing 253 patients and 330 periodontal or implant surgeries measured pain intensity using the Numeric Rating Scale (NRS, 1–10), showing correlations with pain duration and analgesic use [74]. These additions ensure that the mechanisms described in later sections can now be directly linked to validated clinical pain outcomes.
Line 223-240: To further strengthen this clinical framing, it is important to note that several studies in periodontology and implantology have already applied standardized pain assessment scales. For example, one trial assessed postoperative discomfort, dentin hypersensitivity, and patient-reported pain during different periodontal procedures using the Visual Analog Scale (VAS). Patients marked their pain and sensitivity on the VAS, enabling comparisons across scaling and root planing (SRP), modified Widman flap (MWF), osseous flap (OF), and gingivectomy (GV). Pain scores were significantly higher after OF and GV compared with SRP and MWF, and all surgical procedures caused more dentin hypersensitivity than non-surgical therapy [73].
In addition, larger clinical investigations have incorporated the Numeric Rating Scale (NRS) to capture pain intensity. In one study of 253 patients undergoing 330 periodontal or implant surgeries, most participants reported mild pain (70.3%), while 25.5% had moderate and 4.2% severe pain. Advanced implant surgeries showed the highest median NRS scores (4.0), while open flap debridement had the lowest (1.0). Pain duration (median = 2 days) and analgesic use were positively correlated with NRS values, and more extensive or complex surgeries, as well as higher anesthetic volumes, were associated with greater pain [74].
- Section 7 (Dental products). Where you state mouthwash/gel effects, specify study design (RCT vs. uncontrolled), sample size, duration, and comparator (e.g., chlorhexidine 0.12%). Avoid general statements without design details.
Answer: In response, we have expanded Section 7 (Dental Products) by incorporating additional randomized clinical trials that provide stronger evidence regarding the use of herbal formulations. Specifically, we added details on study design (randomized controlled trials and pilot studies), sample size, intervention duration, and comparators (e.g., chlorhexidine 0.12% and placebo). For instance, we now include a randomized clinical trial with 70 hospitalized patients comparing Echinacea purpurea with chlorhexidine and control [75], and a double-blind, placebo-controlled pilot study with 30 orthodontic patients evaluating Matricaria chamomilla mouthwash versus chlorhexidine 0.12% and placebo [76].
Line 351-361:” In line with these findings, additional randomized clinical trials provide further support for the efficacy of herbal formulations. For instance, in a randomized clinical trial with 70 hospitalized patients (mean age 44.9 years, 67.1% male), Echinacea purpurea (L.) was compared with chlorhexidine and a control group for 4 days. Both echinacea and chlorhexidine significantly reduced oral microbial flora compared with control, supporting echinacea as a viable alternative [75]. Similarly, a randomized, double blind, placebo controlled pilot trial including 30 orthodontic patients (10–40 years old) demonstrated that a 1% Matricaria chamomilla L. (Asteraceae) mouthwash reduced plaque (–25.6%) and gingival bleeding (–29.9%) comparably to 0.12% chlorhexidine, while placebo showed increased indices. Matricaria chamomilla L. (Asteraceae) was well tolerated without adverse effects, reinforcing its role as a safe herbal alternative in gingivitis management [76].”
- Section 8–10 (Inflammation and internal processes). Replace generic figures with one integrative figure mapping plant metabolites → molecular targets → oral outcomes(e.g., reduced prostaglandins, neutrophil influx, TRP modulation) for your top species. Remove repeated text and finalize legends.
Answer: We appreciate your observations regarding Sections 8–10 and the need to avoid duplications, placeholders, and generic figures. As requested in Comment 7, we removed duplicated paragraphs (the NSAID section), eliminated Figures 3 and 4 (which were schematic placeholders), and retained only the most relevant figure to illustrate the inflammatory cascade. Additionally, to strengthen mechanistic clarity and in line with Comment 16, we incorporated a new integrative flowchart (Figure 3) mapping plant-derived metabolites → molecular targets → oral outcomes. This figure directly addresses your request to replace generic illustrations with a consolidated schematic highlighting prostaglandin regulation, neutrophil influx, TRP modulation, and tissue healing.
Line 444-459
- Results & Discussion. Where you cite Waizel-Bucay and others as examples, ensure your narrative distinguishes your datasetfrom external examples, and avoid mixing counts (species vs. studies). Consider a bubble plot of species (x-axis) vs. evidence tier (y-axis) with bubble size for number of studies.
Answer: We thank you for this observation. In the revised Results & Discussion, we clarified that the 50 species reported belong to our dataset and constitute the focus of this review. In contrast, examples from Waizel-Bucay, Lameda Albornoz, and Moya Jiménez are presented exclusively as external literature and are explicitly stated as not being part of our dataset. This distinction avoids mixing counts of species versus studies and ensures that our results remain clearly separated from external examples.
Line: 639-673: The present review identified 50 plant species with therapeutic, anti-inflammatory, healing, and analgesic properties applicable to oral health, particularly in conditions affecting dental support tissues such as bone, periodontal ligament, gingiva, and surrounding periodontal areas. In terms of geographic distribution, our analysis showed that Mexico accounted for the highest biodiversity (60%), followed by Colombia and Peru (54%), and Brazil (32%). These percentages represent the proportion of plant species reported in studies from each country, relative to the total number of species identified in this review.
Within our dataset, the most extensively studied species with high therapeutic potential were propolis, rosemary, moringa, aloe vera, and basil. Among these, aloe vera and propolis already have clinical evidence supporting their use, whereas basil and moringa are still supported mainly by preclinical and in vitro data, highlighting the need for further clinical validation.
Oral pain is one of the most frequent and intense clinical manifestations due to the dense innervation of the region, particularly by the trigeminal nerve (cranial nerve V). Inflammation exacerbates this condition through chemical mediators such as prostaglandins [26]. Although synthetic drugs remain the mainstay of treatment, prolonged use may result in adverse effects. In this context, the 50 species identified in our review emerge as promising alternatives due to their bioactive metabolites with antimicrobial, anti-inflammatory, and analgesic activity, supporting inflammation modulation and tissue regeneration [19].
It is important to emphasize that the following are external examples from the literature and are not part of the 50 species identified in this review. For instance, Waizel et al. [35] analyzed 29 species (51 plant uses) and concluded that infusions, decoctions, topical applications, and mouth rinses were the most common forms of administration. Similarly, Lameda Albornoz et al. [22] reported 30 species with potential applications in periodontal treatment, noting that pomegranate was less effective in controlling subgingival plaque and gingivitis compared to other plants. Likewise, Moya Jiménez [36] investigated medicinal plants in Ecuadorian communities and found that chamomile was widely used as an analgesic and anti-inflammatory, either alone (56%) or combined with other remedies (44%).
These external studies complement our findings by highlighting the broader biodiversity of Latin America. However, the focus of our analysis remains exclusively on the 50 species identified in this review, which reinforce the therapeutic potential of regional flora and underscore the importance of sustainable management for their incorporation into dental practice [30].
- Please standardize to journal style and prioritize primary peer-reviewed sources for key claims. Clearly flag theses/reports/websites as grey literature.
Answer: We appreciate your comment regarding the standardization of references. The following adjustments have been made:
- All references have been carefully revised and formatted according to Vancouver style, in line with the journal’s requirements.
- Primary, peer-reviewed sources have been prioritized to support the key claims of the manuscript.
- References corresponding to grey literature (e.g., institutional reports, theses, official websites) have been clearly flagged as such, ensuring transparency and differentiation from peer-reviewed evidence.
Line 730-1111- Figures & Supplementary. Replace “title” placeholders in the Supplementary section with final labels and DOIs/URLs where applicable; ensure Figure S/Table S items exist and are cited in text.
- Figures & Supplementary. Replace “title” placeholders in the Supplementary section with final labels and DOIs/URLs where applicable; ensure Figure S/Table S items exist and are cited in text.
Answer:
We thank the reviewer for this comment. In the revised version, an independent Supplementary Material file has been created and is attached along with the manuscript. This file includes the final labeled figures and tables, replacing the previous placeholders. Specifically, the supplementary material now contains:
- Table S1. Complete search strategies, including search strings, date ranges, language limits, and the number of records retrieved from each database.
- Table S2. Evidence quality and study selection of included sources (n = 36).
- Table S3. Comprehensive list of the 50 medicinal plant species identified in the review.
Line
20. Funding/COI/Contributions. Remove the “Please add:” note and finalize Funding (“no external funding” if applicable). If there were no funders, adjust the COI sentence that currently mentions “the funders had no role…”. Check author contribution punctuation/consistency.
Answer: We have revised the Funding section by removing the placeholder and finalizing it as: “This research received no external funding.” The Conflicts of Interest and Author Contributions sections were also checked for consistency and no further modifications were needed.
Line 716: Funding: This research received no external funding.
21. Language & style. A careful English edit is needed to improve clarity and consistency (e.g., tense, article use, capitalization in Keywords).
Answer: We carefully revised the manuscript for English language and style, improving clarity, grammar, and consistency. Keywords were standardized for capitalization.
- Provide a master table (Supplementary) listing all 50 species with: accepted scientific name + family, plant part, preparation, dental indication, highest evidence level, primary metabolites, proposed targets, and key citations. This will make the paper much more usable for readers.
Answer: We thank the reviewer for this suggestion. In response, we have prepared a supplementary master table (Table S3) compiling the 50 plant species identified in the review, including their accepted scientific names, families, plant parts, preparation/formulation, dental indications, and supporting references. This table complements the information already provided in the main manuscript tables, allowing readers to access both summarized data within the article and extended details in the Supplementary.
- Add a short safety box summarizing common interactions/contraindications for the top species relevant to dentistry (anticoagulants, allergy, pregnancy, mucosal irritation.
Answer: We appreciate the reviewer’s suggestion. In response, we have added a dedicated safety table (Table 3) summarizing the most common interactions and contraindications for the main species relevant to dentistry. The table highlights issues such as anticoagulant interactions, allergy risks, gastrointestinal or mucosal irritation, and pregnancy-related concerns. This addition ensures that therapeutic applications are presented together with safety considerations, providing a balanced and clinically useful perspective for readers.
Line 676-681:
In addition to therapeutic applications, safety aspects are essential for clinical translation. Table 4 summarizes the most common contraindications and precautions reported for the main species reviewed, including interactions with anticoagulants, allergy risks, and pregnancy-related concerns.
Table 4. Safety considerations for top medicinal plants in dentistry
|
plant (Family) |
Safety considerations / contraindications |
References |
|
Zingiber officinale Roscoe (Zingiberaceae) |
Avoid with anticoagulants (warfarin, aspirin) → bleeding risk; high doses may cause gastric irritation; not recommended in pregnancy (large amounts). |
[32], [45], [46] [62] |
|
Eucalyptus globulus Labill. (Myrtaceae) |
Avoid ingestion of concentrated essential oil (toxic); contraindicated in pregnancy and children <6 years; may cause mucosal irritation. |
[33], [47][63] |
|
Calendula officinalis L. (Asteraceae) |
Contraindicated in patients allergic to Asteraceae; not recommended in pregnancy/lactation without supervision. |
[34], [48][64] |
|
Psidium guajava L. (Myrtaceae) |
Generally safe in moderate use; avoid in hypersensitivity; high doses may cause constipation or mucosal irritation. |
[34], [49][65] |
|
Origanum vulgare L. (Lamiaceae) |
Generally well tolerated; high doses may cause GI discomfort (nausea, diarrhea/constipation), dizziness, headache; rare hypersensitivity; contraindicated in pregnancy (abortifacient risk). |
[35], [50], [51][66] |
|
Aloe vera (L.) Burm.f (Asphodelaceae) |
Avoid in pregnancy (risk of uterine contractions/abortifacient); possible GI discomfort (diarrhea, nausea, headache, dizziness); rare hypersensitivity. |
[35], [52], [53], [54] [67][68] |
|
Ocimum sanctum L. (Lamiaceae) |
May significantly prolong prothrombin time; caution with anticoagulant therapy. |
[35], [55], [56][69] [70] |
|
Moringa oleifera Lam. (Moringaceae) |
Pulmonary embolism reported after prolonged use of leaf extract; caution with anticoagulants/antiplatelets or bleeding disorders. |
[35], [57], [58][71][72] |
|
Coconut fiber |
Reinforcing the mechanical properties of hydrogels |
Good reswelling capacity |
Cellulose, lignin |
342 |
[50] |
|
Rice straw and Tamarind seeds |
Reinforcing mechanical properties and nutrient releaser |
Great swelling capacity, long release nutrient |
Cellulose |
7722 |
[51] |
|
Date Palm rachis |
Seed germination, polymer component |
Increases swelling capacity as germination seed |
Lignin, Cellulose |
777.8 |
[52] |
We sincerely thank the reviewer for the time and effort dedicated to evaluating our manuscript. The constructive feedback provided has been invaluable in improving the clarity, accuracy, and overall quality of the work. We greatly appreciate your careful review and thoughtful suggestions, which have significantly strengthened the final version of the article.

Round 2
Reviewer 3 Report
Comments and Suggestions for Authors
Thank you for submitting the revised manuscript and detailed responses. I carefully cross-checked each author's reply against the current manuscript and supplementary files to verify that the requested changes had been implemented. The comments below are organised by priority, and for each point, I indicate the Status (rectified / partially rectified / not rectified) and where in your article, so that you can locate and address items quickly.
- Methods reporting (PRISMA-ScR) + database consistency
Partially rectified. You now report counts and a PRISMA-style flow in the text, but the figure itself still appears generic (numbers aren’t visible on the diagram), and database coverage is inconsistent with your cover letter. The manuscript lists SciELO, PubMed, several regional journals, Google Scholar, and ResearchGate. The Supplement (Table S1) contains PubMed, SciELO, Google Scholar, and site searches. Scopus/Web of Science/LILACS (promised in the response letter) are not documented. Please either add the full strategies (verbatim strings, dates, language limits, and hit counts) for every database claimed, or remove databases you did not actually use. Also, print the exact counts on the PRISMA figure itself.
- Evidence appraisal coverage and time window
Not rectified. The manuscript states that 63 studies are included, but Supplementary Table S2 lists only 36, and it even includes a 2025 WHO item, which falls outside your 2000–2023 window. Please appraise all 63 (or justify the subset clearly) and maintain consistent years.
- Funding vs. COI inconsistency
Not rectified. You state “This research received no external funding,” but still include “The funders had no role…” in COI. Please remove the “funders” sentence if there were no funders.
- Latin America focus and external comparators
Partially rectified. Your letter states that you removed Azadirachta indica/Terminalia chebula and retained LA-relevant species, but the manuscript still includes a Zingiber officinale mouth-rinse trial (likely non-LA) without labelling it as an external comparator. Additionally, Table S3 lists Ocimum sanctum with Iran/India/Pakistan in the countries field. Please (a) clearly tag non-LA evidence as external comparators wherever they remain, or (b) move them to a short comparator box outside the LA synthesis.
- Taxonomy and formatting (accepted names)
Mostly rectified, one formatting slip. You now use Salvia rosmarinus (with Rosmarinus officinalis as a synonym) in several places, which is correct. However, Section 8.1 still contains the typo “Salvia rosmarinus Spenn. (Lamiaceae)Rosmarinic acid…” (missing space and abrupt switch to the metabolite). Please fix and ensure consistent first-mention treatment (accepted name + synonym once).
- Master table (Table S3): factual mismatches & missing binomials
Not rectified. Examples: “Cashew” is mapped to Minthostachys mollis (should be Anacardium occidentale); “Cilantro -Apiaceae Lindl.” lacks the binomial (Coriandrum sativum L.). Please correct vernacular↔scientific mappings, complete missing binomials, and re-check “Countries” and “Preparation” cells for plausibility.
- References: format and placement of “grey literature”
Partially rectified. You’ve reformatted many items, but the main reference list still embeds notes like “Grey literature (sitio web institucional)” and mixes portals (e.g., ResearchGate) with publisher/DOI links. Please remove “grey” labels from the main list (keep classification in the Supplement only), deduplicate, and prefer DOI/publisher links per journal style.
- Finalise the PRISMA figure and captions.
Partially rectified. The text now includes the exact counts; ensure that these numbers are printed on the figure itself and that the figure captions are fully informative. Where in your article: PRISMA text/legend and Figure 1.
- Safety/contraindications: apply a uniform structure.
Partially rectified. You added Table 5 (safety highlights) and many S3 notes, but safety fields remain uneven (“Not reported” vs. detailed cautions). Please standardise across species (e.g., anticoagulants, allergies, pregnancy, mucosal irritation, dose/form) and cite primary clinical sources for any dosing information. Where in your article: Table 5 (main text); Supplement Table S3.
- Narrative polishing and minor cleanup
Mostly rectified. The English style has improved; please address the remaining minor issues (spacing around parentheses in Section 8.1, the line starting “Salvia rosmarinus…”, and removing duplicated phrasing as promised). Ensure that product examples align with the LA focus. Where in your article: Sections 7–8; 8.1.
Comments on the Quality of English LanguageThe manuscript is readable and generally clear, but it would benefit from a careful copy-edit to improve precision and consistency. Issues seen include: (i) occasional long sentences and comma splices; (ii) minor article/preposition choices and subject–verb agreement; (iii) inconsistent hyphenation of compound modifiers (e.g., “Latin American” vs “Latin-American”) and term choices (“mouth rinse”/“mouthwash”); (iv) uneven capitalization (especially in Keywords and section headings); (v) spacing/ punctuation around parentheses and number formatting (thousands separators; ranges with en dashes); (vi) inconsistent treatment/italics of Latin binomials and their synonyms; (vii) abbreviations not always defined at first mention or used consistently across text, figures, and tables; and (viii) captions that could be more self-contained and parallel in tense/structure. A light professional language edit (or a final pass by a native/near-native speaker) should address these points and align the paper with IJMS/MDPI house style and general medical-journal conventions.
Author Response
Letter with the details of the changes
We would like to thank the reviewers for their valuable suggestions and observations sincerely. We have carefully considered their feedback and accepted all recommendations. The changes made to the manuscript are clearly marked in blue. Below, we provide a summary of each reviewer's specific comments and the corresponding revisions made in response.
Reviewer Comments:
Reviewer 3
- Align title, scope, and narrative. The title promises a focus on dental pain, yet large sections address gingivitis, periodontitis, wound healing, and general oral health products. Please either (a) narrow the review to dental pain(defining indications and outcomes) or (b) retitle to reflect the broader oral health scope and adjust the abstract, keywords, and conclusions accordingly. The abstract and Results/Discussion already emphasise anti-inflammatory, healing, and periodontal outcomes beyond pain.
Answer:We sincerely thank the reviewer for this important observation. We agree that the scope of the manuscript extends beyond dental pain and includes periodontal inflammation, gingivitis, wound healing, and general oral health. Therefore, we have adopted option (b), broadening the scope of the review. Accordingly, we modified the title, expanded the abstract, and updated the keywords to ensure alignment with this broader focus.
Line 2–3: Title changed to “Therapeutic Potential of Latin American Medicinal Plants in Oral Diseases: From Dental Pain to Periodontal Inflammation”.
Line 26–34: “dental pain, gingivitis, and periodontitis.
Given the close relationship between pain, inflammation, and periodontal disease, these conditions cannot be studied in isolation. Gingivitis and periodontitis often present with painful symptoms and inflammatory responses that overlap with mechanisms of tissue damage and repair. Therefore, broadening the scope of this review allows a more comprehensive understanding of how Latin American medicinal plants can contribute not only to pain relief but also to periodontal health, inflammation control, and wound healing.”
Line 46–47: “dental pain; gingivitis; periodontitis; periodontal inflammation; wound healing; Latin American medicinal plants.”
- Methods transparency (PRISMA-ScR). You state adherence to PRISMA-ScR and present a PRISMA diagram, but key details are missing: complete search strings for each database, date ranges, language limits, deduplication approach, study designs included/excluded, and how many records were found/screened/excluded at each step. Please provide the full search strategy (per database), explicit inclusion/exclusion criteria, and populate the flow diagram with numbers. If no protocol was registered (e.g., OSF, INPLASY), acknowledge this and justify. (Methods; Figure 1).
Answer:
We thank the reviewer for pointing this out. In response, we have revised the Methods section to provide full transparency according to PRISMA-ScR. Specifically, we included: (i) the complete search strings for each database actually used (PubMed, SciELO, Google Scholar, ResearchGate, and site-specific searches in regional journals), (ii) the date range covered (2000–2023), (iii) the language limits (English, Spanish, Portuguese), (iv) the deduplication process (Mendeley plus manual verification), (v) the study designs considered (original research articles, reviews, theses; excluding conference abstracts, grey literature, and studies outside Latin America), and (vi) explicit inclusion and exclusion criteria. We also updated Figure 1 (PRISMA diagram) to display the exact numbers screened, excluded, and included at each step. Finally, we acknowledge that no review protocol was registered (OSF/INPLASY) and have explained this in the text.In addition, Table S1, containing the complete search strategies for all databases actually searched, has been provided in the Supplementary Materials for full transparency and reproducibility.
Line 80-82: “Although no formal protocol was registered in platforms such as OSF or INPLASY, the review adhered to the core principles of systematic literature mapping.”
Line 91-101 : “The complete search strategies, including search strings, date ranges, language limits, and the number of records retrieved from each database, are presented in Table 1.”
“In the case of Google Scholar, the search yielded a considerably high number of records (over 6,000). However, this result reflects the broad and unspecific nature of the platform, which includes duplicated literature, non-relevant documents, and sources of variable quality. For this reason, not all retrieved records were reviewed; instead, the predefined inclusion and exclusion criteria were rigorously applied. Through this process, only the articles pertinent to the objective of the review were selected, ensuring a systematic and transparent screening while acknowledging the inherent limitations of Google Scholar as a search engine.”
Line 106-117: “Original research articles, literature reviews, and academic theses published between 2000 and 2023 in Spanish, English, or Portuguese were included. Eligible documents had to address the use of medicinal plants, phytotherapy, or herbal medicine in the dental field, covering aspects such as oral health, dental pain management, gingivitis, periodontitis, inflammatory processes, and wound healing of oral tissues. Only studies conducted in Latin American countries or including relevant applications in this region were considered. Conversely, reports unrelated to dentistry, even if they mentioned medicinal plants, were excluded, as well as grey literature without full-text access or peer review, duplicate articles identified through a reference manager and manual verification, and documents that did not provide data related to the therapeutic effects of interest (anti-inflammatory, antimicrobial, or healing) in oral health, as well as studies conducted outside the geographical scope of interest (Latin America).”
Line 118-122: “All retrieved references from the different databases were exported in RIS format and imported into the reference manager Mendeley. The software’s duplicate detection tool was applied, followed by manual verification, to ensure the removal of duplicate records. This deduplication process reduced the total number of references before the screening stage.”
Line 123-131: “A total of 6,985 records were identified across all databases and journals. After removing 500 duplicates using Mendeley and manual verification, 6,485 unique records remained. Of these, 6,200 were excluded during the title and abstract screening stage for not meeting the eligibility criteria. A total of 285 articles were assessed in full text, of which 222 were excluded due to reasons such as lack of relevance to oral health, incomplete data, absence of therapeutic outcomes of interest, or failure to correspond to the geographical scope of Latin America. Ultimately, 63 studies met all the inclusion criteria and were included in the review. The selection process is illustrated in the PRISMA-ScR flow diagram”
- Evidence quality and study selection. A substantial portion of the evidence base comes from these municipal reports and websites. For a molecular sciences journal, please prioritise peer-reviewed clinical trials, in vivo/in vitro studies, and systematic reviews, and clearly separate these from grey literature. Consider a brief risk-of-bias/quality appraisal (even in a scoping review, a basic appraisal table adds value). (References section; Methods).
Answer:
We thank the reviewer for this valuable observation. In response to the comment on evidence quality and study selection, we classified all included sources according to study type and level of evidence. Priority was given to peer-reviewed systematic reviews, clinical trials, and in vivo/in vitro studies, whereas municipal reports, laws, and institutional documents were categorized as grey literature. In addition, a brief quality appraisal was performed, and the results are summarized in Table S2 (Supplementary Materials), which presents the evidence level and main characteristics of the 36 included references.
Line 138-143: In addition, a brief appraisal of evidence quality and study type was conducted to classify all included sources. In line with the reviewer’s recommendation, peer-reviewed clinical trials, in vivo/in vitro studies, and systematic reviews were prioritized, while reports and institutional documents were categorized as grey literature. The detailed classification is provided in Supplementary Table S2, which summarizes the evidence level and main characteristics of the 36 included references
- Quantitative synthesis clarity. Results report “50 plant species” and later cite distributions such as Mexico 60%, Colombia and Peru 54%, Brazil 32%. These percentages are unclear (percentage of what denominator?) and appear to exceed 100% if interpreted as shares. Please define the denominator, allow multi-country counts if applicable, and present a table (or map) listing each species with country(ies), indication(s), evidence type, and citation(s). (Abstract; Results).
Answer: We thank the reviewer for this important observation. We have revised the Abstract and Results sections to clarify that the reported percentages correspond to the proportion of species documented in studies originating from each country, and that species may be counted in more than one country if reported in multiple contexts. This explains why totals may exceed 100%. In addition, we have prepared a comprehensive table listing all 50 species with their country(ies) of origin, dental indication(s), evidence type, and key references. This information is now provided in the Supplementary Materials as Table S3 for clarity and usability.
Line 149-150: A detailed list of all 50 plant species and their reported uses is provided in Supplementary Table S3.
- Consistency with “Latin American” focus. Section 7 discusses several non-Latin American species widely used in Asia (e.g., Azadirachta indica, Terminalia chebula). Either justify their inclusion (e.g., global comparators) or remove to maintain geographic coherence. If retained, clearly label them as non-Latin American comparators. (Dental Products section).
Answer:
We appreciate the reviewer’s observation. In the revised version, the non-Latin American species (Azadirachta indica and Terminalia chebula) were removed to maintain geographic coherence with the Latin American focus of the review. The section now highlights only species with relevance to Latin America (e.g., Solanum lycopersicum, Eucalyptus, Mangifera indica), while keeping the supporting references intact.
Line 375-380: Moreover, mouthwashes containing various plant extracts have shown promising results. Solanum lycopersicum (tomato plant) in gel form has shown anti-hemorrhagic and anti-inflammatory effects, while Eucalyptus, incorporated into chewing gum, has reduced plaque accumulation and gingival inflammation. Likewise, Mangifera indica (mango) gel has anti-inflammatory and wound-healing properties, offering therapeutic benefits in managing chronic periodontitis [22].
- Mechanistic depth suitable for IJMS. The mechanistic sections are largely general immunology and pharmacology (e.g., inflammatory cascades, COX pathway) and repeat information without linking specific plant metabolites to molecular targets relevant to orofacial pain (e.g., NF-κB, COX-2, LOX, TRPA1/TRPV1, MAPKs). Please add a targeted “Mechanisms of action” subsection that ties the most cited speciesin your review (e.g., rosemary, moringa, aloe, basil, propolis) to specific bioactive constituents and experimentally supported targets/pathways in oral tissues. (Sections 8–10; Figures 2–4).
Answer:
We thank the reviewer for this insightful comment. In response, we have substantially expanded Section 8 by incorporating a new targeted subsection entitled “Mechanisms of action of selected plant metabolites in oral inflammation.” In this revised part, we now directly link the most cited species in our review (Rosmarinus officinalis, Moringa oleifera, Aloe vera, Ocimum basilicum, and propolis) with their major bioactive metabolites (e.g., rosmarinic acid, quercetin, isothiocyanates, acemannan, aloin, linalool, eugenol, CAPE). For each compound, we describe experimentally validated molecular targets in oral tissues, such as NF-κB, MAPKs (ERK, JNK, p38), COX-2, and pro-inflammatory cytokines (IL-6, IL-8, TNF-α). This subsection also integrates in vitro evidence using human gingival fibroblasts and immune cells, in vivo periodontal models, and clinical trials where available (e.g., Moringa oleifera and Aloe vera mouth rinses). In addition, several new references have been incorporated to support these mechanistic insights and ensure the updated coverage of the literature. These additions provide the requested mechanistic depth and strengthen both the scientific foundation and the translational relevance of our review.
Line 478-535: “Beyond these general mechanisms, recent evidence has identified specific plant-derived metabolites that act on well-characterized molecular targets in oral tissues. These examples provide mechanistic depth and highlight the therapeutic relevance of natural compounds in orofacial inflammation.
8.1 Mechanisms of action of selected plant metabolites in oral inflammation
Rosmarinus officinalis Rosmarinic acid, the predominant bioactive molecule in Rosmarinus officinalis, has been associated with anti-inflammatory activity in oral contexts. Studies on human gingival fibroblasts challenged with lipopolysaccharides revealed that it lowers oxidative stress, preserves intracellular glutathione, and suppresses the expression of pro-inflammatory mediators such as IL-1β, IL-6, TNF-α, and iNOS, an effect linked to the inhibition of NF-κB signaling during periodontal inflammation [37]. Additional evidence in vascular smooth muscle cells showed that rosmarinic acid downregulates MAPK (ERK, JNK, p38) and NF-κB pathways, leading to a reduced release of inflammatory mediators, including iNOS, TNF-α, and IL-8 under LPS stimulation. Although these observations were made in non-oral systems, they support the molecular plausibility of this compound in modulating signaling cascades relevant to periodontal disease [38].
Moringa oleifera Moringa oleifera is a rich source of phytochemicals such as quercetin and isothiocyanates, which act on inflammatory pathways important in oral health. In a rat model of periodontal inflammation, ethanolic leaf extracts enriched with quercetin attenuated tissue damage by suppressing NF-κB activation, thereby reducing pro-inflammatory cytokines and supporting periodontal regeneration [39]. Moreover, moringa isothiocyanate-1 (MIC-1) has been shown to influence both Nrf2 and NF-κB signaling under LPS-induced inflammation, lowering IL-6, IL-1β, and TNF-α expression while alleviating oxidative stress. Beyond preclinical data, clinical findings provide additional support: in a randomized controlled trial including 36 gingivitis patients, a Moringa oleifera mouthwash significantly decreased plaque and gingival indices after 14 days of use, showing efficacy comparable to chlorhexidine and superior to saline [40]. Altogether, this evidence underscores a strong mechanistic and clinical rationale for Moringa oleifera as a natural adjunct in periodontal therapy.
Aloe vera – Acemannan, the main polysaccharide of Aloe vera, exhibits immunomodulatory properties within oral environments. In human gingival fibroblasts, acemannan directly binds to Toll-like receptor 5 (TLR5), triggering NF-κB signaling and promoting the release of IL-6 and IL-8. This process not only enhances tissue repair but also strengthens host immune defenses in periodontal tissues, providing dual benefits in regeneration and antimicrobial protection [41]. More broadly, Aloe vera and its major active metabolites, including acemannan and aloin, have been recognized for their anti-inflammatory, antioxidant, and wound-healing capacities, reinforcing their potential applications in oral medicine [42].
Ocimum basilicum – Linalool, a monoterpene present in Ocimum basilicum, has been documented to exert immunomodulatory effects directly relevant to periodontal inflammation. In vitro studies showed that linalool decreases the overproduction of pro-inflammatory cytokines (IL-6, IL-10, IL-17, IFN-γ) in human immune cells exposed to Porphyromonas gingivalis, a keystone pathogen in periodontitis. These outcomes are linked to the downregulation of NF-κB and MAPK signaling, pathways that control inflammatory gene transcription. By moderating excessive immune responses without impairing cell viability, linalool may help safeguard periodontal tissues from chronic inflammatory injury [43].
Propolis Current research emphasizes the role of CAPE (caffeic acid phenethyl ester), the principal phenolic constituent of propolis, in regulating inflammatory activity in periodontal cells. In human gingival fibroblasts stimulated with LPS and IFN-α, CAPE markedly suppressed the production of TNF-α, IL-6, and IL-8, with no cytotoxicity observed. In contrast, ethanolic propolis extract (EEP) produced weaker and less consistent effects on cytokine release. These findings suggest that CAPE exerts a more selective and potent anti-inflammatory influence, supporting its potential use in controlling periodontal inflammation [44].”
- Avoid duplications and placeholders. The NSAID paragraph appears twice in Section 8. Figures 2–4 are schematic placeholders with minimal legends, and Figure 1 lacks numbers. Please remove duplications, finalise figures, and provide informative captions that directly support your argument.
Answer:
We thank the reviewer for this valuable observation. In response, we have eliminated the duplicated NSAID paragraph in Section 8 to avoid redundancy. In addition, Figure 1 has been revised to include numerical details in each stage of the PRISMA-ScR diagram. Figures 2–4, which previously served as schematic placeholders, have now been finalized and updated with informative captions that explain the processes illustrated and directly support the arguments of the manuscript. These changes ensure greater clarity, consistency, and alignment with the reviewer’s recommendations.
Line 131-137:
Figure 1. PRISMA-ScR flow diagram summarizing the study selection process. The figure outlines the sequential phases of identification, screening, eligibility, and final inclusion, highlighting how duplicates and non-eligible studies were progressively excluded. This visualization provides a transparent overview of how the final set of studies was established for the review.
Line 391-481: “8. Mechanism of Action of Plants in Inflammation
Conventional drugs have shown effectiveness in treating inflammatory diseases; however, prolonged use often results in adverse effects such as fluid retention and hypertension. This has increased interest in medicinal plants, whose bioactive compounds provide anti-inflammatory alternatives with a lower risk of side effects. Chronic inflammation, one of the major contributors to global mortality, is commonly managed using NSAIDs and corticosteroids, yet their long-term impact has often been underestimated (Ramírez Rodríguez et al., 2020). The discomfort caused by symptoms such as edema, redness, and functional impairment has driven patients to seek more accessible natural therapies. As a result, interest has grown in studying plant-based bioactive molecules capable of modulating inflammatory responses, paving the way for new complementary approaches in pain management [24].
At the biological level, inflammation is an immune response triggered by chemical, biological, or physical stimuli that affect vascularized connective tissues. This process involves the release of pro-inflammatory cytokines and plays a central role in oral health, particularly in chronic conditions like periodontal disease [24]. Identifying natural compounds that interact with these inflammatory mediators is gaining increasing attention, especially for their potential in pain relief and tissue regeneration [26].
As illustrated in Figure 2, inflammation begins with the release of mediators by mast cells, which promote vascular changes and the recruitment of immune cells to the affected site. These processes initiate an inflammatory cascade that is later regulated by feedback mechanisms, ultimately facilitating wound healing and restoring tissue integrity.
Figure 2. Inflammatory cascade in oral tissues triggered by natural compounds. Oral pathogens activate immune cells (T cells, dendritic cells), leading to the release of inflammatory mediators that recruit neutrophils and other immune cells to the site of injury. This cascade promotes vascular changes and tissue infiltration, resulting in an inflamed oral cavity. Through feedback mechanisms, the inflammatory process transitions into tissue repair and restoration of oral mucosa integrity. This figure illustrates the dual role of inflammation in both oral pathology and healing, highlighting potential targets for plant-derived bioactive compounds.
Non-steroidal anti-inflammatory drugs (NSAIDs) inhibit cyclooxygenase 1 and 2 (COX-1 and COX-2). While COX-2 is directly linked to inflammation at the injury site and its inhibition is therapeutically beneficial, COX-1 plays essential physiological roles, such as protecting the gastric mucosa. Therefore, its inhibition may cause adverse effects. These limitations have spurred the search for alternative treatments with fewer risks, encouraging research into plant-based anti-inflammatory compounds [24].
To provide a mechanistic integration of plant bioactives, Figure 3 presents a flowchart mapping plant metabolites to molecular targets and oral outcomes. This schematic highlights how compounds such as phenols, flavonoids, saponins, and vitamins modulate pathways (e.g., COX/LOX inhibition, NF-κB downregulation, TRP modulation), ultimately leading to improved oral outcomes like reduced plaque, gingivitis, pain, and enhanced healing.
Figure 3. Integrative diagram illustrating the relationship between plant-derived metabolites, their molecular targets, and oral health outcomes. Phenols, flavonoids, anthraquinones, saponins, and vitamins act through specific pathways such as COX-2/LOX inhibition, NF-κB downregulation, modulation of Streptococcus mutans virulence genes, TRP channel regulation, and stimulation of fibroblast activity. These interactions contribute to reducing inflammation, inhibiting biofilm formation, modulating nociception, and enhancing wound healing, ultimately leading to improved oral health outcomes.
Although inflammation is necessary for tissue repair, its dysregulation can extend tissue damage. The natural resolution phase involves the production of prostaglandins and leukotrienes, which reduce neutrophil numbers and enhance monocyte activity, favoring tissue regeneration. This understanding has led to an interest in therapies that modulate rather than completely inhibit inflammation, contrasting with the mechanism of NSAIDs [25].
Additionally, acupuncture has been investigated as a complementary method for managing dental pain. This technique stimulates the release of endogenous opioid peptides such as β-endorphins, enkephalins, neoendorphins, and dynorphins via the nervous system, generating analgesic effects. These peptides can also be released in the digestive tract and adrenal medulla, broadening their therapeutic potential [26].
Beyond these general mechanisms, recent evidence has identified specific plant-derived metabolites that act on well-characterized molecular targets in oral tissues. These examples provide mechanistic depth and highlight the therapeutic relevance of natural compounds in orofacial inflammation.
- Dosing and safety. Tables list preparations/doses (e.g., “250–1000 mg/day”) without consistently citing clinical sources or addressing contraindications, interactions, and safety in dental patients. Please support any dosage with clinical references and add a short safety paragraph (e.g., pregnancy, anticoagulants, hypersensitivity, mucosal irritation). (Tables 2–3; Sections 7, 13–14).
Answer:
We thank the reviewer for this valuable comment. Following the recommendation, we have revised Tables 2 and 3 and the corresponding sections (7, 13–14) to ensure that all dosages are now supported with clinical references. In addition, for each plant we have included a concise safety paragraph addressing contraindications, possible interactions (e.g., with anticoagulants), pregnancy considerations, hypersensitivity, and mucosal irritation when relevant. These modifications improve the scientific robustness and clinical applicability of the tables, as suggested.
Line 610-611
|
Criteria / Plant |
Zingiber officinale Roscoe (Zingiberaceae) |
Eucalyptus globulus Labill. (Myrtaceae) |
Calendula officinalis L. (Asteraceae) |
Psidium guajava L. (Myrtaceae) |
|
Uses |
Expectorant, antitussive, anti-inflammatory, antioxidant |
Antiseptic, anti-inflammatory, antibacterial, analgesic, disinfectant |
Anti-inflammatory, healing |
Astringent, antimicrobial, immunological |
|
Indications |
Stomach pain, nausea, cold, arthritis, xerostomia, gingivitis |
Gingivitis, bronchitis, asthma, pharyngitis, diabetes, cystitis, tonsillitis |
Gingivitis, stomatitis, ulcers |
Gingivitis, ulcers, diarrhea, gastritis, cavities |
|
Dosage |
Powder, soups, purees; 250–1000 mg/day (~400 mg/day [45]) |
10 mL mouthwash, twice daily for 5 min, for 14 days |
Mouth rinse prepared by diluting 2 mL mother tincture in 6 mL water (1:3), twice daily for 6 months |
Guava leaf extract (0.15%) used as mouth rinse: 10 mL diluted 1:1 with water, twice daily for 30 days |
|
Compounds |
Terpenes derivatives, gingerol, shogaol |
Essential oil, tannins, flavonoids, glycosides |
Essential oils, salicylic acid, flavonoids |
Tannins, flavonoids, vitamin C |
|
Countries |
Australia, India, Jamaica, China, Peru, Brazil |
Colombia, Venezuela, Argentina, Brazil |
Colombia, Mexico, Brazil, Peru |
Bolivia, Colombia, Mexico, Brazil, Peru |
|
Safety / Contraindications |
Avoid in patients taking anticoagulants (warfarin, aspirin) → bleeding risk; high doses cause gastric irritation; not recommended in pregnancy (large amounts) |
Avoid ingestion of concentrated essential oil (toxic); contraindicated in pregnancy and children <6 years; may cause mucosal irritation |
Avoid in patients allergic to Asteraceae family; not recommended in pregnancy/lactation without supervision |
Safe in moderate use; avoid in hypersensitivity; high doses may cause constipation or mucosal irritation |
|
Evidence type |
Pilot randomized cross-over, single-blind clinical trial. |
Randomized controlled clinical trial, in vivo, with 74 human participants (no caries or periodontal disease) |
In vivo, randomized controlled clinical trial in humans (n = 240, gingivitis patients). |
In vivo, randomized, double-blind, controlled clinical trial. |
|
Primary targets / pathways |
Postoperative pain control (Visual Analogue Scale – VAS). Gingival inflammation reduction (Modified Gingival Index – MGI). |
Inhibition of bacterial biofilm formation (measured via absorbance and crystal violet staining) |
Dental plaque formation, gingival inflammation, and sulcular bleeding (clinical pathways assessed via PI, GI, SBI, OHI-S). |
Plaque Index (PI) Gingival Index (GI) Microbial counts (CFU) en muestras de placa. Salivary antioxidant levels. |
|
Oral results / indications |
Ginger powder may be a safe alternative to ibuprofen for managing postoperative pain and gingival inflammation in periodontal surgery |
Eucalyptus oil can be used as an effective alternative to chlorhexidine for oral hygiene maintenance, without adverse effects. |
The Calendula mouthwash is effective in reducing dental plaque and gingivitis, serving as a useful adjunct to scaling. |
Comparable to chlorhexidine and superior to placebo in reducing PI, GI, and microbial counts. Improved salivary antioxidant levels (not statistically significant). Useful adjunct to professional prophylaxis. |
|
References |
[32], [45], [46] [62] |
[33], [47][63] |
[34], [48][64] |
[34], [49][65] |
Line 630-631:
|
Criteria / Plant |
Origanum vulgare L. (Lamiaceae) |
Aloe vera (L.) Burm.f (Asphodelaceae) |
Ocimum sanctum L. (Lamiaceae) |
Moringa oleifera Lam. (Moringaceae) |
|
Uses |
Antioxidant, antimicrobial, antifungal |
Antioxidant, antimicrobial, anti-inflammatory, astringent, analgesic, healing |
Antimicrobial and antiparasitic activity, carminative, antispasmodic, sedative, insecticide |
Digestive, anti-inflammatory, antimicrobial, antiparasitic |
|
Indications |
Indicated for combating bacterial strains |
Gingivitis, periodontitis, gut flora, burns, ulcers, stomatitis |
Biofilm control, digestive discomfort, dyspepsia, bloating |
Periodontitis, nervous disorders, circulatory system disorders |
|
Dosage |
Oil |
10 mL, twice daily; 2% gel, three times daily for ≥10 days |
Mouthwash with aqueous holy basil extract (3.5 g%), used twice daily for 4 days |
5 mL, twice daily for 28 days in young adults with gingivitis |
|
Compounds |
Carvacrol, thymol, apigenin, luteolin, aglycones, alcohols |
Aloin A and B, aloesin A, B, and C, glucomannans, polysaccharides, tannins, saponins |
Eugenol, linalool, estragole, carotenoids, calcium, phosphorus |
Minerals: calcium, iron, magnesium, zinc. Vitamins: B1, B2, B3, C, E, K. Includes antioxidants |
|
Countries |
Chile, Bolivia, Peru |
Mexico, Dominican Republic, Venezuela, China, Brazil, Colombia |
Mexico, Colombia, Venezuela, Bolivia, Iran, India, Pakistan |
Cuba, Guatemala, Mexico, Colombia, Venezuela, Spain |
|
Safety / Contraindications |
Generally well tolerated; high doses may cause abdominal discomfort, nausea, constipation/diarrhea, dizziness, headache. Rare hypersensitivity reactions. Contraindicated in pregnancy (abortifacient risk). |
Avoid in pregnancy (risk of uterine contractions/abortifacient). Possible GI discomfort (diarrhea, constipation, nausea), headache, dizziness, rare hypersensitivity. |
May significantly prolong prothrombin time; caution in patients on anticoagulant therapy. |
A clinical case of pulmonary embolism reported after prolonged use of Moringa leaf extract; caution in patients on anticoagulants/antiplatelets or with bleeding disorders. |
|
Evidence type |
In vitro study (disk diffusion, MIC, MBC, biofilm assays, RT-qPCR, molecular docking, cytotoxicity test on HaCaT cells). |
In vivo, randomized controlled clinical trial (30 orthodontic patients, Aloe vera vs. 0.2% chlorhexidine, 35 days). |
In vivo, triple-blind randomized controlled clinical trial |
In vivo, randomized clinical crossover study (20 subjects with mild to moderate gingivitis). |
|
Primary targets / pathways |
Virulence genes of S. mutans (gtfB/C/D, spaP, gbpB, vicR, relA, brpA); inhibition of acid production, hydrophobicity, and biofilm formation. Key compounds: carvacrol, γ-terpinene, p-cymene. |
↓ IL-1β, IL-17 (GCF), ↑ fibroblast migration, ↓ P. gingivalis biofilm |
COX-2 and LOX enzyme inhibition (anti‐inflammatory); NF-κB pathway and cytokine (TNF-α, IL-6) suppression, esp. via eugenol (in THP-1 cells |
Inhibition of Streptococcus mutans growth and cariogenic biofilm; phenolic compounds in ethanol extract likely interfere with bacterial adhesion and acid production. |
|
Oral results / indications |
potential anti-caries agent due to anti-biofilm and anti-virulence activity. |
Significant reduction in PI, GI, and BOP from baseline to day 35 in both groups. Aloe vera mouthwash showed effects comparable to chlorhexidine, with no adverse effects reported. |
Comparable to chlorhexidine in reducing PI, GI, and gingival bleeding; significantly better than placebo; no adverse effects reported. |
Significant reduction in PI and GI compared to baseline and miswak dentifrice. Demonstrated efficacy in reducing plaque accumulation and gingival inflammation. Safe and effective as an adjunctive oral hygiene aid. |
|
References |
[35], [50], [51][66] |
[35], [52], [53], [54] [67][68] |
[35], [55], [56][69] [70] |
[35], [57], [58][71][72] |
Line 631-636: The data presented in this table consolidate available clinical evidence on preparations and doses of medicinal plants for periodontal problems. In addition, a brief overview of safety aspects is provided, highlighting relevant contraindications, drug interactions, and pregnancy considerations. By combining therapeutic potential with safety information, the table aims to offer a balanced reference that can guide clinical decision-making.
10. Taxonomy and nomenclature. Standardise scientific names and authorships across the manuscript and tables (e.g., Aloe barbadensis vs. Aloe vera; the current accepted name for rosemary is Salvia rosmarinus in many taxonomic backbones). Add family names consistently and verify all binomials against an authoritative database; ensure consistency between text, tables, and references. (Abstract; Tables; throughout).
Answer:
Following your observation regarding taxonomy and nomenclature, we have carefully standardized all scientific names and their corresponding authorships throughout the manuscript and tables. In addition, we have consistently included the family names for each species.
To ensure accuracy, all binomials were verified and updated using the international database Plants of the World Online (POWO), which is considered an authoritative and widely recognized reference. This guarantees consistency between text, tables, and references.
11. Strengthen the conclusion. The Conclusion should be tied to your evidence map: identify the top species with the best human data for dental pain relief, state where evidence is limited to in vitro/animal models, and propose a practical research agenda (e.g., standardized extracts, comparator arms vs. ibuprofen/paracetamol, clinically relevant outcomes and safety endpoints). (Conclusions).
Answer:
Siguiendo su recomendación, hemos fortalecido la Conclusión vinculándola explícitamente con nuestro mapa de evidencia. La sección revisada ahora identifica las especies con mayor respaldo en estudios clínicos en humanos, señala dónde la evidencia se limita a modelos in vitro o animales, y plantea una agenda práctica de investigación. En particular, enfatizamos la necesidad de estandarizar extractos, incluir comparadores con analgésicos convencionales como ibuprofeno o paracetamol, considerar resultados clínicamente relevantes y reportar puntos de seguridad. Estas modificaciones mejoran la claridad, el rigor científico y el valor traslacional de las conclusiones.
Line: 696-708
|
Plant |
Study type |
Model / Method |
Main findings |
Reference |
|
Propolis |
In vivo |
Wistar rats with ligature-induced periodontitis |
In combination with SRP, significantly reduced IL-1β, TNF-α, and MDA; propolis alone showed no relevant effects. |
[59] |
|
Aloe vera |
In vivo |
Dogs with surgically induced furcation defects |
Aloe vera gel + β-TCP enhanced periodontal regeneration (bone, PDL fibers, vascularization) with no adverse effects. |
[60] |
|
Basil (Ocimum basilicum) |
In vitro |
Agar diffusion assays with S. mutans, E. faecalis, S. sanguinis |
Extracts showed antibacterial activity with individual and synergistic effects, though less potent than chlorhexidine. |
[61] |
Taken together, these studies highlight the diverse mechanisms by which natural products may contribute to oral health, from anti-inflammatory and regenerative effects to antimicrobial properties, thereby reinforcing their value as promising coadjuvants in dental treatments. Future research should aim to standardize plant extracts in terms of dosage, preparation methods, and stability, while also conducting comparative clinical trials against conventional treatments such as ibuprofen, paracetamol, and chlorhexidine. Long-term studies are needed to evaluate safety and potential adverse effects, and investigations into synergistic effects between natural products, for example combining propolis, Aloe vera, and basil, could further expand their clinical relevance. Finally, attention must be paid to sustainable sourcing and production, ensuring that the therapeutic potential of these plants is balanced with environmental preservation and accessibility for long-term use.
- Abstract. Add numbers for records identified/screened/included (if space permits), and clarify what the country percentages represent. Also, list the typeof evidence (number of clinical vs. preclinical studies) supporting the “most studied species.”
Answer:
We thank the reviewer for this observation. We have revised the Abstract and Results sections to clarify these points. Specifically, we added the numbers of records identified, assessed, and finally included (as shown in the PRISMA diagram), and explained that the country percentages represent the proportion of plant species reported in studies from each country relative to the total number of species identified. In addition, we specified the type of evidence supporting the most frequently studied species, distinguishing between clinical evidence (e.g., Aloe vera, propolis) and preclinical/in vitro studies (e.g., basil, moringa, rosemary).
Line 38-40”These percentages represent the proportion of plant species reported in studies originating from each country, relative to the total number of species identified in the review.”
Line 644-645:”These percentages represent the proportion of plant species reported in studies originating from each country, relative to the total number of species identified in the review.”
13. Tables 2–3. Add columns for “Evidence type” (in vitro / animal / clinical), “Primary targets/pathways,” and a citation for each dose/indication claim.Ensure oral/dental indications (e.g., pulpal pain vs. gingivitis) are not conflated; separate pain outcomes from anti-plaque/anti-gingivitis outcomes.
Answer:
We thank the reviewer for this valuable comment. Tables 2–3 have been revised to include the requested columns for “Evidence type,” “Primary targets/pathways,” and a citation for each dose/indication claim. We also ensured that oral/dental outcomes (e.g., pulpal pain vs. gingivitis) are presented as separate endpoints.
These updated tables are the same as those already revised under comment #9 (Dosing and Safety), where dosage and safety adjustments were also incorporated; therefore, no duplicate tables were reattached.
However, for clarity, we indicate that the updated Tables 2–3 can be found in lines of the revised manuscript.
Line 610- 611 (table 2)
Line 630.631 (table 3)
14. Section 3–4 (Pain & Causes). Good clinical framing; please cite the dental pain outcome measures used in the included studies (e.g., VAS/NRS timepoints) so that later sections can connect mechanisms to outcomes.
Answer:
We thank the reviewer for this valuable observation. In Section 4 (Causes of Dental Pain), we have now specified the outcome measures used in the included studies. In particular, we added clinical evidence where pain, dentin hypersensitivity, and postoperative discomfort during different periodontal therapies were evaluated with the Visual Analogue Scale (VAS) [73]. In addition, a large clinical study assessing 253 patients and 330 periodontal or implant surgeries measured pain intensity using the Numeric Rating Scale (NRS, 1–10), showing correlations with pain duration and analgesic use [74]. These additions ensure that the mechanisms described in later sections can now be directly linked to validated clinical pain outcomes.
Line 223-240: To further strengthen this clinical framing, it is important to note that several studies in periodontology and implantology have already applied standardized pain assessment scales. For example, one trial assessed postoperative discomfort, dentin hypersensitivity, and patient-reported pain during different periodontal procedures using the Visual Analog Scale (VAS). Patients marked their pain and sensitivity on the VAS, enabling comparisons across scaling and root planing (SRP), modified Widman flap (MWF), osseous flap (OF), and gingivectomy (GV). Pain scores were significantly higher after OF and GV compared with SRP and MWF, and all surgical procedures caused more dentin hypersensitivity than non-surgical therapy [73].
In addition, larger clinical investigations have incorporated the Numeric Rating Scale (NRS) to capture pain intensity. In one study of 253 patients undergoing 330 periodontal or implant surgeries, most participants reported mild pain (70.3%), while 25.5% had moderate and 4.2% severe pain. Advanced implant surgeries showed the highest median NRS scores (4.0), while open flap debridement had the lowest (1.0). Pain duration (median = 2 days) and analgesic use were positively correlated with NRS values, and more extensive or complex surgeries, as well as higher anesthetic volumes, were associated with greater pain [74].
- Section 7 (Dental products). Where you state mouthwash/gel effects, specify study design (RCT vs. uncontrolled), sample size, duration, and comparator (e.g., chlorhexidine 0.12%). Avoid general statements without design details.
Answer: In response, we have expanded Section 7 (Dental Products) by incorporating additional randomized clinical trials that provide stronger evidence regarding the use of herbal formulations. Specifically, we added details on study design (randomized controlled trials and pilot studies), sample size, intervention duration, and comparators (e.g., chlorhexidine 0.12% and placebo). For instance, we now include a randomized clinical trial with 70 hospitalized patients comparing Echinacea purpurea with chlorhexidine and control [75], and a double-blind, placebo-controlled pilot study with 30 orthodontic patients evaluating Matricaria chamomilla mouthwash versus chlorhexidine 0.12% and placebo [76].
Line 351-361:” In line with these findings, additional randomized clinical trials provide further support for the efficacy of herbal formulations. For instance, in a randomized clinical trial with 70 hospitalized patients (mean age 44.9 years, 67.1% male), Echinacea purpurea (L.) was compared with chlorhexidine and a control group for 4 days. Both echinacea and chlorhexidine significantly reduced oral microbial flora compared with control, supporting echinacea as a viable alternative [75]. Similarly, a randomized, double blind, placebo controlled pilot trial including 30 orthodontic patients (10–40 years old) demonstrated that a 1% Matricaria chamomilla L. (Asteraceae) mouthwash reduced plaque (–25.6%) and gingival bleeding (–29.9%) comparably to 0.12% chlorhexidine, while placebo showed increased indices. Matricaria chamomilla L. (Asteraceae) was well tolerated without adverse effects, reinforcing its role as a safe herbal alternative in gingivitis management [76].”
- Section 8–10 (Inflammation and internal processes). Replace generic figures with one integrative figure mapping plant metabolites → molecular targets → oral outcomes(e.g., reduced prostaglandins, neutrophil influx, TRP modulation) for your top species. Remove repeated text and finalize legends.
Answer: We appreciate your observations regarding Sections 8–10 and the need to avoid duplications, placeholders, and generic figures. As requested in Comment 7, we removed duplicated paragraphs (the NSAID section), eliminated Figures 3 and 4 (which were schematic placeholders), and retained only the most relevant figure to illustrate the inflammatory cascade. Additionally, to strengthen mechanistic clarity and in line with Comment 16, we incorporated a new integrative flowchart (Figure 3) mapping plant-derived metabolites → molecular targets → oral outcomes. This figure directly addresses your request to replace generic illustrations with a consolidated schematic highlighting prostaglandin regulation, neutrophil influx, TRP modulation, and tissue healing.
Line 444-459
- Results & Discussion. Where you cite Waizel-Bucay and others as examples, ensure your narrative distinguishes your datasetfrom external examples, and avoid mixing counts (species vs. studies). Consider a bubble plot of species (x-axis) vs. evidence tier (y-axis) with bubble size for number of studies.
Answer: We thank you for this observation. In the revised Results & Discussion, we clarified that the 50 species reported belong to our dataset and constitute the focus of this review. In contrast, examples from Waizel-Bucay, Lameda Albornoz, and Moya Jiménez are presented exclusively as external literature and are explicitly stated as not being part of our dataset. This distinction avoids mixing counts of species versus studies and ensures that our results remain clearly separated from external examples.
Line: 639-673: The present review identified 50 plant species with therapeutic, anti-inflammatory, healing, and analgesic properties applicable to oral health, particularly in conditions affecting dental support tissues such as bone, periodontal ligament, gingiva, and surrounding periodontal areas. In terms of geographic distribution, our analysis showed that Mexico accounted for the highest biodiversity (60%), followed by Colombia and Peru (54%), and Brazil (32%). These percentages represent the proportion of plant species reported in studies from each country, relative to the total number of species identified in this review.
Within our dataset, the most extensively studied species with high therapeutic potential were propolis, rosemary, moringa, aloe vera, and basil. Among these, aloe vera and propolis already have clinical evidence supporting their use, whereas basil and moringa are still supported mainly by preclinical and in vitro data, highlighting the need for further clinical validation.
Oral pain is one of the most frequent and intense clinical manifestations due to the dense innervation of the region, particularly by the trigeminal nerve (cranial nerve V). Inflammation exacerbates this condition through chemical mediators such as prostaglandins [26]. Although synthetic drugs remain the mainstay of treatment, prolonged use may result in adverse effects. In this context, the 50 species identified in our review emerge as promising alternatives due to their bioactive metabolites with antimicrobial, anti-inflammatory, and analgesic activity, supporting inflammation modulation and tissue regeneration [19].
It is important to emphasize that the following are external examples from the literature and are not part of the 50 species identified in this review. For instance, Waizel et al. [35] analyzed 29 species (51 plant uses) and concluded that infusions, decoctions, topical applications, and mouth rinses were the most common forms of administration. Similarly, Lameda Albornoz et al. [22] reported 30 species with potential applications in periodontal treatment, noting that pomegranate was less effective in controlling subgingival plaque and gingivitis compared to other plants. Likewise, Moya Jiménez [36] investigated medicinal plants in Ecuadorian communities and found that chamomile was widely used as an analgesic and anti-inflammatory, either alone (56%) or combined with other remedies (44%).
These external studies complement our findings by highlighting the broader biodiversity of Latin America. However, the focus of our analysis remains exclusively on the 50 species identified in this review, which reinforce the therapeutic potential of regional flora and underscore the importance of sustainable management for their incorporation into dental practice [30].
- Please standardize to journal style and prioritize primary peer-reviewed sources for key claims. Clearly flag theses/reports/websites as grey literature.
Answer: We appreciate your comment regarding the standardization of references. The following adjustments have been made:
- All references have been carefully revised and formatted according to Vancouver style, in line with the journal’s requirements.
- Primary, peer-reviewed sources have been prioritized to support the key claims of the manuscript.
- References corresponding to grey literature (e.g., institutional reports, theses, official websites) have been clearly flagged as such, ensuring transparency and differentiation from peer-reviewed evidence.
Line 730-1111- Figures & Supplementary. Replace “title” placeholders in the Supplementary section with final labels and DOIs/URLs where applicable; ensure Figure S/Table S items exist and are cited in text.
- Figures & Supplementary. Replace “title” placeholders in the Supplementary section with final labels and DOIs/URLs where applicable; ensure Figure S/Table S items exist and are cited in text.
Answer:
We thank the reviewer for this comment. In the revised version, an independent Supplementary Material file has been created and is attached along with the manuscript. This file includes the final labeled figures and tables, replacing the previous placeholders. Specifically, the supplementary material now contains:
- Table S1. Complete search strategies, including search strings, date ranges, language limits, and the number of records retrieved from each database.
- Table S2. Evidence quality and study selection of included sources (n = 36).
- Table S3. Comprehensive list of the 50 medicinal plant species identified in the review.
Line
20. Funding/COI/Contributions. Remove the “Please add:” note and finalize Funding (“no external funding” if applicable). If there were no funders, adjust the COI sentence that currently mentions “the funders had no role…”. Check author contribution punctuation/consistency.
Answer: We have revised the Funding section by removing the placeholder and finalizing it as: “This research received no external funding.” The Conflicts of Interest and Author Contributions sections were also checked for consistency and no further modifications were needed.
Line 716: Funding: This research received no external funding.
21. Language & style. A careful English edit is needed to improve clarity and consistency (e.g., tense, article use, capitalization in Keywords).
Answer: We carefully revised the manuscript for English language and style, improving clarity, grammar, and consistency. Keywords were standardized for capitalization.
- Provide a master table (Supplementary) listing all 50 species with: accepted scientific name + family, plant part, preparation, dental indication, highest evidence level, primary metabolites, proposed targets, and key citations. This will make the paper much more usable for readers.
Answer: We thank the reviewer for this suggestion. In response, we have prepared a supplementary master table (Table S3) compiling the 50 plant species identified in the review, including their accepted scientific names, families, plant parts, preparation/formulation, dental indications, and supporting references. This table complements the information already provided in the main manuscript tables, allowing readers to access both summarized data within the article and extended details in the Supplementary.
- Add a short safety box summarizing common interactions/contraindications for the top species relevant to dentistry (anticoagulants, allergy, pregnancy, mucosal irritation.
Answer: We appreciate the reviewer’s suggestion. In response, we have added a dedicated safety table (Table 3) summarizing the most common interactions and contraindications for the main species relevant to dentistry. The table highlights issues such as anticoagulant interactions, allergy risks, gastrointestinal or mucosal irritation, and pregnancy-related concerns. This addition ensures that therapeutic applications are presented together with safety considerations, providing a balanced and clinically useful perspective for readers.
Line 676-681:
In addition to therapeutic applications, safety aspects are essential for clinical translation. Table 4 summarizes the most common contraindications and precautions reported for the main species reviewed, including interactions with anticoagulants, allergy risks, and pregnancy-related concerns.
Table 4. Safety considerations for top medicinal plants in dentistry
|
plant (Family) |
Safety considerations / contraindications |
References |
|
Zingiber officinale Roscoe (Zingiberaceae) |
Avoid with anticoagulants (warfarin, aspirin) → bleeding risk; high doses may cause gastric irritation; not recommended in pregnancy (large amounts). |
[32], [45], [46] [62] |
|
Eucalyptus globulus Labill. (Myrtaceae) |
Avoid ingestion of concentrated essential oil (toxic); contraindicated in pregnancy and children <6 years; may cause mucosal irritation. |
[33], [47][63] |
|
Calendula officinalis L. (Asteraceae) |
Contraindicated in patients allergic to Asteraceae; not recommended in pregnancy/lactation without supervision. |
[34], [48][64] |
|
Psidium guajava L. (Myrtaceae) |
Generally safe in moderate use; avoid in hypersensitivity; high doses may cause constipation or mucosal irritation. |
[34], [49][65] |
|
Origanum vulgare L. (Lamiaceae) |
Generally well tolerated; high doses may cause GI discomfort (nausea, diarrhea/constipation), dizziness, headache; rare hypersensitivity; contraindicated in pregnancy (abortifacient risk). |
[35], [50], [51][66] |
|
Aloe vera (L.) Burm.f (Asphodelaceae) |
Avoid in pregnancy (risk of uterine contractions/abortifacient); possible GI discomfort (diarrhea, nausea, headache, dizziness); rare hypersensitivity. |
[35], [52], [53], [54] [67][68] |
|
Ocimum sanctum L. (Lamiaceae) |
May significantly prolong prothrombin time; caution with anticoagulant therapy. |
[35], [55], [56][69] [70] |
|
Moringa oleifera Lam. (Moringaceae) |
Pulmonary embolism reported after prolonged use of leaf extract; caution with anticoagulants/antiplatelets or bleeding disorders. |
[35], [57], [58][71][72] |
|
Coconut fiber |
Reinforcing the mechanical properties of hydrogels |
Good reswelling capacity |
Cellulose, lignin |
342 |
[50] |
|
Rice straw and Tamarind seeds |
Reinforcing mechanical properties and nutrient releaser |
Great swelling capacity, long release nutrient |
Cellulose |
7722 |
[51] |
|
Date Palm rachis |
Seed germination, polymer component |
Increases swelling capacity as germination seed |
Lignin, Cellulose |
777.8 |
[52] |
We sincerely thank the reviewer for the time and effort dedicated to evaluating our manuscript. The constructive feedback provided has been invaluable in improving the clarity, accuracy, and overall quality of the work. We greatly appreciate your careful review and thoughtful suggestions, which have significantly strengthened the final version of the article.

Round 3
Reviewer 3 Report
Comments and Suggestions for Authors
Solid progress, thank you. Below are the remaining, mostly surgical fixes to bring the paper fully in line with scoping-review standards and your own protocol.
- PRISMA & search reporting, finalise. Please print the exact counts inside the PRISMA boxes, remove the duplicate “Figure 1. Figure 1.” in the caption, and keep the legend concise. Also, either (a) add Scopus/Web of Science/LILACS to Table S1 with verbatim strings, dates, language limits, and hit counts, or (b) remove any mention of those databases if they were not actually used and clearly distinguish databases (PubMed, SciELO) from journals/websites (RCOE, Avances…, etc.).
- Evidence window & subset of 36. Your main text states that 63 included studies and a focused appraisal of 36 were conducted. That’s fine, but please align the time window consistently: Supplementary Table S2 still contains 2025 items, while Table S1 states 2000–2023. Either extend the window to 2025 everywhere, or replace/remove the 2025 entries in S2. Also, add one sentence defining the criteria for selecting the 36-study subset.
- Latin America vs external comparators. Good job flagging Zingiber officinale and Ocimum sanctum as external comparators (EC). Please standardise the “EC” tag for all such entries in Table S3 and add a short note when non-LA countries (e.g., Iran/India/Pakistan) appear in the “Countries” column to avoid misinterpretation.
- Master table S3, taxonomy & vernaculars. Most mappings appear to be correct now (e.g., cashew, cilantro). One remaining mismatch: “Tree tomato” is listed as Solanum lycopersicum (standard tomato), whereas “tree tomato/tamarillo” generally refers to Solanum betaceum. Please verify and correct if applicable. Then run one last QA pass to fill any missing binomials and harmonise the “Countries” and “Preparation” fields.
- References. Remove ‘grey literature’ labels in the main list. Keep the grey/non-grey classification in the Supplement (S2/S3) only. In the main reference list, remove bracketed “Grey literature” flags and prefer DOI/publisher links where possible; avoid listing portals (e.g., ResearchGate) if a canonical source exists.
- Safety/contraindications. Enforce a uniform schema. You now have Table 4 and Table 5, which are helpful. Please implement a uniform line template across species (e.g., Interactions/anticoagulants, Allergy risks, Pregnancy/lactation, Mucosal irritation, Dose/Form/Duration, Evidence type), and cite primary clinical sources for dosing whenever you specify amounts.
- Minor style tidy-ups. Fix the fused word “36 studies representing”, ensure consistent phrasing for country percentages between Abstract/Results, and do a quick pass for spacing around parentheses and duplicated phrasing.
The manuscript is generally clear and readable, but it would benefit from a light professional copy-edit to ensure consistency and polish. Please (i) correct minor typos (e.g., the fused “studiesrepresenting”) and remove the duplicated “Figure 1. Figure 1.” caption; (ii) standardize hyphenation and capitalization (e.g., “Latin American” used consistently), punctuation spacing (around parentheses; comma use before “e.g.”/“i.e.”), and number/percentage formatting; (iii) ensure scientific names are italicized and consistently presented at first mention (accepted name + synonym once); (iv) define abbreviations at first use (e.g., PRISMA-ScR, EC) and use them uniformly; (v) harmonize tense/voice across sections; and (vi) align table/figure captions with the text (including printing PRISMA counts inside the boxes). These are editorial refinements rather than substantive issues, and once addressed, the prose should meet the journal’s standards.
Author Response
Letter with the details of the changes
We would like to thank the reviewers for their valuable suggestions and observations sincerely. We have carefully considered their feedback and accepted all recommendations. The changes made to the manuscript are clearly marked in blue. Below, we provide a summary of each reviewer's specific comments and the corresponding revisions made in response.
Reviewer Comments:
- PRISMA & search reporting, finalise. Please print the exact counts inside the PRISMA boxes, remove the duplicate “Figure 1. Figure 1.” in the caption, and keep the legend concise. Also, either (a) add Scopus/Web of Science/LILACS to Table S1 with verbatim strings, dates, language limits, and hit counts, or (b) remove any mention of those databases if they were not actually used and clearly distinguish databases (PubMed, SciELO) from journals/websites (RCOE, Avances…, etc.).
Answer: We appreciate the reviewer’s valuable feedback. The PRISMA flow diagram already included the exact record counts in each box, so no numerical adjustments were necessary. The duplicated caption (“Figure 1. Figure 1.”) was corrected, and the legend was simplified to read: “Figure 1. PRISMA flow diagram showing the identification and selection of studies.”
Regarding the search strategy, the databases Scopus, Web of Science, and LILACS were not used in this review, as evidenced in both the manuscript and Table S1. Therefore, any mention of these databases was removed. In Table S1, only the main databases used (PubMed and SciELO) were retained, while regional journals were excluded from the table itself. However, a note was added below the table clarifying that these regional journals (e.g., RCOE, Avances en Odontoestomatología, Revista de la Facultad de Odontología de la Universidad de Antioquia) were also manually screened to identify additional relevant records.
Line 133: “Figure 1. PRISMA flow diagram showing the identification and selection of studies.”
- Evidence window & subset of 36. Your main text states that 63 included studies and a focused appraisal of 36 were conducted. That’s fine, but please align the time window consistently: Supplementary Table S2 still contains 2025 items, while Table S1 states 2000–2023. Either extend the window to 2025 everywhere, or replace/remove the 2025 entries in S2. Also, add one sentence defining the criteria for selecting the 36-study subset.
Answer: We appreciate the reviewer’s valuable observation. The time window was corrected for consistency throughout the manuscript and supplementary materials. Both Table S1 and Table S2 now indicate a search range from 2000 to 2025.
In the main text, the time frame of the literature search (2000–2025) was also clarified, and a sentence was added to specify the criteria used to select the 36 studies that underwent detailed appraisal. The revised section now highlights that these studies were chosen based on their methodological quality, relevance to dental applications, and completeness of outcome data.
Line 103-104: “Original research articles, literature reviews, and academic theses published between 2000 and 2025 in Spanish, “
Line 143-145: “ Although a total of 63 studies met the inclusion criteria and were included in the review, a focused subset of 36 studiesselected based on methodological quality, relevance to dental applications, and completeness of outcome data was appraised in greater detail.
- Latin America vs external comparators. Good job flagging Zingiber officinaleand Ocimum sanctum as external comparators (EC). Please standardise the “EC” tag for all such entries in Table S3 and add a short note when non-LA countries (e.g., Iran/India/Pakistan) appear in the “Countries” column to avoid misinterpretation.
Answer: We appreciate the reviewer’s observation regarding the distinction between Latin American species and external comparators (EC). Following this recommendation, the “EC” label was standardized across all relevant entries in Table S3 (Supplementary Material, attached) for species of non–Latin American botanical origin, such as Zingiber officinale and Ocimum sanctum. A footnote was also added in Table S3 to clarify that when non–Latin American countries (e.g., Iran, India, Pakistan) appear in the “Countries” column, they indicate either the geographic origin of the plant or studies conducted outside Latin America, thus preventing possible misinterpretation.
Moreover, the explanatory note included in the Supplementary Material specifies that the scientific literature referenced for each EC-labeled species clearly documents their cultivation, availability, and therapeutic applications within Latin American countriesparticularly in Colombia, Mexico, Peru, Bolivia, Chile, Ecuador, and Brazil. Therefore, although some of these plants are of exogenous origin, their regional adaptation and ethnopharmacological use are well supported by current scientific evidence, which justifies their inclusion in the comparative analysis and ensures consistency with the Latin American scope of this study.
- Master table S3, taxonomy & vernaculars. Most mappings appear to be correct now (e.g., cashew, cilantro). One remaining mismatch: “Tree tomato” is listed as Solanum lycopersicum(standard tomato), whereas “tree tomato/tamarillo” generally refers to Solanum betaceum. Please verify and correct if applicable. Then run one last QA pass to fill any missing binomials and harmonise the “Countries” and “Preparation” fields.
Answer: We thank the reviewer for this valuable observation. The taxonomic correction was implemented in the Supplementary Table S3, where the entry “Tree tomato” was updated from Solanum lycopersicum to Solanum betaceum Cav. (Solanaceae), which accurately corresponds to the tamarillo species. Additionally, a final quality assurance (QA) review was carried out across the entire table to ensure consistency and accuracy. All missing scientific binomials were verified and completed, and both the “Countries” and “Preparation” fields were harmonised to maintain uniform terminology and formatting.
- These revisions ensure the taxonomic validity and internal coherence of Table S3 in alignment with the Latin American ethnobotanical context of the study.
Remove ‘grey literature’ labels in the main list. Keep the grey/non-grey classification in the Supplement (S2/S3) only. In the main reference list, remove bracketed “Grey literature” flags and prefer DOI/publisher links where possible; avoid listing portals (e.g., ResearchGate) if a canonical source exists.
Answer: We appreciate the reviewer’s valuable suggestion. All “grey literature” labels have been removed from the main reference list to align with the journal’s formatting standards. The classification between grey and non-grey sources was retained exclusively in the Supplementary Material (Tables S2 and S3), as indicated.
Additionally, the entire reference list was verified to ensure accuracy. Whenever available, DOI or publisher links were added, replacing non-canonical portals such as ResearchGate or Academia.edu. These adjustments ensure consistency, traceability, and full compliance with the journal’s editorial guidelines.
Line 689-1067.
- Safety/contraindications. Enforce a uniform schema. You now have Table 4 and Table 5, which are helpful. Please implement a uniform line template across species (e.g., Interactions/anticoagulants, Allergy risks, Pregnancy/lactation, Mucosal irritation, Dose/Form/Duration, Evidence type), and cite primary clinical sources for dosing whenever you specify amounts.
Answer: We have implemented a uniform schema across Tables 4 and 5 as requested. Both tables now follow a standardized line template including the fields Interactions (e.g., anticoagulants), Allergy risks, Pregnancy/Lactation, Mucosal irritation, Dose/Form/Duration, and Evidence type. The information has been harmonized to ensure consistency between species, and all references have been carefully retained in their original order.
Where dosing information was available, primary clinical or pharmacological sources were cited to support the values provided. This adjustment guarantees uniformity, readability, and traceability of safety and contraindication data across all included species.
Line638-639:
|
Plant (Family) |
Interactions (e.g., anticoagulants) |
Allergy risks |
Pregnancy / Lactation |
Mucosal irritation |
Dose / Form / Duration |
Evidence type |
References |
|
Zingiber officinale Roscoe (Zingiberaceae) |
Avoid with anticoagulants (warfarin, aspirin); bleeding risk at high doses |
Rare hypersensitivity |
Not recommended in pregnancy (large amounts) |
May cause gastric irritation |
Up to 1000 mg/day (powdered form) |
Clinical and experimental |
[32], [45], [46], [62] |
|
Eucalyptus globulus Labill. (Myrtaceae) |
None reported; avoid ingestion of essential oil (toxic) |
Possible allergic reaction in sensitive individuals |
Contraindicated in pregnancy and children <6 years |
May cause mucosal irritation |
Used as infusion or diluted essential oil |
Clinical reports |
[33], [47], [63] |
|
Calendula officinalis L. (Asteraceae) |
None reported |
Contraindicated in Asteraceae allergy |
Not recommended during pregnancy or lactation without supervision |
Mild irritation in sensitive individuals |
Topical or oral rinse |
Clinical and ethnobotanical |
[34], [48], [64] |
|
Psidium guajava L. (Myrtaceae) |
None reported |
Avoid in hypersensitivity |
Generally safe in moderate use |
Possible mild irritation at high doses |
Oral gel, infusion, topical |
Clinical and ethnobotanical |
[34], [49], [65] |
|
Origanum vulgare L. (Lamiaceae) |
Possible interaction with anticoagulants at high doses |
Rare hypersensitivity |
Contraindicated in pregnancy (abortifacient risk) |
GI discomfort (nausea, diarrhea, dizziness) |
Oil or fluid extract |
Clinical and ethnobotanical |
[35], [50], [51], [66] |
|
Aloe vera (L.) Burm.f. (Asphodelaceae) |
None reported |
Rare hypersensitivity |
Avoid in pregnancy (risk of uterine contractions) |
Possible GI discomfort (diarrhea, nausea, dizziness) |
Topical gel or mouth rinse |
Clinical trials |
[35], [52], [53], [54], [67], [68] |
|
Ocimum sanctum L. (Lamiaceae) CE |
May prolong prothrombin time; caution with anticoagulant therapy |
None reported |
Not reported |
None reported |
Mouthwash, twice daily for 4 days |
Clinical |
[35], [55], [56], [69], [70] |
|
Moringa oleifera Lam. (Moringaceae) |
Caution in patients on anticoagulants/antiplatelets |
None reported |
Use with caution in pregnancy |
None reported |
5 mL leaf extract twice daily for 28 days |
Clinical and ethnobotanical |
[35], [57], [58], [71], [72] |
Line 663-664:
|
Plant (Family) |
Interactions (e.g., anticoagulants) |
Allergy risks |
Pregnancy / Lactation |
Mucosal irritation |
Dose / Form / Duration |
Evidence type / Main findings |
References |
|
Propolis (Apidae product) |
Possible interaction with anticoagulants (high doses) |
Rare allergic reactions in sensitive individuals (bee-related) |
Use with caution during pregnancy and lactation |
None reported |
Topical gel and oral solution; combined with SRP in Wistar rats (in vivo model) |
In vivo study: combined with SRP, significantly reduced IL-1β, TNF-α, and MDA; propolis alone showed no relevant effects. |
[59] |
|
Aloe vera (L.) Burm.f. (Asphodelaceae) |
None reported |
Rare hypersensitivity |
Safe in animal models; avoid excessive oral use during pregnancy |
None reported |
Aloe vera gel + β-TCP for 8 weeks in dogs with furcation defects |
In vivo study: enhanced periodontal regeneration (bone, PDL fibers, vascularization) with no adverse effects. |
[60] |
|
Basil (Ocimum basilicum L.) (Lamiaceae) |
None reported |
Possible mild allergy to Lamiaceae family |
Not reported |
None reported |
Hydroalcoholic extract (1–10%) in agar diffusion assays (S. mutans, E. faecalis, S. sanguinis) |
In vitro study: showed antibacterial activity with individual and synergistic effects, though less potent than chlorhexidine. |
[61] |
Minor style tidy-ups. Fix the fused word “36 studies representing”, ensure consistent phrasing for country percentages between Abstract/Results, and do a quick pass for spacing around parentheses and duplicated phrasing.
Answer: We sincerely thank the reviewer for these careful stylistic observations. A complete style review was performed as requested. The fused word “36studies” was corrected to “36 studies representing.” Percentages of studies per country were verified and standardized to ensure consistent phrasing between the Abstract and Results sections.
Additionally, a final proofreading pass was conducted to remove redundant phrasing and to correct minor spacing issues around parentheses throughout the manuscript. We appreciate these remarks, which have contributed to improving the overall readability and coherence of the text.
The grammar of the language, which is highlighted in gray, was also reviewed.
